# Decoding gray matter, large-scale analysis of brain cell morphometry to inform microstructural modeling of diffusion MR signals
Charlie Aird-Rossiter [1,2] ✉, Hui Zhang [3], Daniel C. Alexander [3], Derek K. Jones[1] & Marco Palombo [1,2] ✉

Grey matter structure is central to neuroscience, as cellular morphology varies by type and is influenced by neurological conditions. Understanding these variations is essential for studying brain function and disease mechanisms. Diffusion-weighted MRI (dMRI) offers a non-invasive way to examine cellular microstructure, but its accuracy depends on identifying which morphological features influence its measurements. Despite increasing interest, no systematic study has defined the key neural cell features relevant to dMRI interpretation. Here, we analyzed over 11,800 three-dimensional cellular reconstructions across three species and nine cell types, establishing reference values for critical traits grouped into structural, shape, and topological categories. We also identified which of these traits are most relevant to dMRI sensitivity. In addition, we provide high-resolution 3D surface meshes representative of each cell type and species. These meshes, compatible with Monte Carlo simulators, offer a valuable resource for modeling and interpreting gray matter microstructure.

Despite the widespread use of biophysical models in diffusion-weighted Magnetic Resonance Imaging (dMRI) to infer cellular-scale structure, gray matter (GM) remains poorly understood due to a lack of ground-truth morphological data. Unlike white matter (WM), where axonal morphology is well characterized, GM cellular features—critical for accurate dMRI modeling—are rarely quantitatively analysed. This knowledge gap leads to overly simplistic or unfounded model assumptions, potentially reducing the reliability of microstructural imaging in GM. Here, we address this challenge directly by systematically characterizing the statistical distribution of key morphological features across different species, providing a much-needed empirical foundation for improving GM dMRI models.

Grey matter is composed of a range of cells, mainly differentiated into neuronal and glial cells, featuring a plethora of morphological characteristics. Neurons are fundamental functional units of the nervous system, specialized in the transmission and integration of electrical and chemical signals within the brain. Supporting the neurons are the glial cells (e.g., astrocytes, microglia and oligodendrocytes) that are crucial in maintaining health and functionality.

First studied in depth by Ramon y Cajal[1], neuronal morphology offers insights into the complex structure and function of the brain.

Neural cells exhibit a remarkable diversity in shape and size, each adapted to its specific function[2]. They can be classified based on morphological, molecular, and physiological characteristics[3]. Both neurons and glial cells share a common structural organization, consisting of a central soma and branching projections, yet they vary significantly across brain regions[4,5]. For instance, Purkinje cells in the cerebellum display intricate dendritic arborization, whereas granule cells in the cerebral cortex exhibit a much simpler morphology. Furthermore, neurons of the same cell type can exhibit significant morphological differences across brain regions[6], and even display substantial variability between cortical layers within the same region[7].

The brain contains approximately 86 billion of these neural (neural refers to the nervous system broadly (neurons and glia), whereas neuronal refers specifically to neurons) cells[8]. Cortical GM is composed of 10–40% cell bodies (soma) of neural cells; 40–75% neurites: neuronal dendrites, short-range intra-cortical axons, the stems of long-range axons extending into the WM and glial cell projections which intermingle with each other to form a

[1]Cardiff University Brain Research Imaging Centre (CUBRIC), School of Psychology, Cardiff University, Cardiff, UK. [2]School of Computer Science and Informatics, Cardiff University, Cardiff, UK. [3]UCL Hawkes Institute and Department of Computer Science, University College London, London, UK. ✉e-mail: aird-rossiterc@cardiff.ac.uk; palombom@cardiff.ac.uk

dense and complex network; 15–30% highly tortuous extra-cellular space (ECS); and 1–5% vasculature[9–11]. In adults, the glia to neuron ratio is 1.32/1.40 for males/females respectively. The proportion of glial cells (by cell count) was estimated to be 77% oligodendrocytes, 17% astrocytes and 6% microglia[12,13].

The ECS occupies a volume fraction of 15–30% in normal adult brain tissue, with a typical value of 20%, that falls to 5% during global ischemia (the expected state during classical fixation)[14]. The ECS an average tortuosity (defined as the ratio between the true diffusion coefficient and the effective diffusivity of small molecules such as inulin and sucrose) of 2–3[14], due to its labyrinthine porous matrix, the presence of long-chain macromolecules, transient trapping in dead-space microdomains, and transient physical-chemical interaction with the cellular membranes. The average neuron-microvessel distance in brain GM is 20 $\mu m$[15].

The morphology of neurons and glia is revealed through staining techniques such as confocal or electron microscopy[16–18], which can image cellular structures with very high resolution (down to a few nanometers)[19].

Currently, there are no methods to directly observe cellular microstructure in vivo, as its scale (measured in micrometers) exceeds the resolution of clinical MRI, which typically operates at the millimeter scale[20]. While non-invasive imaging techniques like MRI can provide insights, they cannot directly capture cellular-level details. With many neurological conditions, such as dementia[21], as well as aging[22], altering the brain structure on this cellular scale, there is a strong incentive to develop means of revealing the morphology of neural cells, in-vivo.

Although dMRI has still millimeter-scale resolution, it is sensitive to micrometer-scale structures by measuring the diffusion of endogenous molecules (e.g., water). This makes it a promising technique for overcoming MRI's resolution limits and characterizing the brain's *microstructure* (i.e., its cellular-scale organization) in vivo. However, dMRI's sensitivity to microstructure is indirect, and biophysical modelling of the brain tissue and the subsequent interpretation of the dMRI signal is essential to quantify histologically meaningful features of the cellular structure, and gain specificity to their changes. To this end, the *microstructure imaging* paradigm has been introduced over a decade ago[23]: the approach fits a biophysical model voxel-wise to the set of signals obtained from images acquired with different sensitisations to tissue microstructure, yielding maps of model parameters that it is hoped are proxies of the corresponding underlying microstructural features.

Successful examples of the microstructure imaging paradigm include Neurite Orientation Dispersion and Density Imaging (NODDI)[24] and the White Matter Tract Integrity (WMTI)[25] to characterise the diffusion of water within WM, revealing insight into the structure of axonal bundle tracks and other anatomical features, such as axon diameter[26,27].

Building on the success of dMRI in WM, there has been growing interest in applying it to GM to characterize cellular morphology in vivo[28–34]. dMRI has already been used to distinguish different cortical regions and reveal laminar structures within GM[35]. Significant effort has been made to develop also models to better describe the diffusion signal within GM and reveal anatomical information about the microstructure, such as the Soma And Neurite Density Imaging (SANDI)[32] to characterize soma and neurite density, the Neurite EXchange Imaging (NEXI)[33] and the Standard Model with EXchange (SMEX)[34] to characterize the water permeative exchange between neurites and extracellular space, and combinations of these, such as SANDI with exchange (SANDIx)[34].

While substantial effort has been made to design, validate, and translate to clinics biophysical models for WM[23,36,37], the GM counterpart is lagging. This disparity stems from the greater complexity of the tissue, which renders the design of biophysical models for GM microstructure imaging more challenging[38].

Building accurate dMRI models for GM requires a fundamental understanding of its microstructural features. Key questions remain: what morphological properties influence the dMRI signal? How can they be measured reliably? How should these properties guide model development? With cellular morphology varying by cell type as well as brain region, there is a necessity for a thorough analysis of characteristics based on these criteria.

Here, we aim to correct this imbalance with a comprehensive analysis of GM cellular morphology, looking at structural and topological morphology, and shape descriptors of 11,850 real three-dimensional (3D) reconstructions from mouse, rat, monkey, and human brain cortex. To complement the statistical analysis, we provide 50 high resolution 3D surface meshes for each cell type and species; fully compatible with Monte Carlo simulators, offering a valuable resource for the modelling community.

The paper is organized as follows. We first provide quantitative information on the anatomy of brain GM tissue at the cellular scale. We characterize the morphology of neural cells using structural, topological and shape descriptors. We then review the range of dMRI measurements and biophysical models available to probe this anatomy and highlight limitations and caveats, ultimately providing guidelines on how to model GM microstructure from dMRI signals.

## Materials and Methods
### Microscopy dataset
In order to measure characteristic features for specific cell types, the open access repository Neuromorpho.org[39,40] which has a comprehensive range of cellular reconstructions available was used. We downloaded and analysed three dimensional reconstructions of 11,850 brain cells, from mouse, rat, monkey, and human, in the form of SWC files (an ASCII text-based file that describes three-dimensional neural morphology). The SWC file defines a set of labelled nodes connected by edges characterising the three-dimensional structure of each cell Fig. 1.

Nine representative cell-types were acquired: microglia, astrocyte, oligodendrocyte, pyramidal, granule, purkinje, glutamatergic, gabaergic, and basket cells, from mouse/rat ($N = 9001$), monkey ($N = 525$), and human ($N = 2324$). Although the NeuroMorpho database contains over 270,000 cellular reconstructions, only 11,850 satisfied our inclusion criteria: healthy controls; having complete reconstruction of soma and all the dendrites; containing information about diameters and angles; and being 3D reconstructions (see Supplementary Fig. 1). Our quality assessment criteria include: consistent estimates of dendritic diameters (e.g., most of the rejected reconstructions had a fixed nominal diameter for all the branches instead of the real one); continuity of the cellular processes.

Despite their essential role in synaptic formation and the brains micro connectivity, dendritic spines (small protrusions on neuronal dendrites that form synaptic connections) were largely absent from the reconstructions. Furthermore, their inclusion biases the statistical analysis of overall cellular morphology, such as artificially reducing branch length and inflating branch order. As a result, for all reconstructions spines were identified (if present) and removed. Additionally, due to the inconsistency in axonal reconstructions (axons largely being absent or heavily truncated) in the reconstructions, they were not considered in the analysis (if axonal components were present, they were identified from the SWC format and removed from the reconstruction). As a result, the statistics of the projections are for dendritic projections only.

The cellular reconstructions were analysed in Matlab using custom scripts, exploiting functions from validated suites (TREES[41], Blender[42], ToolboxGraph[43]). All the codes and SWC files used in this work will be made publicly available on GitHub https://github.com/Charlie-Aird/Decoding-Grey-Matter upon publication.

### Structural descriptors
The structural analysis describes the constituent parts of the cell, including the soma and cellular projections, offering vital information about the cell's fundamental structure, such as the effective soma radius and the branch angle between daughter branches.

For the structural analysis, such features or descriptors, determined to be crucial to microstructure modelling based on current literature[32,34,44–52], were estimated from the acquired cellular reconstructions. This set of features allows for a deeper understanding of how each fundamental aspect of

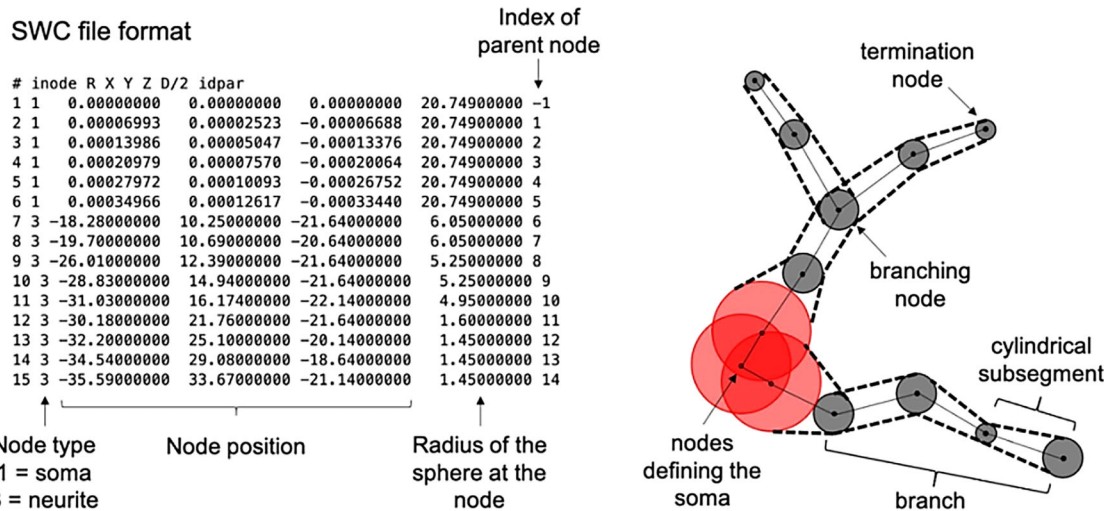

**Fig. 1 | An example of SWC file and how it relates to the cellular geometry.** We highlight the structural elements used to estimate the morphological features. Note that the first node in the SWC file is the so called `root`. It often coincides with the soma's centre and it is used to compute metrics.

brain cell morphology influences the diffusion of molecules within the intracellular space. For instance, the size of the soma and cellular projections can provide insights into the characteristic length scales of intracellular restrictions, while the branching, tortuosity, undulation, and calibre variation of the projections can inform on time-dependent diffusion processes. Additionally, the surface-to-volume ratios of the soma and projections offer valuable information on exchange dynamics.

The features of the soma and cellular projections differ significantly, and our analysis accounts for this by organizing the structural descriptors into relevant categories. These categories are: (1) *soma* (the characteristics defining the cell body); (2) *projections* (the set of characteristics defining the cellular projections' structure as interconnected branches), and (3) *general* (describing the general cellular characteristics).

*General*
- $R_{domain}$: the extent of the cellular domain
- $N_{projection}$: the number of primary projections radiating from the soma
- $BO$: the degree of branching of the cellular projections
- $S/V_{domain}$: the surface-to-volume ratio of the complete cellular structure (soma and branches)

*Soma*
- $R_{soma}$: the effective radius of the soma (the radius of a sphere of equivalent volume as the soma)
- $R_{MRsoma}$: the effective MR radius of the soma radii distribution
- $\eta_{soma}$: the proportion of the surface area covered by projection interfaces
- $S/V_{soma}$: the soma surface-to-volume ratio

*Projections*
- $<R_{branch}<_s$: the mean effective radius of segments along branch s
- $<R_{MRbranch}<_s$: the effective MR radius of the branch radii distribution
- $CV_{branch}$: a measure of branch beading
- $L_{branch}$: the branch length
- $S/V_{branch}$: the branch surface-to-volume ratio
- $<\mu OD_{branch}<_s$: a measure of mean branch undulation
- $<R_c<_s$: a measure mean branch curvature
- $\tau_{branch}$: a measure of branch tortuosity
- $\theta_{branch}$: the angle formed by two bifurcating branches

Figure 2 illustrates these descriptors, and a summary of their definitions is reported in Table 1.

From the information in the cellular reconstruction SWC file, the nodes/edges defining the projections from those belonging to the soma were

separated: all the nodes and corresponding edges within the nominal soma radius (radius of the first node in the SWC file) from the first node, namely the 'root', were assigned to the soma; the remaining ones to the projections.

Using this soma threshold the cell was resampled, preserving the nodes that lie within the soma threshold. From this resampled soma the 3D surface mesh was constructed using Blender, and the volume, $V_{soma}$, and surface, $S_{soma}$, of the soma calculated. The soma volume can be expressed in terms of the effective soma radius $R_{soma}$.

From the distribution of soma radii the effective MR soma radius was calculated using the following equation[34],

$$R_{MRsoma} = ( < R_{soma}^7 > / < R_{soma}^3 > )^{(1/4)}$$

Additionally, we calculated also the soma surface-to-volume ratio ratio as $S_{soma}/V_{soma}$; and the fraction of the soma surface covered by the cellular projections as sum of the projection connection area divided by the soma surface area ($\eta_{soma}$).

From the nodes/edges defining the projections, the individual branches composing the cellular projections were identified, delimited by either branching or termination nodes. The individual branches are comprised of cylindrical sub-segments, defined by the edges and their associated radius. The central line defined by these sub-segments defines a curvilinear path, $s$. From this path, $s$, metrics for the projections features were computed. Branch beading is reported as the coefficient of variation of the branch radius along the branch length ($CV_{branch}$). The branch surface-to-volume ratio ($S/V_{branch}$) was determined as the sum of the sub-segments area divided by the sum of the sub-segments volume. Branch undulation ($<\mu OD_{branch}<_s$) is calculated as the mean angle subtended by the vector of the individual sub-segments and the vector made by the branch start and end points.

Branch radius, $R_{branch}$, is reported as the average branch radius along path $s$. From the distribution of branch radii the effective MR branch radius was calculated using equation[27],

$$R_{MRbranch} = ( < R_{branch}^6 > / < R_{branch}^2 > )^{(1/4)}$$

Finally, the general metrics were computed. The number of projections $N_{projection}$ was found by identifying the number of branches that cross the soma threshold. And the branch order $BO$ is defined as the number of consecutive bifurcations of the cellular projections.

We provide the first, second, and third quartiles for each structural descriptor for each cell type for each species analysed. Moreover, we provide value distributions for features of relevance to biophysical modeling of diffusion in GM, such as $R_{soma}, L_{branch}, S/V_{branch}$ and $S/V_{domain}$.

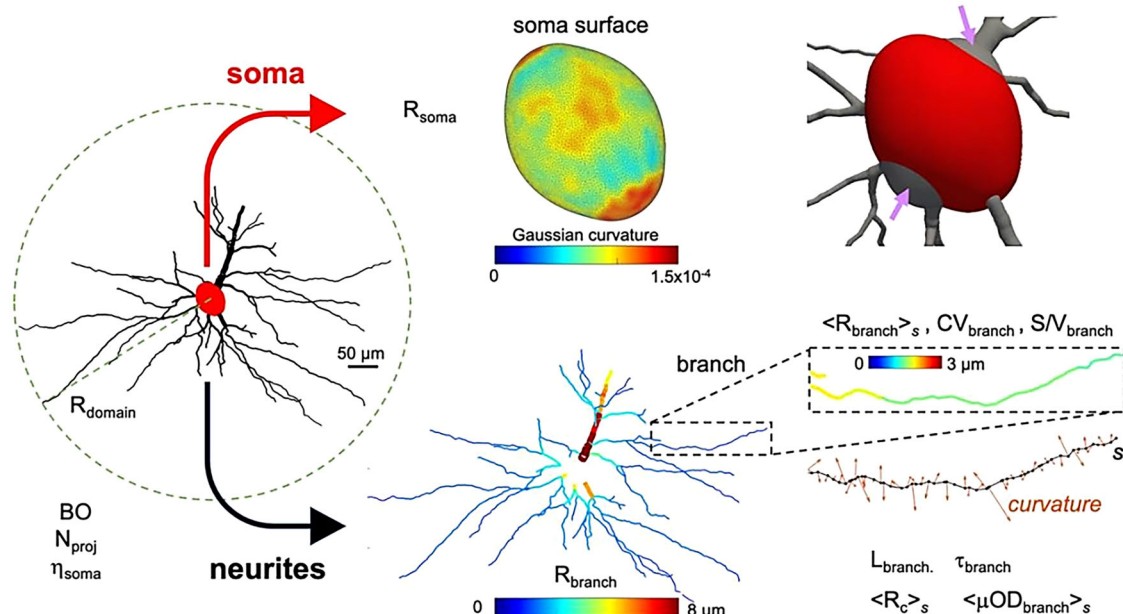

**Fig. 2 | Illustration of the structural descriptors investigated for an exemplar cell.** We estimated general features of the whole structure and separated soma from projections, processing them individually to estimate a set of other relevant features. Additionally, we display the Gaussian curvature of the soma surface to show that it is a non-spherical geometry (always positive but not constant). A limitation of the current approach (and the majority of existing tools[86–88]) is the slightly inaccurate definition of the soma surface, as shown in the top right corner (arrows).

**Table 1 | Definition of the structural descriptors investigated**

| Morphological feature | Definition |
|---|---|
| $R_{domain}$ | Distance of the furthest node from the soma |
| $N_{proj}$ | The number of primary projections radiating from the soma |
| $BO$ | The number of consecutive bifurcations of the cellular projections |
| $S/V_{domain}$ | Ratio between the complete (soma and branches) cellular surface and cellular volume |
| $R_{soma}$ | Radius of sphere of volume equivalent to the soma volume |
| $R_{MRsoma}$ | Effective MR soma radius[34] |
| $\eta_{s}oma$ | Ratio between the total cross-sectional area of the $N_{proj}$ primary projections and the total surface area |
| $S/V_{soma}$ | Ratio between the soma mesh surface and soma mesh volume |
| $S/V_{branch}$ | Ratio between the sum of the surfaces of all the subsegments in s and the sum of their volume |
| $< L_{branch} >$ | Mean of the Sum of subsegments' length in s |
| $<R_{branch}>_s (<R^2_{branch}>_s)^{1/2}$ | Mean (standard deviation) of subsegments' radius along s |
| $R_{MRbranch}$ | Effective MR branch radius $(<R^6_{soma}>/<R^2_{soma}>)^{(1/4)}>$ [34] |
| $CV_{branch}$ | Coefficient of variation of branch radius |
| $< \mu OD_{branch} >_s$ | Mean microscopic orientation dispersion of subsegments along s |
| $< R_c >_s$ | Mean radius of curvature of s |
| $\tau_{branch}$ | Ratio between distance between ends of s and $L_{branch}$, branch tortuosity is $\tau^{-1}_{branch}$ |

We also estimated intracellular residence times, $\tau_{ic}$, based on the surface-to-volume ratio of the whole cell ($S/V_{domain}$) and the individual branches ($S/V_{branch}$) using the following equation[53]:

$$\tau_i = \frac{1}{S/V\kappa}$$

where $\kappa$ is the membrane permeability, and corresponding exchange times[54]:

$$\tau_{ex} = \tau_i f_{ec}$$

where $f_{ec}$ is the extracellular volume fraction, assumed to be 30%.

## Shape descriptors

Many diffusion MRI models represent cellular structure as a collection of randomly oriented cylinders[24,55]. Initially applied to the signal from white matter where axons can be simplified to a collection of sticks, this model has been shown to apply to the dendrites of neurons[32,33] and has been incorporated into current grey matter diffusion MRI-based models.

To compute the fractional anisotropy (FA) for the cellular structure, the cellular structure was modeled as a collection of cylinders, following the method described in ref. 46. First, the cells were decomposed into primary projections and further segmented into cylinders of length 10 $\mu m$ (Fig. 3). FA is calculated by first computing the scatter matrix of the weighted line segments orientations (weighted by volume), from which the eigenvalues

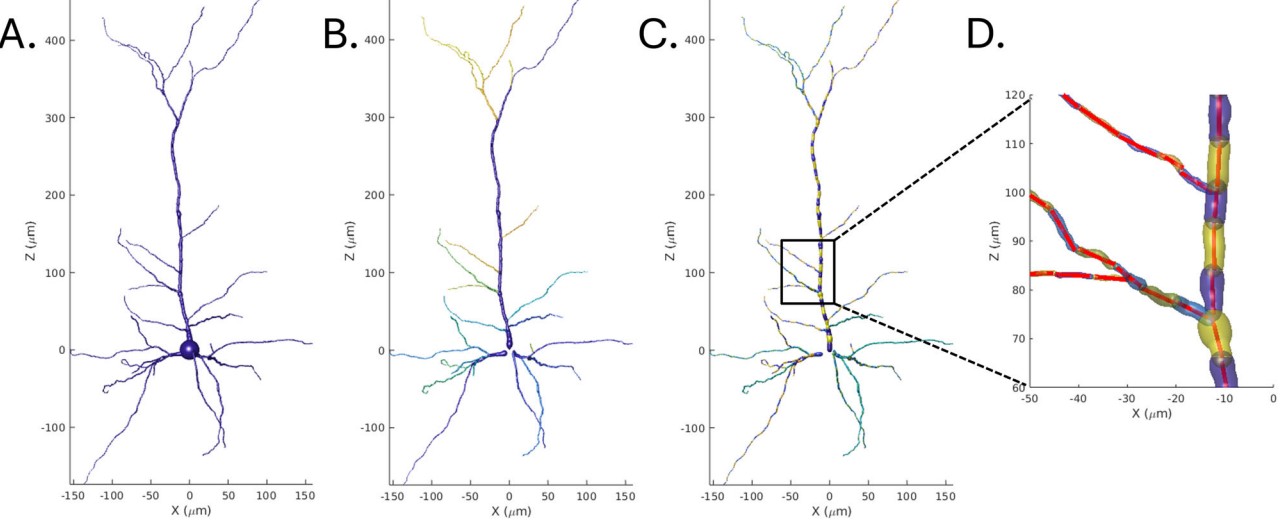

**Fig. 3 | A demonstration of procedure used to decompose the cellular structure into a set of average lines segments. A** the complete cell. **B** Cell decomposed into topological persistence components. **C** Cell further decomposed into 10 $\mu m$ segments. **D** Average line segments fitted to cell segments.

$(\tau_i)$ are derived. FA is then calculated as:

$$FA = \sqrt{\frac{3}{2} \frac{(\tau_1 - \tau)^2 + (\tau_2 - \tau)^2 + (\tau_3 - \tau)^2}{\tau_1^2 + \tau_2^2 + \tau_3^2}}$$

In some cellular reconstructions, limitations in depth of field resulting from the imaging technique lead to anisotropic inaccuracies (see Supplementary Fig. 2), with cells appearing compressed along the axis perpendicular to the acquisition plane (Z-axis). This artifact can result in overestimated fractional anisotropy in said reconstructions. For this reason, alongside the FA, we also report the estimated eigenvalues in ascending order ($\tau_1 \leq \tau_2 \leq \tau_3$) and an adjusted FA (Adj.FA) where we assumed $\tau_1 = \tau_2$.

The mean and standard deviation are reported for all cell types.

In addition to the FA of the line segments, their orientation dispersion (OD) was also computed. OD provides additional insight into the degree of anisotropy in cellular structures, complementing FA by describing the variability in the orientations of the neuronal projections. The variability is typically characterised through a Watson distribution, which is a probability distribution of orientations around the primary axis on the unit sphere[56]. With the degree of clustering defined by the concentration parameter, $\kappa$. This concentration parameter can be used to calculate the OD through the following equation[24].

$$OD = \frac{2}{\pi} arctan(1/\kappa)$$

For cases where the data is not axially symmetric, such as cellular reconstructions with limited depth, the Watson distribution is insufficient. Instead, we applied the more general Bingham distribution, using MATLAB toolbox libDirectional[57] (Fig. 4), which accounts for orientation variability along multiple axes. The Bingham distribution models anisotropic orientation data and provides two concentration parameters ($\kappa_1$ and $\kappa_2$), corresponding to the clustering along two orthogonal directions.

From the resulting concentration parameters, if the first concentration parameter was significantly larger than the second $\kappa_1 <> \kappa_2$, indicating the orientation data was highly planar, $\kappa_2$ was used as the Watson distribution parameter $\kappa$. Otherwise, the average of $\kappa_1$ and $\kappa_2$ was used as the Watson distribution parameter $\kappa$.

This approach provides a robust calculation of the Watson distribution across isotropically and anisotropically oriented data sets.

## Topological descriptors

Another way to characterize cells is by analysing their topology, which captures the complexity of their branching structures. One approach to this is the Topological Morphology Descriptor (TMD)[58], which computes a topological persistence barcode for a given structure. This barcode provides a compact representation of the cell's branching architecture and has proven effective in classifying neurons[58]. In this study, we use the TMD to quantify cellular projections and compare topological structures across different cell types and species.

Topological persistence quantifies how long specific structural features, such as branch paths, remain across different scales. The TMD captures this by measuring the path lengths of connected branches, tracking both their initiation and termination points relative to the soma[58]. This process preserves longer (more persistent) structural components while filtering out shorter ones.

To compute a neuron's topological descriptor, terminal points are evaluated by their path length from the soma. At each branching point, the shorter of the two sibling branches is removed, and its initiation and termination distances from the soma are recorded in the topological barcode. The longer branch is retained, and this process is repeated until all terminal points have been assessed. The resulting barcode is a multiset of value pairs, each representing the initiation and termination lengths of a branch segment relative to the soma. The longer branch is retained, and this process is repeated until all terminal points have been assessed. The resulting barcode is a multiset of value pairs, each representing the initiation and termination points with respect to the soma.

These barcodes retain rich structural information and can be used to categorize and identify cell types[59]. In our work, each cell was first decomposed into its principal projections by identifying dendritic projections emanating from the soma. Persistence barcodes were then computed for each projection and used to generate persistence images (Fig. 5), which visualize the distribution of persistent features by their initiation and termination lengths. To improve visual smoothness, kernel density estimation was applied, and each image was normalized so that the sum of its pixel values equaled one.

Comparisons were made between persistence images by computing the global topological distance[58], D, between images,

$$D_{celltype1,celltype2} = \sum_{i=1}^{n} |celltype1_i - celltype2_i|$$

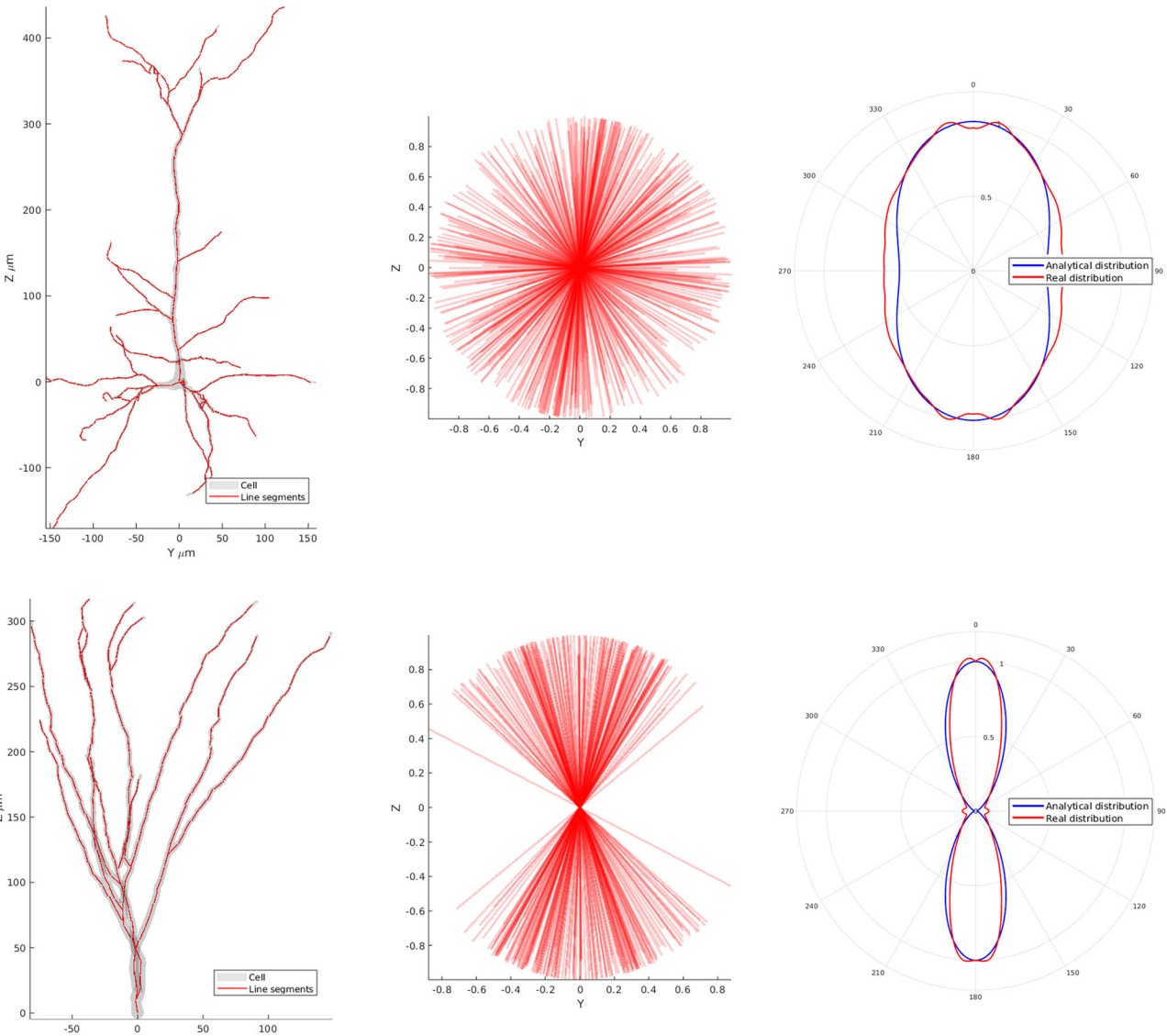

**Fig. 4 | A comparison between two cell types, mouse/rat pyramidal and granule cells.** Showing exemplar cells overlaid with decomposed line segments, the line segments centered at the origin, and orientation distribution about the z axis of the line segments and the analytical distribution given the calculated Watson concentration parameter.

where *celltype*1 and *celltype*2 are the two persistence images being compared. With D ranging from zero, the persistent images are identical and over lab completely, to two, the persistent images share no overlap or similarity. This topological distance was calculated for cell types within species and also cell types between species.

### Statistics and reproducibility

In total 11,850 cellular reconstructions were alanlysed in this study, spanning three species and nine cell types. We included all cellular reconstructions that satisfied our search criteria and were available at the time of download (Neruomorpho v8.6.83).

Custom Matlab scripts, utilising the TREES toolbox and Blender, were used to extract morphological features for individual reconstructions. Features were grouped into three categories (structural, shape, and topological). First, second and third quartiles are reported for structural and shape features. Additionally, the relationship between structural features was analysed by calculating the Spearman's rank correlation coefficient with Bonferonni correction. The significance of the topological distances was computed using bootstrapping ($n = 1000$).

### Reporting summary

Further information on research design is available in the Nature Portfolio Reporting Summary linked to this article.

## Results

### Morphological features' reference values

A summary of the typical values of structure features is given in Table 2. Some features show little variation between cell type and species ($< R_{branch} <_s$, $CV_{branch}$, $<\mu OD_{branch} <_s$, and $\tau_{branch}$). The remaining features displayed a wide range of values, suggesting higher inter-cellular and intra-species variability.

### Comparing neuronal and glial cells

Given the growing interest in differentiating neurons and glial cells, the mean values of the structural features were computed across only neuronal and glial cells and reported in Table 3. The findings show, compared to glial cells, neurons have larger soma and cell domain and longer branches; and reduced branching, branch curvedness, number of primary projections, and proportion of soma surface covered by projections; the remaining features displayed similar values.

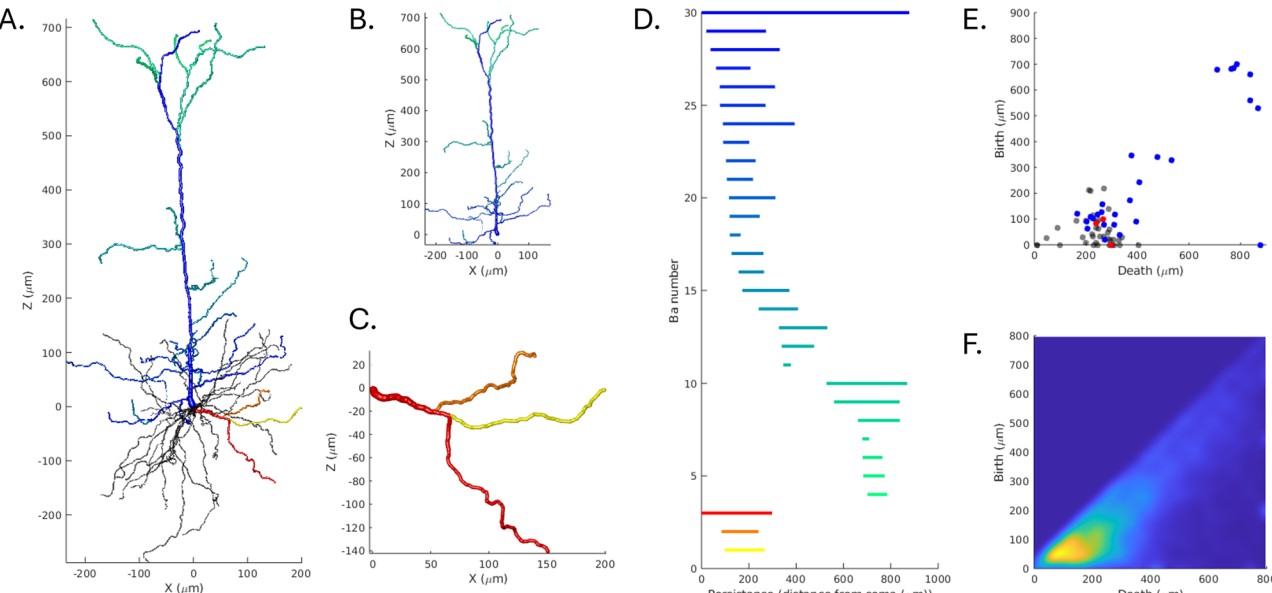

**Fig. 5 | A representation of the process of decomposing a cellular structure (here a mouse/rat pyramidal cell) into its corresponding topological persistence bar code for an apical and basal projection. A** The complete cellular structure of an exemplar cell, apical projection in blue, and exemplar basal projection in red. **B**, **C** Detail of apical and basal projections being assessed. **D** Corresponding barcodes for apical (in blue) and exemplar basal (in red/orange) components. **E** persistence diagram of the complete cell, (blue points corresponding to apical and red points to the exemplar basal projection, black points for the remaining projections). **F** The resulting persistence image for all mouse/rat pyramidal cells.

## Correlations between morphological features

The correlations between structural features are illustrated in Supplementary Fig. 4. Some consistent and expected patterns are observed, particularly involving surface-to-volume ratio measures: the soma surface-to-volume ratio is obviously negatively correlated with soma radius, and the branch surface-to-volume ratio is obviously negatively correlated with branch radius. Furthermore, the surface-to-volume ratio of both the soma and branches are obviously positively correlated with the surface-to-volume ratio of the domain. Additional, micro-orientation dispersion shows a positive correlation with branch tortuosity, $\tau$, across majority of cell types. No further consistent correlations between features were observed across cell types.

## Cellular shape reference values

A summary of the shape descriptors typical values for each cell type and species is reported in Table 4. Cells with highly oriented and polarized projections, such as Purkinje cells, granule cells and pyramidal neurons have high FA and adjFA and low orientation dispersion; while most of the glial cells projections are highly dispersed and with low FA and adjFA.

## Distribution of some structural features of interest for dMRI modelling

The full distributions of values for some structural features of interest, obtained by merging together all the estimated values from all the species for each cell type, are shown in Fig. 6. Given the increasing interest of the diffusion-based microstructural imaging community in estimating glia microstructure and the striking difference between the neuronal components of cerebral and cerebellar cortices (e.g., Purkinje cells only in cerebellum), we decided to group them into three classes: glia, cortical neurons and cerebellar neurons, and use three different colors to simplify the visualization of the results. The distributions for all the morphological features are reported in Supplementary Fig. 3.

Figure 7 shows the distributions of intra-cellular and intra-branch residence times (grouped by general cell types) for two representative membrane permeabilities (low and high), as well as mean residence and exchange times as a function of different values of cell membrane permeability.

## Topological distance between cell types

The persistence maps of all the cell types for all species together are shown in Fig. 8. These persistence maps are used for each species and each cell type to estimate cell-type specific topological distances, shown in Fig. 9. Under bootstrapping of $n = 1000$ iterations, it was found all distances were statistically significant between cell types, except basket and glutamatergic rodent cells with a $p = 0.16$.

In rodent cells, there is a notably high global topological distance between glial cells and neurons, indicating a fundamentally different topological organisation between them. This difference in topology becomes even more evident when considering the comparison along the same length scale. Since glial cells are significantly smaller in size, they inherently display a much smaller topological persistence. This size-related constraint underscores their distinct spatial and structural properties relative to neurons.

Within rodent neurons, granule cells stand out by exhibiting a higher topological distance compared to other neuronal types. This disparity is likely related to their characteristically low branch number, which reflects their simpler dendritic structures and reduced connectivity compared to more complex neurons.

In human cells, however, no significant or consistent trends in topological distance are observed, suggesting greater variability or less pronounced differences in cellular topology among the various neuronal cell types.

When examining hominid cells, which include both monkey and human samples, microglia demonstrate a higher topological distance compared to neurons. This observation reinforces the trend seen in rodent cells, further supporting the notion that glial cells maintain a distinct and conserved topological profile across different species.

In the cross-species comparison between rodent and hominid cells, microglia consistently show a high topological distance relative to neurons, mirroring the trend observed within each species. Additionally, the comparatively low topological distance for microglia across species suggest that their topological properties are consistent, pointing to a shared structural and functional organization in microglia across species.

## Table 2 | Summary of the morphological features computed for each species and cell type

| Celltype | Animal | N | $R_{domain}$ Q1 | med | Q3 | $N_{proj}$ Q1 | med | Q3 | BO Q1 | med | Q3 | $S/V_{domain}$ (µm⁻¹) Q1 | med | Q3 | $S/V_{soma}$ (µm⁻¹) Q1 | med | Q3 | $R_{soma}$ (µm) Q1 | med | Q3 | $R_{diff\,soma}$ (µm) Q1 | med | Q3 | $n_{soma}$ (%) Q1 | med | Q3 | $S/V_{branch}$ (µm⁻¹) Q1 | med | Q3 | $L_{branch}$ (µm) Q1 | med | Q3 | $R_{branch}$ (µm) Q1 | med | Q3 | $CV_{branch}$ Q1 | med | Q3 | $R_{diff\,branch}$ (µm) Q1 | med | Q3 | $\mu OD_{branch}$ Q1 | med | Q3 | $R_c$ (µm) Q1 | med | Q3 | $\tau_{branch}$ Q1 | med | Q3 |
|---|---|---|---|---|---|---|---|---|---|---|---|---|---|---|---|---|---|---|---|---|---|---|---|---|---|---|---|---|---|---|---|---|---|---|---|---|---|---|---|---|---|---|---|---|---|---|---|---|---|---|---|
| Microglia | Mouse/Rat | 4100 (4072) | 33 | 41 | 49 | 3 | 5 | 6 | 3.7 | 7.3 | 9.5 | 2.6 | 3.3 | 3.6 | 0.10 | 1.3 | 2.0 | 1.6 | 2.5 | 3.1 | 3.1 | — | 3.5 | 4 | 48 | 99 | 2.3 | 3.4 | — | 6.4 | 7.4 | 9.8 | 0.67 | 1.0 | 1.6 | — | 0.58 | — | 1.0 | — | — | 0.19 | 0.2 | 0.25 | 3.4 | 3.9 | 7.5 | 0.77 | 0.81 | 0.82 |
| | Monkey | 60 (60) | 37 | 42 | 47 | 7 | 8 | 9 | 3.3 | 3.8 | 9 | 2.9 | 3.9 | 4.9 | 1 | 1.2 | 1.3 | 2.3 | 2.6 | 3.0 | 3.0 | — | 3 | 0.7 | 1.1 | 1.6 | 15 | 17 | 18 | 8.9 | 9.4 | 9.9 | 0.18 | 0.2 | 0.36 | — | 0.63 | — | 0.2 | — | — | 0.25 | 0.27 | 0.29 | 13 | 16 | 18 | 0.73 | 0.74 | 0.75 |
| | Human | 310 (307) | 32 | 39 | 45 | 5 | 7 | 12 | 3.1 | 4 | 10 | 0.37 | 4.1 | 5.5 | 0.8 | 0.9 | 1.2 | 2.5 | 3.5 | 4.0 | 4.0 | — | 4.3 | 3 | 27 | 70 | 4.2 | 6.7 | — | 6.5 | 7.8 | 9.3 | 0.46 | 0.78 | 1.6 | — | 0.93 | — | 0.54 | — | — | 0.21 | 0.27 | 0.3 | 5.1 | 9.6 | 14 | 0.66 | 0.69 | 0.72 |
| Astrocyte | Mouse/Rat | n.a | n.a | | | n.a | | | n.a | | | n.a | | | n.a | | | n.a | | | n.a | | | n.a | | | n.a | | | n.a | | | n.a | | | n.a | | | n.a | | | n.a | | | n.a | | | n.a | | |
| | Monkey | n.a | n.a | | | n.a | | | n.a | | | n.a | | | n.a | | | n.a | | | n.a | | | n.a | | | n.a | | | n.a | | | n.a | | | n.a | | | n.a | | | n.a | | | n.a | | | n.a | | |
| | Human | 58 (58) | 85 | 100 | 130 | 5 | 7 | 9 | 6.7 | 7.5 | 9 | 2.8 | 3 | 3.1 | 0.61 | 0.74 | 0.90 | 3.4 | 4.1 | 5.0 | 5.0 | — | 5 | 4 | 7 | 13 | 3.9 | 4.8 | — | 22 | 28 | 32 | 0.57 | 0.82 | 1.3 | — | 1.20 | — | 0.64 | — | — | 0.15 | 0.16 | 0.21 | 21 | 27 | 34 | 0.77 | 0.81 | 0.83 |
| Oligodendrocyte | Mouse/Rat | n.a | n.a | | | n.a | | | n.a | | | n.a | | | n.a | | | n.a | | | n.a | | | n.a | | | n.a | | | n.a | | | n.a | | | n.a | | | n.a | | | n.a | | | n.a | | | n.a | | |
| | Monkey | n.a | n.a | | | n.a | | | n.a | | | n.a | | | n.a | | | n.a | | | n.a | | | n.a | | | n.a | | | n.a | | | n.a | | | n.a | | | n.a | | | n.a | | | n.a | | | n.a | | |
| Pyramidal | Mouse/Rat | 3005 (3000) | 170 | 290 | 490 | 4 | 6 | 7 | 4.7 | 6.9 | 7 | 0.34 | 0.34 | 0.86 | 0.35 | 0.45 | 0.7 | 4.4 | 6.7 | 8.8 | 8.8 | — | 11 | 1 | 3 | 10 | 3.2 | 4.6 | — | 37 | 52 | 72 | 0.48 | 0.84 | 1.3 | — | 0.63 | — | 0.99 | — | — | 0.17 | 0.23 | 0.29 | 14 | 26 | 41 | 0.76 | 0.8 | 0.84 |
| | Monkey | 510 (508) | 250 | 370 | 630 | 6 | 7 | 9 | 3.8 | 4.4 | 9 | 2.5 | 2.9 | 3.3 | 0.37 | 0.43 | 0.5 | 6.1 | 7.2 | 8.3 | 8.3 | — | 8.5 | 0.2 | 3.2 | 8.1 | 4.5 | 6.3 | — | 53 | 62 | 72 | 0.44 | 0.69 | 1.2 | — | 0.74 | — | 0.79 | — | — | 0.18 | 0.22 | 0.29 | 22 | 39 | 54 | 0.75 | 0.8 | 0.83 |
| | Human | 2403 (2400) | 230 | 270 | 310 | 5 | 6 | 7 | 2.9 | 3.1 | 7 | 3 | 3.3 | 3.5 | 0.32 | 0.37 | 0.42 | 7.2 | 8.3 | 9.4 | 9.4 | — | 9.7 | 4.6 | 6.9 | 9.6 | 4.1 | 5.2 | — | 64 | 74 | 86 | 0.61 | 0.69 | 1.2 | — | 0.58 | — | 0.88 | — | — | 0.24 | 0.27 | 0.3 | 49 | 63 | 79 | 0.72 | 0.75 | 0.78 |
| Granule | Mouse/Rat | 930 (928) | 100 | 160 | 230 | 1 | 1 | 1 | 2.5 | 3.1 | 1 | 1.6 | 2 | 2.7 | 0.64 | 0.73 | 0.85 | 3.6 | 4.2 | 4.8 | 4.8 | — | 5.1 | 1.0 | 1.8 | 3.8 | 3.2 | 4.1 | — | 27 | 45 | 78 | 0.63 | 0.93 | 1.3 | — | 0.45 | — | 0.78 | — | — | 0.12 | 0.16 | 0.21 | 23 | 35 | 56 | 0.8 | 0.84 | 0.87 |
| | Monkey | n.a | n.a | | | n.a | | | n.a | | | n.a | | | n.a | | | n.a | | | n.a | | | n.a | | | n.a | | | n.a | | | n.a | | | n.a | | | n.a | | | n.a | | | n.a | | | n.a | | |
| | Human | 82 (82) | 460 | 530 | 580 | 2 | 3 | 4 | 2.7 | 3.1 | 4 | 3.6 | 2.6 | 3.6 | 0.41 | 0.77 | 1.0 | 3.1 | 4.0 | 7.4 | 7.4 | — | 7.9 | 4 | 8 | 15 | 3.6 | 4.2 | — | 110 | 130 | 150 | 0.53 | 0.64 | 0.75 | — | 0.49 | — | 0.76 | — | — | 0.18 | 0.20 | 0.24 | 15 | 18 | 23 | 0.83 | 0.85 | 0.87 |
| Purkinje | Mouse/Rat | 140 (140) | 130 | 160 | 190 | 1 | 2 | 2 | 9.3 | 10 | 2 | 2.2 | 2.5 | 2.7 | 0.36 | 0.41 | 0.46 | 6.6 | 7.4 | 8.3 | 8.3 | — | 8.3 | 2 | 26 | 33 | 3.4 | 4.3 | — | 8.9 | 9.9 | 11 | 0.76 | 0.99 | 1.4 | — | 0.22 | — | 0.7 | — | — | 0.25 | 0.28 | 0.31 | 17 | 25 | 32 | 0.73 | 0.78 | 0.81 |
| | Monkey | n.a | n.a | | | n.a | | | n.a | | | n.a | | | n.a | | | n.a | | | n.a | | | n.a | | | n.a | | | n.a | | | n.a | | | n.a | | | n.a | | | n.a | | | n.a | | | n.a | | |
| | Human | n.a | n.a | | | n.a | | | n.a | | | n.a | | | n.a | | | n.a | | | n.a | | | n.a | | | n.a | | | n.a | | | n.a | | | n.a | | | n.a | | | n.a | | | n.a | | | n.a | | |
| Basket | Mouse/Rat | 441 (440) | 190 | 240 | 300 | 4 | 5 | 7 | 2.4 | 3.2 | 7 | 1.5 | 2.0 | 2.7 | 0.39 | 0.51 | 0.66 | 4.6 | 5.9 | 7.8 | 7.8 | — | 9.1 | 1 | 3 | 38 | 2.7 | 4.0 | — | 50 | 64 | 81 | 0.49 | 0.61 | 0.94 | — | 0.77 | — | 0.93 | — | — | 0.22 | 0.27 | 0.33 | 11 | 23 | 45 | 0.73 | 0.77 | 0.81 |
| | Monkey | n.a | n.a | | | n.a | | | n.a | | | n.a | | | n.a | | | n.a | | | n.a | | | n.a | | | n.a | | | n.a | | | n.a | | | n.a | | | n.a | | | n.a | | | n.a | | | n.a | | |
| | Human | 11 (11) | 220 | 270 | 380 | 4 | 5 | 6 | 2.3 | 3.3 | 6 | 3 | 3 | 3.2 | 0.56 | 0.58 | 0.67 | 4.6 | 5.2 | 5.4 | 5.4 | — | 5.6 | 7 | 8 | 45 | 3.4 | 4 | — | 51 | 62 | 110 | 0.92 | 0.95 | 1.5 | — | 0.49 | — | 0.67 | — | — | 0.21 | 0.23 | 0.27 | 14 | 16 | 22 | 0.77 | 0.79 | 0.82 |
| Gabaergic | Mouse/Rat | 180 (179) | 180 | 250 | 440 | 4 | 6 | 8 | 2.9 | 3.7 | 8 | 1.8 | 2.4 | 3 | 0.38 | 0.47 | 0.53 | 5.7 | 6.5 | 8.1 | 8.1 | — | 8.3 | 1 | 1.5 | 4.8 | 4.3 | 6.1 | — | 35 | 46 | 65 | 0.34 | 0.67 | 1.2 | — | 0.80 | — | 0.67 | — | — | 0.22 | 0.29 | 0.33 | 9.8 | 15 | 50 | 0.73 | 0.78 | 0.81 |
| | Monkey | 4 (4) | 120 | 250 | 370 | 2 | 3 | 10 | 2.2 | 2.7 | 10 | 2.6 | 3.1 | 3.6 | 0.55 | 0.62 | 0.68 | 4.7 | 5.1 | 5.8 | 5.8 | — | 5.5 | 0.27 | 0.27 | 0.9 | 4.9 | 5.5 | — | 35 | 67 | 120 | 0.72 | 0.76 | 0.93 | — | 0.51 | — | 0.44 | — | — | 0.3 | 0.34 | 0.37 | 2.1 | 2.3 | 2.5 | 0.7 | 0.72 | 0.76 |
| | Human | 6 (6) | 140 | 190 | 220 | 3 | 5 | 7 | 2.2 | 2.8 | 5 | 3.5 | 4 | 4.5 | 0.67 | 0.69 | 0.72 | 4.2 | 4.5 | 4.6 | 4.6 | — | 4.5 | 0.38 | 0.75 | 7.5 | 7.5 | 7.8 | — | 63 | 65 | 83 | 0.48 | 0.5 | 0.67 | — | 0.57 | — | 0.38 | — | — | 0.28 | 0.30 | 0.33 | 13 | 17 | 27 | 0.73 | 0.74 | 0.74 |
| Glutamatergic | Mouse/Rat | 106 (106) | 150 | 250 | 360 | 3 | 5 | 7 | 2.2 | 2.8 | 7 | 1.2 | 2.7 | 2.3 | 0.34 | 0.42 | 0.50 | 6.1 | 7.3 | 9.0 | 9.0 | — | 9.7 | 1 | 1.8 | 7.2 | 3.9 | 5.1 | — | 42 | 54 | 72 | 0.21 | 0.63 | 0.80 | — | 0.73 | — | 0.98 | — | — | 0.21 | 0.27 | 0.34 | 20 | 27 | 56 | 0.71 | 0.81 | 0.84 |
| | Monkey | n.a | n.a | | | n.a | | | n.a | | | n.a | | | n.a | | | n.a | | | n.a | | | n.a | | | n.a | | | n.a | | | n.a | | | n.a | | | n.a | | | n.a | | | n.a | | | n.a | | |
| | Human | n.a | n.a | | | n.a | | | n.a | | | n.a | | | n.a | | | n.a | | | n.a | | | n.a | | | n.a | | | n.a | | | n.a | | | n.a | | | n.a | | | n.a | | | n.a | | | n.a | | |

N is the number of cellular structures investigated with complete information about the neurite structure; same information for soma in brackets; given the known limitations of the approach used, the number of the corresponding feature may be slightly (on average <20%) under- or over-estimated, respectively, given the known limitations of the approach used. The reported values are mean ± s.d. over the corresponding sample. The '≥' and '≤' are used when the estimated value of the corresponding feature may be slightly under- or over-estimated, respectively. n.a. not available.

**Table 3 | Reference values for all the morphological features of neuronal and glial cells**

| Morphological Feature | Value Range | Value Mean | | |
|---|---|---|---|---|
| | | **All Cell Types** | **Only Glia** | **Only Neuron** |
| $R_{domain}$ | $\geq 8 - 750$ | 240 | 60 | 300 |
| $N_{proj}$ | $\geq 1 - 13$ | 7 | 7 | 7 |
| $BO$ | $1 - 17$ | 5 | 7 | 4 |
| $S/V_{domain}$ | $0.2 - 18$ | 3 | 3 | 3 |
| $R_{soma}$ | $0.5 - 16$ | 5 | 3 | 6 |
| $R_{MRsoma}$ | $3 - 11$ | 7 | 4 | 8 |
| $\eta_{s}oma$ | $0.5 - 100$ | 17 | 25 | 13 |
| $S/V_{soma}$ | $0.05 - 30$ | 0.7 | 1 | 0.6 |
| $S/V_{branch}$ | $4 - 19$ | 7 | 8 | 6 |
| $L_{branch}$ | $3 - 190$ | 54 | 13 | 68 |
| $R_{branch}$ | $\leq 0.04 - 1.6$ | 0.6 | 0.5 | 0.6 |
| $R_{MRbranch}$ | $0.20 - 1.05$ | 0.7 | 0.6 | 0.7 |
| $CV_{branch}$ | $0.05 - 4.6$ | 0.6 | 0.8 | 0.6 |
| $\mu OD_{branch}$ | $0.05 - 0.60$ | 0.25 | 0.23 | 0.26 |
| $R_c$ | $\geq 1 - 640$ | 29 | 16 | 32 |
| $\tau_{branch}$ | $0.43 - 0.95$ | 0.77 | 0.75 | 0.78 |

The ranges and mean values obtained from the whole dataset investigated are reported, together with mean values for only neurons and glia. The '≥' and '≤' are used when the estimated value of the corresponding feature may be slightly (on average < 20%) under- or over-estimated, respectively, given the known limitations of the approach used.

**Table 4 | Summary statistics of shape descriptors for each cell type and species**

| Cell Type | Species | $\kappa$ | OD | $(\tau_1, \tau_2, \tau_3)$ | FA | adjFA |
|---|---|---|---|---|---|---|
| | Mouse/Rat | $1.4 \pm 2.2$ | $0.51 \pm 0.22$ | $(0.13, 0.31, 0.56)$ | $0.55 \pm 0.22$ | $0.33 \pm 0.23$ |
| Microglia | Monkey | $1.1 \pm 0.6$ | $0.50 \pm 0.19$ | $(0.01, 0.35, 0.64)$ | $0.76 \pm 0.05$ | $0.36 \pm 0.17$ |
| | Human | n.a. | n.a. | n.a. | n.a. | n.a. |
| | Mouse/Rat | $1.6 \pm 1.3$ | $0.43 \pm 0.22$ | $(0.01, 0.29, 0.61)$ | $0.63 \pm 0.20$ | $0.41 \pm 0.22$ |
| Astrocyte | Monkey | n.a. | n.a. | n.a. | n.a. | n.a. |
| | Human | n.a. | n.a. | n.a. | n.a. | n.a. |
| | Mouse/Rat | $6.0 \pm 4.6$ | $0.19 \pm 0.19$ | $(0.03, 0.15, 0.82)$ | $0.87 \pm 0.15$ | $0.76 \pm 0.23$ |
| Oligodendrocyte | Monkey | n.a. | n.a. | n.a. | n.a. | n.a. |
| | Human | n.a. | n.a. | n.a. | n.a. | n.a. |
| | Mouse/Rat | $1.6 \pm 0.12$ | $0.44 \pm 0.20$ | $(0.09, 0.25, 0.66)$ | $0.70 \pm 0.16$ | $0.52 \pm 0.23$ |
| Pyramidal | Monkey | $0.9 \pm 0.6$ | $0.57 \pm 0.19$ | $(0.06, 0.29, 0.65)$ | $0.72 \pm 0.10$ | $0.45 \pm 0.23$ |
| | Human | $1.1 \pm 0.8$ | $0.54 \pm 0.20$ | $(0.14, 0.27, 0.59)$ | $0.60 \pm 0.14$ | $0.43 \pm 0.21$ |
| | Mouse/Rat | $4.4 \pm 2.8$ | $0.18 \pm 0.11$ | $(0.04, 0.15, 0.81)$ | $0.87 \pm 0.09$ | $0.77 \pm 0.16$ |
| Granule | Monkey | n.a. | n.a. | n.a. | n.a. | n.a. |
| | Human | $3.7 \pm 1.5$ | $0.19 \pm 0.08$ | $(0.05, 0.16, 0.78)$ | $0.85 \pm 0.07$ | $0.75 \pm 0.13$ |
| | Mouse/Rat | $1.5 \pm 0.7$ | $0.42 \pm 0.17$ | $(0.01, 0.28, 0.71)$ | $0.80 \pm 0.08$ | $0.52 \pm 0.19$ |
| Purkinje | Monkey | n.a. | n.a. | n.a. | n.a. | n.a. |
| | Human | n.a. | n.a. | n.a. | n.a. | n.a. |
| | Mouse/Rat | $1.3 \pm 0.8$ | $0.49 \pm 0.19$ | $(0.11, 0.29, 0.60)$ | $0.62 \pm 0.16$ | $0.40 \pm 0.20$ |
| Basket | Monkey | n.a. | n.a. | n.a. | n.a. | n.a. |
| | Human | $1.2 \pm 0.7$ | $0.49 \pm 0.20$ | $(0.12, 0.34, 0.54)$ | $0.56 \pm 0.13$ | $0.28 \pm 0.14$ |
| | Mouse/Rat | $1.3 \pm 0.9$ | $0.49 \pm 0.20$ | $(0.07, 0.30, 0.63)$ | $0.69 \pm 0.13$ | $0.43 \pm 0.21$ |
| Gabaergic | Monkey | $3.2 \pm 1.6$ | $0.24 \pm 0.15$ | $(0.07, 0.20, 0.73)$ | $0.78 \pm 0.13$ | $0.64 \pm 0.24$ |
| | Human | $2.5 \pm 1.4$ | $0.29 \pm 0.14$ | $(0.3, 0.25, 0.62)$ | $0.64 \pm 0.15$ | $0.50 \pm 0.15$ |
| | Mouse/Rat | $1.1 \pm 0.9$ | $0.55 \pm 0.21$ | $(0.08, 0.33, 0.59)$ | $0.65 \pm 0.12$ | $0.33 \pm 0.21$ |
| Glutamatergic | Monkey | n.a. | n.a. | n.a. | n.a. | n.a. |
| | Human | n.a. | n.a. | n.a. | n.a. | n.a. |

*n.a.* not available.

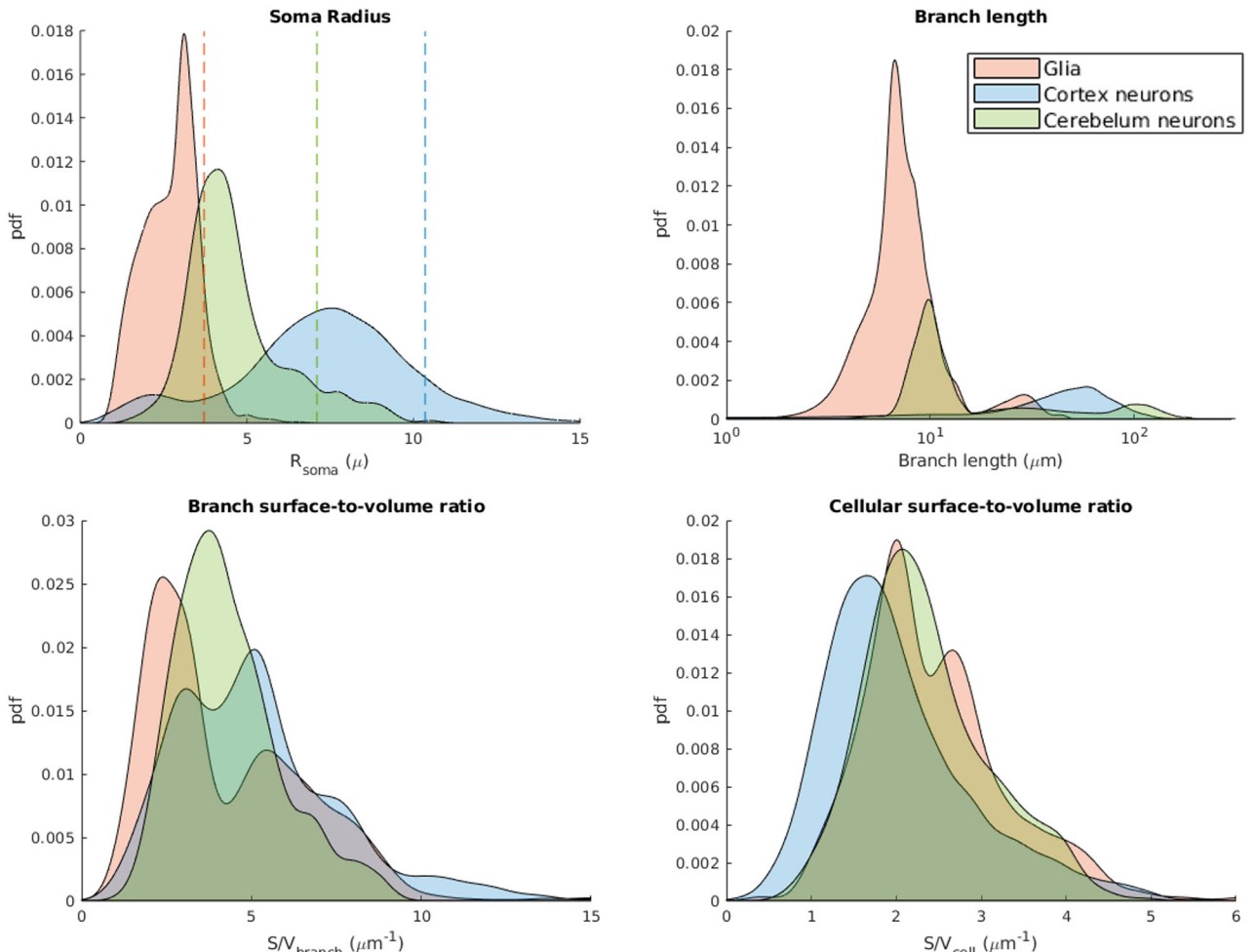

**Fig. 6 | Distributions of soma radius, branch length, branch surface-to-volume ratio, and cellular surface-to-volume ratio (grouped by general cell types).** Dashed lines in soma radius distribution plot indicate the effective MR radius for each general cell type (Glia $R_{MRsoma}$ = 3.7 $\mu m$, Cortex neurons $R_{MRsoma}$ = 10.4 $\mu m$, and Cerebellum neuron $R_{MRsoma}$ = 7.1 $\mu m$.

## Discussion

### A quantitative view of gray matter microstructure

**GM intra-cellular space**. There is currently a lack of in-depth morphological analysis of brain-cell structures of relevance for modelling water diffusion in the GM intra-cellular space. In this work we propose a first analysis with the aim to fill this gap. We estimated a comprehensive set of morphological features useful to GM microstructure modelling from reconstructions of microscopy data from three species. We estimated that neural soma size ranges from 2 to 30 $\mu m$ in radius with an average of 5 $\mu m$ and surface-to-volume ratio S/V 0.7 $\mu m^{-1}$. Neurons have on average soma twice as big as glial cells, similar number of projections radiating from the soma but less projection coverage of soma surface (suggesting slower exchange of diffusing molecules between soma and dendrites). The radius of cellular projections ranges from 0.05 to 1.5 $\mu m$ with average value 0.6 $\mu m$ and S/V 7 $\mu m^{-1}$, similar between neurons and glia. Cellular projections' microscopic orientation dispersion, as defined in ref. 48, is 0.05–0.60, with average value 0.25, similar between neurons and glia; curvature radius is 1–640 $\mu m$, with average value of 29 $\mu m$. On average, neuronal projections have a curvature radius 2 times larger than glial projections. The branching order of neural cell is 1–17, with average value of 7. Glial cells have branching order 1.3 times larger than neurons. The tortuosity of the branch of neural cell projections is 0.43–0.85, with average value of 0.77, similar between neurons and glial cells. The projections of neural cells extend to distances of 8-750 $\mu m$, with neurons on average 300 $\mu m$ and glial cells on average 60 $\mu m$. For completeness, we

also report on the relevant features of neural cell membrane permeability to water from the literature.

**Neural cell membrane permeability to water**. Several works suggest water exchange between unmyelinated neurites and ECS and/or soma occurring on time scales comparable to typical dMRI clinical acquisitions, i.e., 10–100 ms. Although there is not a consensus yet, some works report exchange times $t_{ex}$ for ex vivo mouse brain 5–10 ms[34,60,61], suggesting membrane permeability 125 $\mu m$/s (like red blood cells), others report $t_{ex}$ for in vivo mouse brain of 20–40 ms[33,62,63], suggesting membrane permeability 2–35 $\mu m$/s. The broad range of permeability values present in the literature result in a large range of residence and exchange times. Consequently, to accurately quantify the impact of permeability on the dMRI signal, an accurate measure of the membrane permeability of the tissue being investigated is needed. Here, we provide reference values of intra-branch, intra-cellular residence times and corresponding exchange times for any possible permeability value within the range 2–35 $\mu m$/s, observed in vivo (Fig. 7). Using a representative value of low (2 $\mu m$/s) and high (20 $\mu m$/s) permeability, we also estimated the distribution of intra-cellular and intra-branch residence times, which span from a few milliseconds to hundreds of milliseconds (Fig. 7).

One limitation of our analysis is that we did not include spines, boutons and glial leaflets, that can occupy up to 20% of the GM volume[64]. Dendritic spines influence intracellular diffusion by increasing the surface-to-volume ratio and introducing structural complexity and compartmentalisation

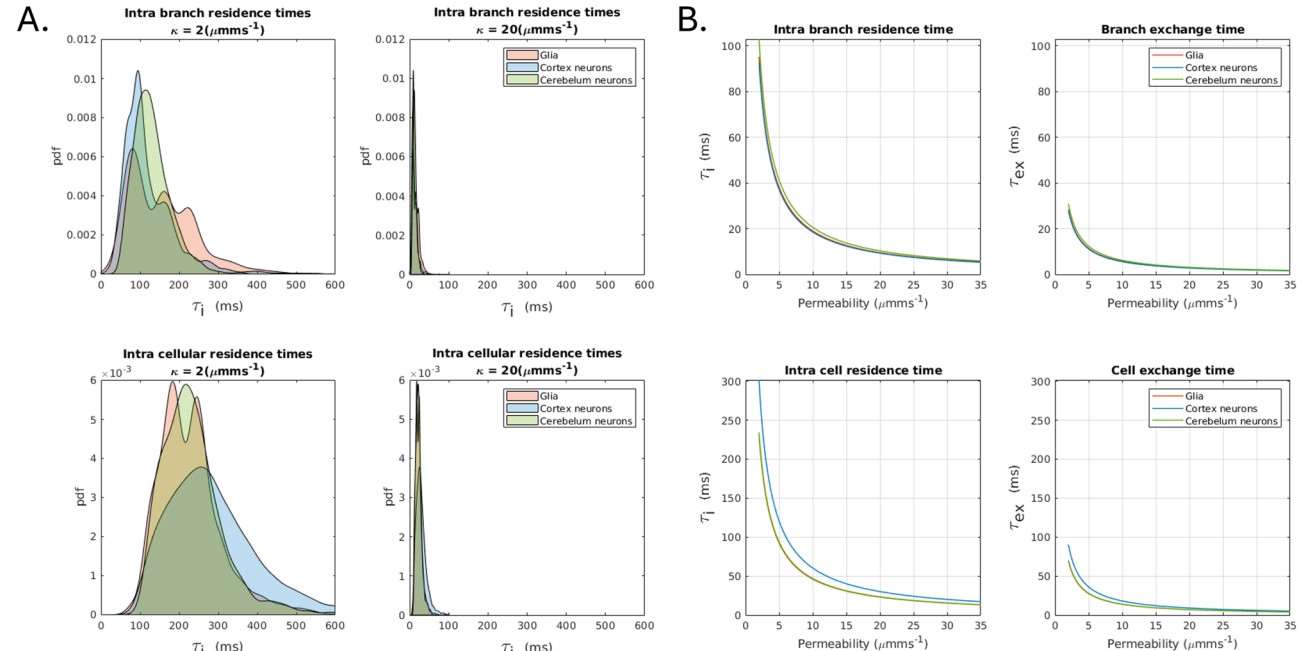

**Fig. 7 | A visualisation of residence times distributions and plots of exchange times against permeability. A** Distributions of intra-cellular and intra-branch residence times (grouped by general cell types) for two membrane permeabilities (low and high). Residence time distributions were derived from surface-to-volume ratio distributions using permeability values of 2 and 20 $\mu m/s$ to represent low and

high permeability, respectively. **B** Mean residence and exchange times as a function of membrane permeability. The mean values were calculated from the distributions of exchange times obtained for different surface-to-volume ratios and membrane permeabilities. See Section 2.2 for details.

within dendrites[65–67]. Using simple modeling (see Supplementary Fig. 5), we estimate that a spine density of 1 spine/$\mu m$ increases the branch surface-to-volume ratio by $\approx 20-30\%$ in rodent neurons, leading to a $\approx 15 - 20\%$ reduction in residence time. At a density of 2 spines/$\mu m$, the surface-to-volume ratio increases by $\approx 50\%$, with a corresponding $\approx 35\%$ reduction in residence time.

Beyond these surface effects, spines act as distinct compartments: their narrow necks serve as diffusion barriers, restricting molecular exchange between the spine head and dendritic shaft and thereby altering diffusive dynamics[66,67]. Consequently, spine morphology represent an important morphological feature to be considered in models of molecular diffusion to accurately reflect in vivo conditions, but its in depth investigation is beyond the scope of this work.

In general, exchange cannot be completely ignored, nor does exchange fully average out the restriction effects of cell membranes, supporting including both neurite exchange and a soma compartment in a parsimonious biophysical model of diffusion in GM, as proposed by Olesen et al.[34] with the SANDIX model. There are however some caveats to bear in mind when interpreting the model estimates: (a) the soma apparent MR radius is unavoidably an overestimation of the true soma radius (Table 2); (b) the compartmental signal fractions reflect, apart from volume, also unknown T2 and T1 weighting, which might differ among the compartments; (c) fast exchange across the neurite membrane's could be problematic for models based on the Kärger's model because it could violate the model's assumptions (i.e., barrier-limited diffusion: $t_{ex} \gg d^2/D$, where $d$ is the neurite diameter and $D$ the molecular diffusivity; and $\delta \ll t_{ex}$, where $\delta$ is the gradient pulse duration of a single diffusion encoding dMRI measurement) and lead to biased estimates; (d) molecular diffusion in structural features such as dendritic spines and glial leaflets can lead to diffusion-mediated exchange mechanisms that occur on the same time scales of permeative exchange, making the interpretation of exchange measurements through MRI solely due to permeative exchange fundamentally wrong[65–67]; (e) using single diffusion encoding acquisitions it is impossible to disentangle restriction and exchange; more refined acquisitions could allow the separation and more accurate quantification of these two effects[61,66–68]. The estimates of the

model parameters should therefore only be taken as an indication of the true tissue features and validation against realistic numerical simulations (e.g., using the high resolution 3D exemplar meshes we provide for each cell type) and/or alternative measurements in controlled phantoms (e.g., biomimetic tissues and brain organoids) and/or post-mortem samples (e.g., optical, confocal or electron microscopy) remains essential.

### General implications for biophysical modelling

Here we provide a few illustrative examples demonstrating how our results can inform biophysical modelling. We discuss both general considerations and a representative in vivo acquisition case using parameters typical for clinical scanners.

Assuming an intra-cellular diffusivity D = 2 $\mu m^2/ms$ representative of water and 0.40 $\mu m^2/ms$ representative of intracellular brain metabolites, and considering a typical single diffusion encoding acquisition with gradient pulse duration $\delta < 30\ ms$, gradient pulse separation $\Delta < 70\ ms$ and diffusion time $t_d < 60\ ms$, we can infer from from our results:

- **Impact of soma restriction**. The impact of soma restriction is measurable when (from ref. [69]) $5Dt_d \geq R^2_{soma}$, that is for $t_d \geq 2.5\ ms$ for water and $t_d \geq 12.5\ ms$ for metabolites, given a soma radius $R_{soma} \approx 5\ \mu m$. These reference values become slightly longer if we consider the effective MR radius $R_{MRsoma} \approx 7\ \mu m$: $t_d \geq 5\ ms$ for water and $t_d \geq 24.5\ ms$ for metaboilites. Given the exemplar case of the in vivo acquisition, this suggests that soma restriction can have a measurable impact for both water and metabolites. This is supported by previous findings which demonstrate the impact of soma contribution on the measured dMRI signal in GM[32,34]

- **Impact of diffusion-mediated exchange between soma and projections**. Considering the total area of the soma surface covered by the openings towards cellular projections $A_{proj} \approx N_{proj}\pi R^2_{branch}$ and the soma volume $V_{soma} \approx \frac{4}{3}\pi R^3_{soma}$, the expected residence time within the soma is in first approximation $\tau^{soma}_i \approx \frac{V_{soma}}{4\sqrt{\frac{A_{proj}}{\pi}}D} = \frac{\pi R^3_{soma}}{3R_{branch}\sqrt{N_{proj}}D}$. Assuming that the soma to projection volume ratio is approximately

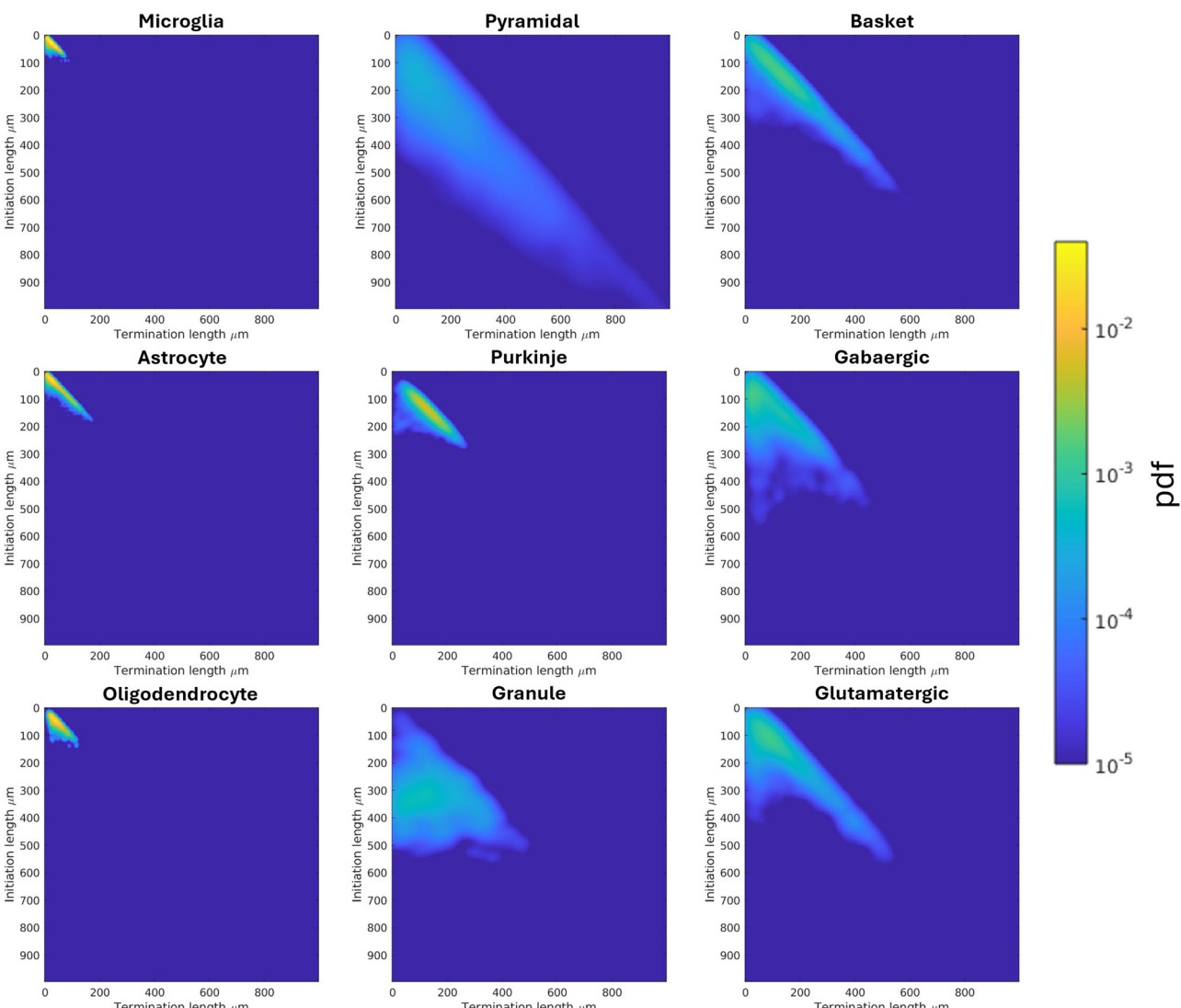

**Fig. 8 | Persistence maps for each cell types and all species together.** The persistence map shows at what length scales a given topological feature, here the path length of connected branches, persists. It is computed by tracking the initiation points and termination points, with respect to path length from the soma, of the connected branches at different length scales.

1:4 (from our data we estimated 1:4.1 for glia, 1:4.7 for cortical neurons and 1:4.4 for cerebellar neurons), the impact of diffusion-mediated exchange between soma and projection is measurable if $t_d \geq (\frac{1}{\tau_i^{soma}} + \frac{1}{\tau_i^{branch}})^{-1} = (\frac{1}{\tau_i^{soma}} + \frac{1}{3}\frac{1}{\tau_i^{soma}})^{-1}$, hence for $t_d \geq 31\ ms$ for water and $t_d \geq 155\ ms$ for metabolites. These estimates become longer if we consider the effective MR radii $R_{MRsoma}$ and $R_{MRbranch}$: $t_d \geq 73\ ms$ for water and $t_d \geq 364\ ms$ for metabolites. Given the exemplar case of the in vivo acquisition, this suggests that the impact of diffusion-mediated exchange between soma and projections is likely negligible or minimal for both water and metabolites.

- **Impact of cellular domain restriction:** given the size of the cellular domain, it will only significantly restrict molecular diffusion when $5Dt_d \geq R_{domain}^2$, that is for $t_d \geq 360\ ms$ for water and $t_d \geq 1800\ ms$ for metabolites, given $R_{domain} \geq 60\ \mu m$, which is far longer than typically used diffusion times $t_d$. Given the exemplar case of the in vivo acquisition, this suggests that the impact of cellular domain restriction is generally negligible for both water and metabolites.

- **Impact of projections curvedness.** The impact of curvedness can only be significant when (from ref. 70, $2D\Delta, 2D\delta \geq (<R_c>_s)^2$, that is for

$\Delta$, $\delta \geq 225\ ms$ for water and $\Delta$, $\delta \geq 1125\ ms$ for metabolites, given $<R_c<_s \approx 30\mu m$, which is far longer than the values typically used. Given the exemplar case of the in vivo acquisition, this suggests that the impact of projections curvedness is generally negligible for both water and metabolites. This is further supported by ref.[34].

- **Impact of projection undulation:** the values of $<\mu OD_{branch}<_s$ estimated from the real cellular data match those simulated in ref. 48, from which it can be concluded that undulation can have a measurable impact for both water and metabolites and can bias the estimation of projection radius in GM. However, the average branch radius $<R_{branch}<_s \approx 0.6\ \mu m$ is far below the resolution limit of conventional water dMRI techniques[71].

- **Impact of branching.** Given the branch length $L_{branch} \approx 54\ \mu m$, the exchange between branches is negligible for $t_d < <L_{branch}^2/(2D) \approx 750\ ms$ for water and $t_d << 3750\ ms$ for metabolites, significantly longer than typical $t_d$ used. Given the exemplar case of the in vivo acquisition, this suggests that the impact of branching is generally negligible for both water and metabolites. Further supported by refs. 45,34.

- **Impact of water permeative exchange between cellular projections and extra-cellular space.** The average intra-branch residence times

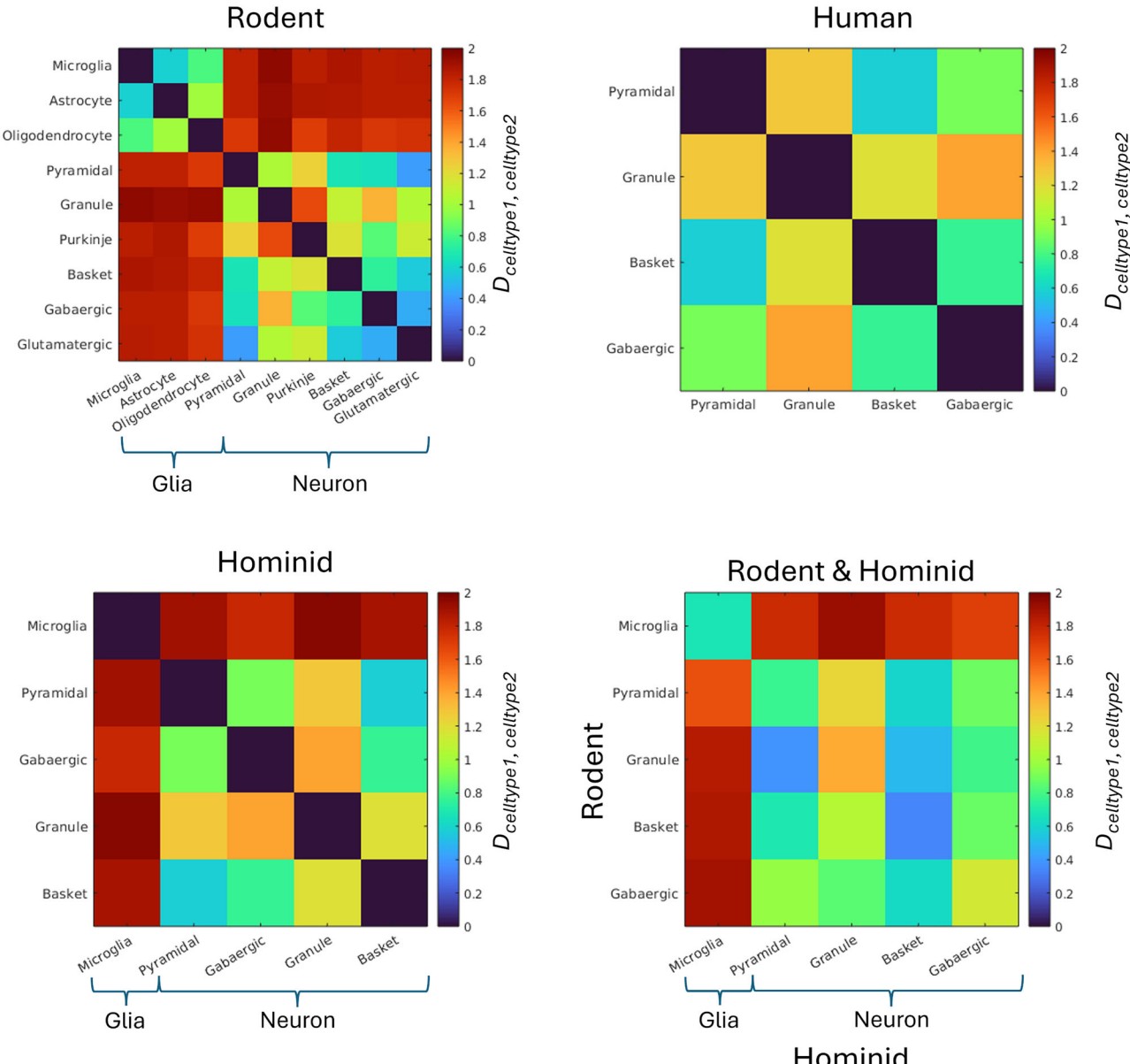

**Fig. 9 | Topological comparison between cell types.** Matrices show the pair wise topological distance between cell types for rodent, human, hominid inter species comparison, and a comparison between rodent and hominid cell types.

given the estimated $S/V_{branch}$ range from $\approx 10\,ms$ to $\approx 98\,ms$ for neurons and from $\approx 9\,ms$ to $\approx 95ms$ for glial cells (these are purely based on projections morphology and exemplar membrane permeability $2$–$20\,\mu m/s$; we do not account for active transport nor water channels). The corresponding exchange times (considering 30% in volume occupied by extra-cellular space) range from $\approx 3ms$ to $\approx 30ms$ for neurons and from $\approx 3ms$ to $\approx 29ms$ for glial cells. The permeative exchange is thus negligible only for $t_d<30ms$ for both neuronal and glial projections; which is not satisfied in conventional water dMRI applications. Given the exemplar case of the in vivo acquisition, this suggests that the impact of water permeative exchange between cellular projections and extra-cellular space is measurable. Further supported by refs. 33,34,63.

- **Impact of projections orientation dispersion**. Cells with highly oriented and polarized projections, such as Purkinje and granule cells have high FA (<0.50) and low orientation dispersion (<0.25); while the projections of most of the glial cells and other neuronal cells have FA (<0.50) and high orientation dispersion (<0.25). Given the exemplar

case of the in vivo acquisition, this suggests that the impact of projections orientation dispersion is measurable. This supports growing evidence that FA, e.g., from DTI, and orientation dispersion estimates, e.g., from NODDI, can discriminate different cytoarchitectural domains (or layers) in hippocampus, cortex and cerebellum[72–76].

- **Glia topology is significantly different from neurons topology in rodents**. The overall cellular topology can be conceptualized as a network of interconnected compartments, comprising the soma and its cellular projections. This large-scale organization is as critical for intracellular diffusion as the finer structural details. In neural cells, projections form a branched tree-like architecture, where each branching point functions as a diffusion junction. As a result, molecular diffusion - and consequently the dMRI signa - is strongly constrained by this branching topology. Biophysical models of the intracellular dMRI signal must therefore explicitly account for these topological features. Results in Fig. 9 show that glial cells in rodents have a very different topology from neuronal cells, suggesting that it may be possible (with appropriate modelling) to disentangle glial and

neuronal contributions to the measured dMRI signal. This is further supported by ref. 77.

- **Microglia topology is similar across different species**. Results in Fig. 9 suggest that a single biophysical model of diffusion in microglia would likely hold across species. In contrast, all the other cell-types likely need a dedicated model that accounts for the species-specific topological differences.

## From Ex-vivo histology to in-vivo dMRI

Although the translation of our ex-vivo histological results to in-vivo dMRI is not straightforward, some considerations can be made to aid the interpretation of in-vivo dMRI measurements and design numerical simulations mirroring more closely the expected in-vivo condition.

- **Temperature / intrinsic diffusivity:** as for any synthetic signal, be it through an analytical expression or through simulation within a given geometry, the diffusivity is a variable parameter. In first approximation, we can use the Stokes-Einstein[78] relationship to link the diffusivity to the temperature: $D = k_B T / (3\pi\eta\sigma)$, where D is the diffusion constant, $k_B$ is the Boltzmann constant, T is absolute temperature, $\eta$ is the fluid's shear viscosity, and $\sigma$ is the particle's diameter. As such the diffusivity can be defined as to reflect the experimental conditions, whether it be ex-vivo or in-vivo. Thus, whilst temperature affects the value of water diffusivity, it does not complicate the translation of the morphological data to in-vivo conditions.

- **Membrane permeability and exchange:** post-mortem changes may alter membrane integrity and therefore water exchange, however, as for water diffusivity, this parameter can be tailored to reflect the appropriate conditions when generating synthetic dMRI signals. Many Monte Carlo based simulators of dMRI signals, such as CAMINO[79], MCDC[80] and MCMR[81] allow for tuneable surface permeability. The remaining challenge will be to determine which permeability value to use, which depends on the specific investigation. Future work combining in-vivo dMRI methods sensitive to exchange with simulated signals will help constrain appropriate permeability parameters.

- **Tissue shrinkage during fixation:** tissue samples undergo many processing steps before imaging which can cause shrinkage. Analysis of the meta data of the reconstructions used here revealed that the reported tissue shrinkage ranged from <1% to 50% with an average of 15%. Whilst no attempt was made to correct for this in our analysis, as most reconstructions do not report tissue shrinkage, structural characteristics or the provided meshes can be scaled appropriately given this information to remove potential bias.

- **Cellular apoptosis or other viability changes:** apoptosis is minimal postmortem, as it is an active process that rapidly ceases after death[82]. Cell membranes are also resistant to decomposition, with no significant changes in cell size or shape even after 22 h postmortem[82]. As such, the analyzed cellular reconstructions likely represent their in-vivo counterparts. However, if accounting for apoptosis or other changes in cellular composition is necessary to match the experimental conditions (e.g., ex-vivo vs in-vivo), this can be addressed by scaling accordingly the relative contribution of the total dMRI signal from each cellular component. The total dMRI signal from a macroscopic imaging voxel is the volume weighted sum of the dMRI signals from each cellular compartment. Therefore, the associated weights can be adjusted to match any known composition.

## Limitations

Whilst neural reconstructions, like the ones analysed here, offer the closest ground truth for cellular morphology they are not without their limitations. During the process of acquiring and preparing tissue samples, distortions of the cellular structure can occur, such as tissue shrinkage, or the truncation of neural projections, implicating the accuracy of the morphological measures. Furthermore, the method used to trace and reconstruct the cellular structure

from microscopy images, whether done manually or automated, can impact the accuracy and detail of morphological characteristics present in the final reconstruction. As such, it is important to acknowledge that the characteristics reported here may deviate from actual in vivo measurements.

Our investigation, while thorough, is not exhaustive (nor could it be). Other features could certainly be measured and tabulated, such as spine density. However, we focus here on those deemed relevant for biophysical modeling in the current literature on microstructure imaging via dMRI. We will release our code openly and freely to allow future work to complement this study with additional features and information as needed. Our focus is on a selected set of real three-dimensional reconstructions, as we aim to characterize cellular morphology in the healthy brain. Future studies can use this code to incorporate more and improved reconstructions of healthy brain tissue as they become available (e.g., through updates to NeuroMorpho or large electron microscopy studies[83,84]). We provide a few illustrative examples to demonstrate how this study can inform biophysical modeling of dMR signals. However, the information obtained from our investigation has broader applications, and we hope the scientific community will find it valuable for advancing our understanding of gray matter microstructure as a whole. Throughout the manuscript, we discussed the limitations of the morphometric approach used (e.g., an expected underestimation of <20% for soma volume and branch radius). Within the constraints of currently available tools, we provide reference values that were previously unavailable. We have taken great care to report error estimates and uncertainties for all measurements to account for these limitations. Future work is needed to improve morphometric algorithms, but this is beyond the scope of the current study.

## Conclusion

This work provides quantitative information on brain cell structures essential to design sensible biophysical models of the dMRI signal in gray matter. Reporting typical values of relevant features of brain cell morphologies, this study represents a valuable guidebook for the microstructure imaging community and provides illustrative examples demonstrating how to inform biophysical modelling.

## Data availability

Neural reconstructions were obtained from Neuromorpho.org, for full list of archives please see Supplementary Material. The meshes described in the text are available at (https://doi.org/10.17035/cardiff.30491159)[85].

## Code availability

The Matlab scripts used for the analysis are available at (https://github.com/Charlie-Aird/Decoding-Grey-Matter).

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

## Acknowledgements

We would like to thank Dr. Kanari for informative discussion regarding the topological analysis. This work, C.A.R. and M.P. are supported by the UKRI Future Leaders Fellowship MR/T020296/2. D.K.J. was supported by a Wellcome Trust Strategic Award (104943/Z/14/Z) and Wellcome Discovery Award (227882/Z/23/Z).

## Author contributions

Conceptualization, M.P., C.A., D.A., and H.Z.; Methodology, M.P. and C.A.R.; Validation, M.P., C.A.R.; Formal analysis, M.P., and C.A.R.; Writing—original draft preparation, M.P., and C.A.R.; Writing—review and editing, M.P., C.A.R., D.A., D.J. and H.Z.; Supervision, M.P.; Funding acquisition, M.P. and D.J.; All authors have read and agreed to the published version of the manuscript.

## Competing interests

The authors declare no competing interests.
