## [Transparent Peer Review file · Communications Biology]

Decoding Gray Matter: large-scale analysis of brain cell morphometry to inform microstructural modeling of diffusion MR signals

Corresponding Author: Mr Charlie Aird-Rossiter

Version 0:

Reviewer comments:

Reviewer #1

(Remarks to the Author)

In this very well-written and structured manuscript, the team concisely presents a large-scale quantitative analysis of gray matter cellular morphometry, detailing their processing and analysis methods applied to 3D cell reconstructions sourced from an online database. They systematically compute and report a comprehensive set of structural, shape, and topological descriptors across various cell types and species. They are addressing a knowledge gap by providing quantitative benchmarks for GM microstructure, the insightful discussion linking these features to the interpretation and limitations of dMRI biophysical models. While the overall quality is high, the manuscript would benefit from a careful proofreading pass, as there are a few minor typographical errors scattered throughout the text.

Minor comments:

- Intro: The paper is organized as following (as follows)
- Intro: subsequent interpretation if (typo, of)
- A few Missing citation (??, probably a Latex issus, maybe double-check)
- Section 2.1: dentritic projections (dendritic)
- Section 2.3: cylinders of length 10m (Missing μ ?)
- section 2.4: with respect to the path distance fro the soma (fro - from)
- section 3.3: τ , across majority (of) cell types. (missing of)
- section 3.6: Additionally, the comparatively low (comparatively)
- section 4.1: One potential limitation (limitation) of our analysis is that we did not include spines, boutons and glial leaflets (leaflets)
- section 4.1: when analysing and intepreting (interpreting) time-dependent dMRI
- section 4.1: fast exhchange (exchange)
- Section 4.2: which is far longer than the values typically. (missing the word used?)
- Section 4.2: Typo L^2 _baranch

In section 2.3, the paragraph starting with 'To calculate FA we followed' and the next one starting with 'The cellular structure is' seems to be explaining the same thing twice, is this an error?

Reviewer #2

(Remarks to the Author)

The manuscript "Decoding Gray Matter: large-scale analysis of brain cell morphometry to inform microstructural modeling of diffusion MR signals" of Aird-Rossiter et al. summarises the main features of cellular morphometry like soma radius as well as lenght, radius and dispersion of neuronal projects from 3500 3D cellular reconstructions across species and cell types. Through an exhaustive method and careful considerations due to inherent biases due to the specific histology approach processing (neither in the paper or in the website specify which histology technique was used to create the SWC files),

demonstrate differences and similarities between cell types and species; as well as an indication on how to use this information for biophysical modelling in diffusion MRI. This work, in synthesis, could be used as a benchmark or reference when improving or defining new biophysical models for dMRI in grey matter for in vivo and ex vivo human brain.

The manuscript is, however, quite difficult to understand at first glance making necessary to read it two or three times the methods section to understand the results. Specially the topological descriptors section, in which the motivation and use of quantities like the topological distance was still not clear. It is also important to mention that the average and standard deviation of different metrics could be biased given that the distribution (or PDFs) of these metrics are not Gaussian or Normal. This is not only important for coherence (or to use the proper estimates for the correct distributions) but also because they require a different discussion and comparison with the estimated biophysical parameters of dMRI. Take for example the discussion between mean and effective radius between histology and dMRI from Veraart et al. 2020 (<https://pubmed.ncbi.nlm.nih.gov/32048987/>). Lastly, it is important to emphasize that the morphometry description of these cells cannot be compared directly with in vivo human brain. I think it is important to give an insight on how these results (or future references) could benefit in the validation of biophysical modelling for in vivo dMRI.

Before acceptance and publication, I would encourage the authors to tackle the aforementioned points. I have attached a PDF that contains a more detailed review on the manuscript (I hope my handwriting is clear enough - if not, please don't hesitate to contact me).

Reviewer #3

(Remarks to the Author)

The authors present a large-scale quantitative analysis of neural cell morphology across multiple species and cell types to inform gray matter diffusion MRI (dMRI) modeling. This is a topic of interest to the dMRI community, as gray matter (GM) analysis often lags behind white matter (WM) analysis due to the diversity of cell types and the complexity of GM microstructure.

The manuscript is well organized, and the analysis appears thorough. However, several methodological assumptions and presentation choices limit the clarity of how the findings translate to microstructural modeling.

Given the complexity of GM and the challenges involved in validating dMRI models, the standards for establishing ground-truth reference values are necessarily high, and further rigor is expected.

In addition, considering the impact and scope expected of Communications Biology, this work reads more as a technical resource or descriptive dataset than a biological discovery study or conceptual advance.

While the quantitative characterization of cellular features is valuable, the manuscript currently lacks sufficient mechanistic insights, novel biological hypotheses, or modeling validation that would justify publication in a high-impact biology journal. At its current stage, this work may be better suited for a resource-focused venue such as Scientific Data.

Main concerns:

1. Selection Bias and Incompleteness of Reconstructions:

Although quality control is mentioned, there is no quantitative assessment of how representative the selected cells are of true cellular diversity. Many reconstructions lack axons and spines, which could significantly influence diffusion properties and branching patterns. Please quantify what fraction of the original database was excluded for each cell type and discuss how this might bias the reported morphological distributions (e.g., preferential selection of larger somas or simpler dendritic trees).

2. Impact of Missing Features on Biophysical Modeling:

The analysis does not account for important structures such as dendritic spines, boutons, or glial leaflets, which can substantially affect surface-to-volume ratios and exchange dynamics. Although this limitation is acknowledged in the discussion, its implications should be evaluated quantitatively. For example, how would including a typical spine density (~1–2 spines/ μm) alter S/Vbranch estimates? A simple modeling exercise would strengthen the conclusions regarding exchange timescales.

3. Topological Comparisons Lack Statistical Testing:

The paper reports global topological distances between cell types based on persistence diagrams but does not assess the statistical significance of these differences. Is it possible to perform permutation tests or bootstrapping to determine whether the observed distances (e.g., neuron vs. glia) are robust and meaningful beyond random variation?

4. Relevance of Residence Time Estimates for In Vivo dMRI:

In Figure 6, intracellular residence time distributions are computed assuming a fixed permeability of 20 $\mu\text{m}/\text{s}$. However, recent studies suggest a broader range of permeability values for axons (e.g., https://doi.org/10.1007/978-3-031-43993-3_13). Do soma bodies and dendritic branches show similarly variable permeability, or is the fixed value assumption justified?

Given previous work shows the residence time depends on the permeability and radius

(<http://dx.doi.org/10.1016/j.neuroimage.2017.02.013>),

it would strengthen the analysis to show how sensitive the residence time distributions are to different permeability values (e.g., plotting low and high permeability cases). This would help assess the robustness of the conclusion that exchange cannot be ignored.

5. Resource Sharing:

While the descriptors are well defined, they are largely derived from existing modeling frameworks (e.g., DTI, NODDI), which themselves have known limitations. Will the authors share the 3D meshes of the analyzed cells? Making these data publicly available would be a valuable resource for the community, enabling studies of dMRI signal formation (e.g., random walk simulations, Bloch–Torrey equation modeling) and the development of more accurate microstructural models.

Minor comments:

Typographical Errors:

Minor typos appear throughout, including "intepreting" → "interpreting", "barnch" → "branch", incomplete sentence "To calculate FA we followed the method outlined in.", and missing figure references such as "The distributions for all the morphological features are reported in supplementary Fig.??." Please proofread carefully.

Clarify Figure 2 Annotation:

The figure illustrating structural descriptors would benefit from directly labeling the soma and projections on the 3D rendering for improved clarity.

Persistence Diagram Methods:

It is mentioned that kernel density estimation was applied to persistence images. Please specify the kernel bandwidth used, and whether it was kept consistent across all cell types to ensure comparability.

Version 1:

Reviewer comments:

Reviewer #2

(Remarks to the Author)

Authors have addressed most of my points from the last review. One point that is missing, or not clear yet, is the translation their ex-vivo histological results to in-vivo dMRI. For example, if I want to simulate diffusion signal in grey matter using the segmented/reconstructed microstructure, how could I compensate the obtained signal with what I would expect to see in in-vivo dMRI measurements? do I need to compensate by tissue shrinkage, temperature, diffusivity, % of cellular apoptosis, etc?.

After this point is clearly discussed in the text, I would approve this manuscript for publication.

Reviewer #3

(Remarks to the Author)

All of my comments have been addressed.

Decoding Gray Matter: large-scale analysis of brain cell morphometry to inform microstructural modeling of diffusion MR signals

Dear Nature Communications Biology

We appreciate the time and effort the reviewers have dedicated to providing their valuable feedback on our manuscript. We are grateful to the reviewers for their insightful comments. We have incorporated changes to address every point raised by the reviewers. Here is a point-by-point response to the reviewers' comments and concerns, with each of the reviewers' point in black and our replies in red&blue lines.

Reviewer #1 (Remarks to the Author):

In this very well-written and structured manuscript, the team concisely presents a large-scale quantitative analysis of gray matter cellular morphometry, detailing their processing and analysis methods applied to 3D cell reconstructions sourced from an online database. They systematically compute and report a comprehensive set of structural, shape, and topological descriptors across various cell types and species. They are addressing a knowledge gap by providing quantitative benchmarks for GM microstructure, the insightful discussion linking these features to the interpretation and limitations of dMRI biophysical models. While the overall quality is high, the manuscript would benefit from a careful proofreading pass, as there are a few minor typographical errors scattered throughout the text.

Minor comments:

- Intro: The paper is organized as following (as follows)
- Intro: subsequent interpretation if (typo, of)
- A few Missing citation (??, probably a Latex issue, maybe double-check)
- Section 2.1: dendritic projections (dendritic)
- Section 2.3: cylinders of length 10m (Missing μ ?)
- section 2.4: with respect to the path distance fro the soma (fro - from)
- section 3.3: τ , across majority (of) cell types. (missing of)
- section 3.6: Additionally, the comparatively low (comparatively)
- section 4.1: One potential limitation (limitation) of our analysis is that we did not include spines, boutons and glial leaflets (leaflets)
- section 4.1: when analysing and intepreting (interpreting) time-depepnt dMRI
- section 4.1: fast exhchange (exchange)
- Section 4.2: which is far longer than the values typically. (missing the word used?)
- Section 4.2: Typo L^2 _barnch

In section 2.3, the paragraph starting with 'To calculate FA we followed' and the next one starting with 'The cellular structure is' seems to be explaining the same thing twice, is this an error?

Thank you for highlighting these errors.

All the minor corrections have been addressed and corrected in the text.

Reviewer #2

The manuscript "Decoding Gray Matter: large-scale analysis of brain cell morphometry to inform microstructural modeling of diffusion MR signals" of Aird-Rossiter et al. summarises the main

features of cellular morphometry like soma radius as well as length, radius and dispersion of neuronal projects from 3500 3D cellular reconstructions across species and cell types. Through an exhaustive method and careful considerations due to inherent biases due to the specific histology approach processing (neither in the paper or in the website specify which histology technique was used to create the SWC files), demonstrate differences and similarities between cell types and species; as well as an indication on how to use this information for biophysical modelling in diffusion MRI. This work, in synthesis, could be used as a benchmark or reference when improving or defining new biophysical models for dMRI in grey matter for in vivo and ex vivo human brain.

The manuscript is, however, quite difficult to understand at first glance making necessary to read it two or three times the methods section to understand the results. Specially the topological descriptors section, in which the motivation and use of quantities like the topological distance was still not clear.

Response (point 1)

We appreciate the feedback regarding the difficulty of understanding the topological analysis, especially given it is an unfamiliar topic for many in the field of diffusion MRI. Therefore, we have rephrased the topological description and the process of computing the barcodes to make it more accessible. Briefly, persistence diagrams capture the scale at which branch structures appear and disappear, preserving long-lived features while filtering out shorter ones (Kanari, 2020).

Amendments

New version of the topological description and the process of computing the barcodes:

“Topological persistence quantifies how long specific structural features, such as branch paths, remain across different scales. The Topological Morphology Descriptor (TMD) captures this by measuring the path lengths of connected branches, tracking both their initiation and termination points relative to the soma (Kanari 2018). This process preserves longer (more persistent) structural components while filtering out shorter ones.

To compute a neuron's topological descriptor, terminal points are evaluated by their path length from the soma. At each branch point, the shorter sibling branch is removed, and its start and end distances are recorded in a “barcode,” while the longer branch is retained. Repeating this process for all terminal points yields a barcode of paired values, each representing the initiation and termination distances from the soma.

The longer branch is retained, and this process is repeated until all terminal points have been assessed. The resulting barcode is a multiset of value pairs, each representing the initiation and termination points with respect to the soma.”

It is also important to mention that the average and standard deviation of different metrics could be biased given that the distribution (or PDFs) of these metrics are not Gaussian or Normal. This is not only important for coherence (or to use the proper estimates for the correct distributions) but also because they require a different discussion and comparison with the estimated biophysical parameters of dMRI. Take for example the discussion between mean and effective radius between histology and dMRI from Veraart et al. 2020 (<https://pubmed.ncbi.nlm.nih.gov/32048987/>).

Response (point 2)

These are extremely valid points, as such, instead of reporting the mean and standard deviation, we now report the 1st, 2nd, and 3rd quartiles. Additionally, the effective radius for both soma and branches are calculated from the estimated distributions and reported.

Effective radius measures for the gaussian phase approximation were calculated from the distributions of radii with the following equations obtained from (Olesen 2022):

$$R_{soma\ effective} = \left(\frac{\langle R_{soma}^7 \rangle}{\langle R_{soma}^3 \rangle} \right)^{\frac{1}{4}}$$

for soma effective radius; and

$$R_{branch\ effective} = \left(\frac{\langle R_{branch}^6 \rangle}{\langle R_{branch}^2 \rangle} \right)^{\frac{1}{4}}$$

for branch effective radius.

Amendments

Instead of mean and standard deviation, 1st, 2nd, and 3rd quartiles have been reported, as well as effective MR measures of radii.

Lastly, it is important to emphasize that the morphometry description of these cells cannot be compared directly with in vivo human brain. I think it is important to give an insight on how these results (or future references) could benefit in the validation of biophysical modelling for in vivo dMRI.

Response (point 3)

We agree with the reviewer. It is important to note that when analysing reconstructions from histology the sample must go through a series of processes (fixation, clearing ...) that will alter the sample morphology, such as cell shrinkage. However, these data sets, with their limitations, provide high resolution information that is still highly valuable and the only source of ground truth data with regards to cellular morphology.

Whilst this is mentioned in the text, we added more emphasis in the limitations.

Amendments

The following text has been added to the limitations section of the manuscript:

“Whilst neural reconstructions, like the ones analysed here, offer the closest ground truth for cellular morphology they are not without their limitations. During the process of acquiring and preparing tissue samples distortions of the cellular structure can occur, such as tissue shrinkage or the truncation of neural projections, implicating the accuracy of the morphological measures. Furthermore, the method used to trace and reconstruct the cellular structure from microscopy images, whether done manually or automated, can impact the accuracy and detail of morphological characteristics present in the final reconstruction. As such, it is important to acknowledge that the characteristics reported here may deviate from actual in vivo measurements.”

Before acceptance and publication, I would encourage the authors to tackle the aforementioned points. I have attached a PDF that contains a more detailed review on the manuscript (I hope my handwriting is clear enough - if not, please don't hesitate to contact me).

From the Reviewer's PDF, we report here the relevant comments and our replies:

- *“Scale problem and lack of detailing of dMRI images – signal ~\~ cellular interpretation” (point 4)*

This is a crucial point, we have now clarified it in the new introduction paragraph below:

“Currently, there are no methods to directly observe cellular microstructure in vivo, as its scale (measured in micrometers) exceeds the resolution of clinical MRI, which typically operates at the millimeter scale [20]. While non-invasive imaging techniques like MRI can provide insights, they cannot directly capture cellular-level details. With many neurological conditions, such as dementia [21] and aging [22], altering the brain structure on this cellular scale, there is a strong incentive to develop means of revealing the morphology of neural cells, in-vivo.

Although dMRI has the same millimeter-scale resolution as conventional MRI, it is sensitive to micrometer-scale structures by measuring the diffusion of endogenous molecules (e.g., water). This makes it a promising technique for overcoming MRI’s resolution limits and characterizing the brain’s *microstructure* (i.e. its cellular-scale organization) in vivo. However, dMRI’s sensitivity is indirect, and biophysical modelling of the brain tissue and the subsequent interpretation of the dMRI signal is essential to quantify histologically meaningful features of the cellular structure, and gain specificity to their changes. To this end, the microstructure imaging paradigm has been introduced over a decade ago [23]: the approach fits a biophysical model voxel-wise to the set of signals obtained from images acquired with different sensitivities to tissue microstructure, yielding maps of model parameters that it is hoped are proxies of the corresponding underlying microstructural features.”

- *“Do cell projections vary by brain region” (point 5)*

This highlights the complexity of neuronal classification, as yes, the morphology of cell types is highly variable and varies by brain region, and also by cortical layer.

We added the following comments to the introduction:

“Furthermore, neurons of the same cell type can exhibit significant morphological differences across brain regions (Peng 2021), even displaying substantial variability between cortical layers within the same region (van Aerde 2015). “

- *““Limited depth of field in some cellular reconstructions” what does this mean?” (point 6)*

We appreciate this point was somewhat ambiguous in the manuscript. Some cellular reconstructions were ‘flattened’ as a result of limitations in the imaging technique used. Please see figure below

Figure demonstrating the limit depth of cellular reconstructions as a result of imaging method (some

reconstructions appeared even more ‘flattened’), and the resulting FA tensors for unadjusted eigenvalues and adjusted eigenvalues. The corresponding FA for unadjusted eigenvalues = 0.67, and for adjusted eigenvalues = 0.30, so reduced dimensional representation results in increased FA.

The solution employed was to sort in ascending order the estimated eigenvalues ($\tau_1 \geq \tau_2 \geq \tau_3$) and set $\tau_1 = \tau_2$ (i.e., assuming the tensor to be axially symmetric around its main eigenvector), artificially recovering the complete FA tensor of the cell. However, this may provide biased estimates for cell

types known to have planar morphology, such as the Purkinje cells. For this reason, we decided to include the above Figure in the Supplementary Material and include in the new Table 4 the mean eigenvalues, the unadjusted FA, and the adjusted FA.

For clarity we reworded the paragraph in the Methods section as below:

“In some cellular reconstructions, limitations in depth of field resulting from the imaging technique lead to anisotropic inaccuracies (see Supplementary Figure 3), with cells appearing compressed along the axis perpendicular to the acquisition plane (Z-axis). This artifact can result in overestimated fractional anisotropy in said reconstructions. For this reason, alongside the FA, we also report the estimated eigenvalues in ascending order ($\tau_1 \geq \tau_2 \geq \tau_3$) and an adjusted FA (Adj.FA) were we assumed $\tau_1 = \tau_2$.”

- *“Topology on identity axis”, i.e. scale the cells to be all of the same size and then estimate topology. (point 7)*

This was an interesting idea that we did investigate as potential to identify scalable topological structures. However, as the topological descriptor encodes information about branch length and cellular extent it was determined, after consulting Dr Lida Kanari, that it was more appropriate to compare structures on their spatial dimensions and not their rescaled dimensions. We have now added an acknowledgment to Dr. Kanari’s help.

Please see the figure below for the persistence images for barcodes scaled to unity and the corresponding persistence distance matrix

- *“Use of ‘significant’ in discussion referring to significant impact on diffusion signal” (point 8)*

In this context ‘significant’ refers to the potential impact on the dMRI signal, not statistical significance. We acknowledge the ambiguity and have thus rephrased it as “measurable impact”, as well as modified section 4.2 to reflect this.

- Several minor errors have been corrected in the text.

Thank you for the thorough feedback and comments you provided in the pdf

Reviewer #3

The authors present a large-scale quantitative analysis of neural cell morphology across multiple species and cell types to inform gray matter diffusion MRI (dMRI) modeling. This is a topic of interest to the dMRI community, as gray matter (GM) analysis often lags behind white matter (WM) analysis due to the diversity of cell types and the complexity of GM microstructure.

The manuscript is well organized, and the analysis appears thorough. However, several methodological assumptions and presentation choices limit the clarity of how the findings translate to microstructural modeling.

Given the complexity of GM and the challenges involved in validating dMRI models, the standards for establishing ground-truth reference values are necessarily high, and further rigor is expected.

In addition, considering the impact and scope expected of *Communications Biology*, this work reads more as a technical resource or descriptive dataset than a biological discovery study or conceptual advance.

While the quantitative characterization of cellular features is valuable, the manuscript currently lacks sufficient mechanistic insights, novel biological hypotheses, or modeling validation that would justify publication in a high-impact biology journal.

At its current stage, this work may be better suited for a resource-focused venue such as *Scientific Data*.

Response

We thank the reviewer for raising this important point regarding novelty and scope. We fully recognize that our work primarily provides a large-scale quantitative and technical resource, and we agree that it does not offer mechanistic biological experiments in the traditional sense. Our intention, however, is to contribute a foundation that enables new avenues of discovery: by systematically characterizing cellular morphology across species and cell types, we provide a resource that can inform and constrain biophysical models of dMRI and thereby generate new biological insights into gray matter microstructure.

We also note that the editor's decision to invite revision suggests that the study is considered within the scope of *Communications Biology*. With this in mind, we have revised the manuscript to more clearly highlight both the resource aspect (as a benchmark dataset) and the conceptual contribution (linking cellular morphometry to diffusion modeling). We hope this dual framing addresses the reviewer's concern and clarifies how the work can serve both as a technical reference and as a platform for future mechanistic and hypothesis-driven studies.

Main concerns:

1. Selection Bias and Incompleteness of Reconstructions:

Although quality control is mentioned, there is no quantitative assessment of how representative the selected cells are of true cellular diversity. Many reconstructions lack axons and spines, which could significantly influence diffusion properties and branching patterns. Please quantify what fraction of the original database was excluded for each cell type and discuss how this might bias the reported morphological distributions (e.g., preferential selection of larger somas or simpler dendritic trees).

Response (point 1)

We thank the reviewer for this important comment. We agree that a quantitative assessment of inclusion/exclusion criteria is necessary to evaluate potential bias. To address this, we re-ran the entire analysis using the most recent Neuromorpho dataset (v8.6.83), in which both the number of available reconstructions and their metadata have substantially increased. This expanded our dataset from 3,598 to 11,850 reconstructions (mouse/rat: N=9,001; monkey: N=525; human: N=2,324). For each cell

type, we now explicitly report both the total number of reconstructions available in the database and the number that met our inclusion criteria. These fractions are provided in a new supplementary table.

This updated analysis allows us to demonstrate that our inclusion criteria are transparent and reproducible, and that the overall summary statistics remain stable between the earlier (~3k) and updated (~11k) datasets. We believe this reduces the likelihood of meaningful selection bias, though we acknowledge that the absence of axons and spines in many reconstructions remains an inherent limitation of the underlying data (now noted explicitly in the revised manuscript).

Search criteria									Total number of cells
	Species	Development	Condition	Celltype	Structural Domain	Morphological Attributes	Number	Rejection rate (%)	total for species
Accepted	Mouse	Adult, young, young adult	Control	Microglia	Dendrite, processes, soma, axon, no axon	Diameter, 3D, Angles	2696		5991
Total	Mouse	Adult, young, young adult	Control	Microglia	n.a	n.a	27142	90.06705475	
Accepted	Mouse	Adult, young, young adult	Control	Astrocyte	Dendrite, processes, soma, axon, no axon	Diameter, 3D, Angles	197		87.74113255
Total	Mouse	Adult, young, young adult	Control	Astrocyte	n.a	n.a	1607		
Accepted	Mouse	Adult, young, young adult	Control	Oligodendrocyte	Dendrite, processes, soma, axon, no axon	Diameter, 3D, Angles	53		56.55737705
Total	Mouse	Adult, young, young adult	Control	Oligodendrocyte	n.a	n.a	122		
Accepted	Mouse	Adult, young, young adult	Control	Pyramidal	Dendrite, processes, soma, axon, no axon	Diameter, 3D, Angles	1660		87.09677419
Total	Mouse	Adult, young, young adult	Control	Pyramidal	n.a	n.a	12865		
Accepted	Mouse	Adult, young, young adult	Control	Granule	Dendrite, processes, soma, axon, no axon	Diameter, 3D, Angles	772		68.03312629
Total	Mouse	Adult, young, young adult	Control	Granule	n.a	n.a	2415		
Accepted	Mouse	Adult, young, young adult	Control	Purkinje	Dendrite, processes, soma, axon, no axon	Diameter, 3D, Angles	139		62.33062331
Total	Mouse	Adult, young, young adult	Control	Purkinje	n.a	n.a	369		
Accepted	Mouse	Adult, young, young adult	Control	Basket	Dendrite, processes, soma, axon, no axon	Diameter, 3D, Angles	324		32.64033264
Total	Mouse	Adult, young, young adult	Control	Basket	n.a	n.a	481		
Accepted	Mouse	Adult, young, young adult	Control	Gabaergic	Dendrite, processes, soma, axon, no axon	Diameter, 3D, Angles	122		91.65526676
Total	Mouse	Adult, young, young adult	Control	Gabaergic	n.a	n.a	1462		
Accepted	Mouse	Adult, young, young adult	Control	Glutamaergic	Dendrite, processes, soma, axon, no axon	Diameter, 3D, Angles	28		15.15151515
Total	Mouse	Adult, young, young adult	Control	Glutamaergic	n.a	n.a	33		
Accepted	Rat	Adult, young, young adult	Control	Microglia	Dendrite, processes, soma, axon, no axon	Diameter, 3D, Angles	1379		66.60208283
Total	Rat	Adult, young, young adult	Control	Microglia	n.a	n.a	4129		
Accepted	Rat	Adult, young, young adult	Control	Astrocyte	Dendrite, processes, soma, axon, no axon	Diameter, 3D, Angles	139		44.4
Total	Rat	Adult, young, young adult	Control	Astrocyte	n.a	n.a	250		
Accepted	Rat	Adult, young, young adult	Control	Oligodendrocyte	Dendrite, processes, soma, axon, no axon	Diameter, 3D, Angles	5		80
Total	Rat	Adult, young, young adult	Control	Oligodendrocyte	n.a	n.a	25		
Accepted	Rat	Adult, young, young adult	Control	Pyramidal	Dendrite, processes, soma, axon, no axon	Diameter, 3D, Angles	1150		56.58739147
Total	Rat	Adult, young, young adult	Control	Pyramidal	n.a	n.a	2649		
Accepted	Rat	Adult, young, young adult	Control	Granule	Dendrite, processes, soma, axon, no axon	Diameter, 3D, Angles	112		79.18215613
Total	Rat	Adult, young, young adult	Control	Granule	n.a	n.a	538		
Accepted	Rat	Adult, young, young adult	Control	Purkinje	Dendrite, processes, soma, axon, no axon	Diameter, 3D, Angles	1		0
Total	Rat	Adult, young, young adult	Control	Purkinje	n.a	n.a	1		
Accepted	Rat	Adult, young, young adult	Control	Basket	Dendrite, processes, soma, axon, no axon	Diameter, 3D, Angles	90		79.06976744
Total	Rat	Adult, young, young adult	Control	Basket	n.a	n.a	430		
Accepted	Rat	Adult, young, young adult	Control	Gabaergic	Dendrite, processes, soma, axon, no axon	Diameter, 3D, Angles	56		13.84615385
Total	Rat	Adult, young, young adult	Control	Gabaergic	n.a	n.a	65		
Accepted	Rat	Adult, young, young adult	Control	Glutamatergic	Dendrite, processes, soma, axon, no axon	Diameter, 3D, Angles	78		0
Total	Rat	Adult, young, young adult	Control	Glutamatergic	n.a	n.a	78		
Accepted	Monkey	Adult, young, young adult	Control	Microglia	Dendrite, processes, soma, axon, no axon	Diameter, 3D, Angles	60		525
Total	Monkey	Adult, young, young adult	Control	Microglia	n.a	n.a	60		
Accepted	Monkey	Adult, young, young adult	Control	Astrocyte	Dendrite, processes, soma, axon, no axon	Diameter, 3D, Angles	0		100
Total	Monkey	Adult, young, young adult	Control	Astrocyte	n.a	n.a	110		
Accepted	Monkey	Adult, young, young adult	Control	Oligodendrocyte	Dendrite, processes, soma, axon, no axon	Diameter, 3D, Angles	0		#DIV/0!
Total	Monkey	Adult, young, young adult	Control	Oligodendrocyte	n.a	n.a	0		
Accepted	Monkey	Adult, young, young adult	Control	Pyramidal	Dendrite, processes, soma, axon, no axon	Diameter, 3D, Angles	461		73.44470046
Total	Monkey	Adult, young, young adult	Control	Pyramidal	n.a	n.a	1736		
Accepted	Monkey	Adult, young, young adult	Control	Granule	Dendrite, processes, soma, axon, no axon	Diameter, 3D, Angles	0		#DIV/0!
Total	Monkey	Adult, young, young adult	Control	Granule	n.a	n.a	0		
Accepted	Monkey	Adult, young, young adult	Control	Purkinje	Dendrite, processes, soma, axon, no axon	Diameter, 3D, Angles	0		#DIV/0!
Total	Monkey	Adult, young, young adult	Control	Purkinje	n.a	n.a	0		
Accepted	Monkey	Adult, young, young adult	Control	Basket	Dendrite, processes, soma, axon, no axon	Diameter, 3D, Angles	0		100
Total	Monkey	Adult, young, young adult	Control	Basket	n.a	n.a	9		
Accepted	Monkey	Adult, young, young adult	Control	Gabaergic	Dendrite, processes, soma, axon, no axon	Diameter, 3D, Angles	4		0
Total	Monkey	Adult, young, young adult	Control	Gabaergic	n.a	n.a	4		
Accepted	Monkey	Adult, young, young adult	Control	Glutamatergic	Dendrite, processes, soma, axon, no axon	Diameter, 3D, Angles	0		#DIV/0!
Total	Monkey	Adult, young, young adult	Control	Glutamatergic	n.a	n.a	0		
Accepted	Human	Adult, young, young adult	Control	Microglia	Dendrite, processes, soma, axon, no axon	Diameter, 3D, Angles	0		2324
Total	Human	Adult, young, young adult	Control	Microglia	n.a	n.a	0	#DIV/0!	
Accepted	Human	Adult, young, young adult	Control	Astrocyte	Dendrite, processes, soma, axon, no axon	Diameter, 3D, Angles	0		100
Total	Human	Adult, young, young adult	Control	Astrocyte	n.a	n.a	465		
Accepted	Human	Adult, young, young adult	Control	Oligodendrocyte	Dendrite, processes, soma, axon, no axon	Diameter, 3D, Angles	0		#DIV/0!
Total	Human	Adult, young, young adult	Control	Oligodendrocyte	n.a	n.a	0		
Accepted	Human	Adult, young, young adult	Control	Pyramidal	Dendrite, processes, soma, axon, no axon	Diameter, 3D, Angles	2304		8.643933386
Total	Human	Adult, young, young adult	Control	Pyramidal	n.a	n.a	2522		
Accepted	Human	Adult, young, young adult	Control	Granule	Dendrite, processes, soma, axon, no axon	Diameter, 3D, Angles	5		0
Total	Human	Adult, young, young adult	Control	Granule	n.a	n.a	5		
Accepted	Human	Adult, young, young adult	Control	Purkinje	Dendrite, processes, soma, axon, no axon	Diameter, 3D, Angles	0		#DIV/0!
Total	Human	Adult, young, young adult	Control	Purkinje	n.a	n.a	0		
Accepted	Human	Adult, young, young adult	Control	Basket	Dendrite, processes, soma, axon, no axon	Diameter, 3D, Angles	9		30.76923077
Total	Human	Adult, young, young adult	Control	Basket	n.a	n.a	13		
Accepted	Human	Adult, young, young adult	Control	Gabaergic	Dendrite, processes, soma, axon, no axon	Diameter, 3D, Angles	6		0
Total	Human	Adult, young, young adult	Control	Gabaergic	n.a	n.a	6		
Accepted	Human	Adult, young, young adult	Control	Glutamatergic	Dendrite, processes, soma, axon, no axon	Diameter, 3D, Angles	0		#DIV/0!
Total	Human	Adult, young, young adult	Control	Glutamatergic	n.a	n.a	0		

We only accepted reconstructions containing soma and projections, with 3D spatial information and branch diameters. Given these criteria we believe we only accepted the most accurate reconstructions available to us and selection bias to be negligible. This is demonstrated by the summary statistics being largely stable between the old data set (N~3k) and the new one (N~11k). See comparison between the previous Table 2 and the new one below:

Structural statistics for new data set

animal	celltype	N	R doimain	N projectoins	BO	SV soma	R soma	μ	SV branch	L branch	R branch	Cv branch	μ OD branch	R curvature	τ
MouseRat	microglia	4100	42	5	7.8	1.3	2.6	0.49	4.8	8.8	0.66	58	0.23	6.3	0.79
Monkey	microglia	60	42	8.4	9.2	1.2	2.7	0.014	16	9.5	0.19	63	0.27	16	0.74
Human	microglia	0	0	0	0	0	0	0	0	0	0	0	0	0	0
MouseRat	astrocyte	310	39	9.4	12	1.1	3.3	0.38	7.1	8.3	0.47	93	0.25	11	0.69
Monkey	astrocyte	0	0	0	0	0	0	0	0	0	0	0	0	0	0
Human	astrocyte	0	0	0	0	0	0	0	0	0	0	0	0	0	0
MouseRat	oligodendrocyte	58	110	7.5	8.1	0.81	4.2	0.12	4.8	27	0.56	120	0.18	28	0.8
Monkey	oligodendrocyte	0	0	0	0	0	0	0	0	0	0	0	0	0	0
Human	oligodendrocyte	0	0	0	0	0	0	0	0	0	0	0	0	0	0
MouseRat	pyramidal	3000	390	5.8	8.7	0.66	6.9	0.16	6.4	58	0.65	63	0.23	35	0.8
Monkey	pyramidal	510	460	7.7	8.5	0.44	7.6	0.56	7.3	64	0.58	74	0.24	41	0.79
Human	pyramidal	2400	300	6.1	7.1	0.38	8.4	0.69	5.4	75	0.77	58	0.27	65	0.75
MouseRat	granule	930	170	1.3	3.8	0.82	4.3	0.215	5.6	54	0.57	45	0.17	43	0.84
Monkey	granule	0	0	0	0	0	0	0	0	0	0	0	0	0	0
Human	granule	82	510	3	5.1	0.77	5.1	0.13	4.1	130	0.65	49	0.21	20	0.84
MouseRat	purkinje	140	160	1.7	10	0.45	7.3	0.21	4.7	10	0.57	22	0.28	26	0.77
Monkey	purkinje	0	0	0	0	0	0	0	0	0	0	0	0	0	0
Human	purkinje	0	0	0	0	0	0	0	0	0	0	0	0	0	0
MouseRat	basket	440	260	5.5	6.8	0.6	6.5	0.24	4.7	66	0.76	77	0.28	33	0.76
Monkey	basket	0	0	0	0	0	0	0	0	0	0	0	0	0	0
Human	basket	11	300	5.3	5.9	0.61	5.1	0.22	4	81	0.62	49	0.24	18	0.8
MouseRat	gabaergic	180	360	6	7.4	0.49	8.2	0.81	9.1	68	0.67	80	0.28	59	0.77
Monkey	gabaergic	4	250	8	6.3	0.62	5.2	0.27	5.8	79	0.39	51	0.33	2.3	0.73
Human	gabaergic	6	190	3.7	5.4	0.72	4.3	0.0074	7.9	68	0.36	57	0.3	19	0.74
MouseRat	glutamatergic	110	270	5.4	6.8	0.47	7.9	0.16	12	57	0.6	73	0.28	38	0.77
Monkey	glutamatergic	0	0	0	0	0	0	0	0	0	0	0	0	0	0
Human	glutamatergic	0	0	0	0	0	0	0	0	0	0	0	0	0	0

Structural statistics for old data set.

animal	celltype	N	R doimain	N projectoins	BO	SV soma	R soma	μ	SV branch	L branch	R branch	Cv branch	μ OD branch	R curvature	τ
MouseRat	microglia	150	42	7.2	8.1	0.86	3.7	0.32	4.8	12	0.56	74	0.27	25	0.74
Monkey	microglia	61	42	8.4	9.2	1.2	2.7	0.014	16	9.5	0.19	63	0.27	16	0.74
Human	microglia	0	0	0	0	0	0	0	0	0	0	0	0	0	0
MouseRat	astrocyte	260	38	7	9.1	1.3	3	0.31	5.8	9.8	0.61	100	0.23	18	0.73
Monkey	astrocyte	0	0	0	0	0	0	0	0	0	0	0	0	0	0
Human	astrocyte	0	0	0	0	0	0	0	0	0	0	0	0	0	0
MouseRat	oligodendrocyte	89	69	12	10	0.71	8.5	0.25	13	13	0.26	120	0.18	28	0.82
Monkey	oligodendrocyte	0	0	0	0	0	0	0	0	0	0	0	0	0	0
Human	oligodendrocyte	0	0	0	0	0	0	0	0	0	0	0	0	0	0
MouseRat	pyramidal	350	540	5.1	8.6	0.53	7.3	0.17	11	79	0.56	58	0.28	84	0.76
Monkey	pyramidal	870	450	5.9	7.1	0.52	6.2	0.24	5.8	99	0.59	68	0.22	70	0.81
Human	pyramidal	1300	340	6.3	7.2	0.35	9.8	0.39	5.7	80	0.88	62	0.26	61	0.76
MouseRat	granule	130	340	1.7	3.6	0.78	4.1	0.17	5.2	110	0.5	29	0.17	53	0.88
Monkey	granule	0	0	0	0	0	0	0	0	0	0	0	0	0	0
Human	granule	83	500	3	5.1	0.78	5	0.13	4.1	120	0.65	49	0.21	20	0.84
MouseRat	purkinje	120	160	1.8	9.8	0.41	7.6	0.24	4.5	10	0.58	20	0.29	27	0.78
Monkey	purkinje	0	0	0	0	0	0	0	0	0	0	0	0	0	0
Human	purkinje	0	0	0	0	0	0	0	0	0	0	0	0	0	0
MouseRat	basket	46	360	4.2	15	0.36	3.3	0.25	8	78	0.44	77	0.3	17	0.78
Monkey	basket	21	400	6	8.7	0.6	5.6	0.044	8.4	70	0.49	51	0.32	21	0.74
Human	basket	29	180	3.8	3.7	0.66	4.7	0.33	5.6	55	0.94	120	0.25	17	0.88
MouseRat	gabaergic	190	260	5.6	5.8	0.7	4.5	0.48	7.2	69	0.74	48	0.26	72	0.78
Monkey	gabaergic	5	230	8	6.4	0.63	5.1	0.27	6.2	72	0.37	52	0.33	2.3	0.73
Human	gabaergic	8	170	4.8	5.5	0.71	4.3	0.0076	8	65	0.4	67	0.29	19	0.75
MouseRat	glutamatergic	36	270	5.1	7.8	0.44	8.4	0.58	3.5	44	0.82	51	0.36	20	0.69
Monkey	glutamatergic	0	0	0	0	0	0	0	0	0	0	0	0	0	0
Human	glutamatergic	110	630	4.5	6.3	0.16	28	0.004	4	150	0.97	94	0.12	470	0.87

The below table displays the average percentage difference in descriptor between the two data sets ([new dataset – old dataset] / new dataset * 100).

Descriptor	Rdoimain	Nprojectoin	BO	SVsoma	Rsoma	μ	SVbranch	Lbranch	Rbranch	Cvbranch	μ ODbranch	Rcurvature	τ
Change (%)	4.7	2.4	3.3	7.3	7.4	9.9	8.9	1.8	7.6	5.1	-2.3	-3.6	-0.5

Given the inconsistent reconstruction of axonal components across the datasets, we decided to keep every cellular reconstruction but remove the axonal component from the SWC file to avoid any potential bias. Similarly, spines were inconsistently included in reconstructions, the majority of reconstruction not including spines, as such a function was defined to identify and remove spines from every reconstructions to avoid potential bias. We would like to clarify that no reconstruction was rejected based on it's inclusion of either axon or spines.

Amendments

All analysis has been rerun with the new contemporary data set and tables and statistics have been updated accordingly.

A table of the search criteria and rejection rate have been compiled and is included as Supplementary Table 1 in the Supplementary Material

Text to clarify how axons and spines were handled:

“Despite their essential role in synaptic formation and the brains micro connectivity, dendritic spines (small protrusions on neuronal dendrites that form synaptic connections) were largely absent from the reconstructions. Furthermore, their inclusion biases the statistical analysis of overall cellular morphology, such as artificially reducing branch length and inflating branch order. As a result, for all reconstructions spines were identified (if present) and removed.

Additionally, due to the inconsistency in axonal reconstructions (axons largely being absent or heavily truncated) in the datasets, they were not considered in the analysis (if axonal components were present, they were identified from the swc format and removed from the reconstruction). As a result, the statistics of the projections are for dendritic projections only.”

2. Impact of Missing Features on Biophysical Modeling:

The analysis does not account for important structures such as dendritic spines, boutons, or glial leaflets, which can substantially affect surface-to-volume ratios and exchange dynamics. Although this limitation is acknowledged in the discussion, its implications should be evaluated quantitatively. For example, how would including a typical spine density ($\sim 1-2$ spines/ μm) alter S/Vbranch estimates? A simple modeling exercise would strengthen the conclusions regarding exchange timescales.

Response (point 2)

Thank you for drawing attention to this point, whilst it was not possible to accurately characterise axons and spines given their absence in the reconstructions available to us, their presence in real brain tissue will affect the diffusive dynamics.

For example, axons have a smaller radius, larger branching order, and greater cellular domain (which is the reason axonal projections are frequently heavily truncated or not reconstructed as axons often extend well beyond the sample slice). Their smaller radii will result in greater surface to volume ratio, which will have an impact on residence and exchange times.

Moreover, spines have a significant impact on diffusivity, not only in increasing the surface to volume ratio but introducing compartments that act to ‘trap’ and restrict diffusion processes within the cell (Palombo et al. Neuroimage 2017, Simsek et al. arxiv 2025, Chakwizira et al. arxiv 2025).

Assuming a spine density of 1 spine/ μm , we estimated its effect on both cellular and branch surface-to-volume ratios. As shown in the figure below, this density increases the surface-to-volume ratio by approximately 1.2–1.3 times (i.e. 20-30% larger) compared to the value without spines.

Additionally, for an increased spine density of 2 spine/ μm , the resulting increase of branch surface to volume ratio is $\sim 50\%$.

This would of course have a consequential impact on residence time estimations, leading to 15-35% shorter times. We have now added this results in the Supplementary Materials and a paragraph discussing this aspect in the Discussion section.

Amendments

Impact of spines already stated in discussion:

“One limitation of our analysis is that we did not include spines, boutons and glial leaflets, that can occupy up to 20% of the GM volume (Kasthuri et al, Cell, 2015). Dendritic spines influence intracellular diffusion by increasing the surface-to-volume ratio and introducing structural complexity and compartmentalization within dendrites (Palombo et al., NeuroImage, 2017; Simsek et al., arXiv, 2025; Chakwizira et al., arXiv, 2025). Using simple modelling (see Supplementary Fig.5), we estimated that a spine density of 1 spine/ μm increases the branch surface-to-volume ratio by approximately 20–30% in rodent neurons, leading to a $\sim 15\text{--}20\%$ reduction in residence time. At a density of 2 spines/ μm , the surface-to-volume ratio increases by $\sim 50\%$, with a corresponding $\sim 35\%$ reduction in residence time. Beyond these surface effects, spines act as distinct compartments: their narrow necks serve as diffusion barriers, restricting molecular exchange between the spine head and dendritic shaft and thereby altering diffusive dynamics (Simsek et al., arXiv, 2025; Chakwizira et al., arXiv, 2025). Consequently spine morphology represent an important morphological feature to be considered in models of molecular diffusion to accurately reflect in vivo conditions.”

3. Topological Comparisons Lack Statistical Testing:

The paper reports global topological distances between cell types based on persistence diagrams but does not assess the statistical significance of these differences. Is it possible to perform permutation tests or bootstrapping to determine whether the observed distances (e.g., neuron vs. glia) are robust and meaningful beyond random variation?

Response (point 3)

Thank you for the feedback regarding the statistical significance of the topological analysis, it is a highly valid point.

Amendments

Under bootstrapping with iterations $n=1000$, it was found that all distances were statistically significantly between cell types, except basket and glutamatergic rodent cells with a p value of 0.16.

4. Relevance of Residence Time Estimates for In Vivo dMRI:

In Figure 6, intracellular residence time distributions are computed assuming a fixed permeability of $20 \mu\text{m/s}$. However, recent studies suggest a broader range of permeability values for axons (e.g., https://doi.org/10.1007/978-3-031-43993-3_13). Do soma bodies and dendritic branches show similarly variable permeability, or is the fixed value assumption justified?

Given previous work shows the residence time depends on the permeability and radius (<http://dx.doi.org/10.1016/j.neuroimage.2017.02.013>),

it would strengthen the analysis to show how sensitive the residence time distributions are to different permeability values (e.g., plotting low and high permeability cases). This would help assess the robustness of the conclusion that exchange cannot be ignored.

Response (point 4)

We thank the reviewer for this very helpful suggestion. We agree that assuming a single fixed permeability may oversimplify the biological variability, particularly given recent evidence for a broader range in axons.

To our knowledge there is no reported difference in membrane permeability between cell soma and cellular branches; we could only find speculations regarding lower permeability in cell soma due to thicker and more complex membrane.

Following the suggestion, we have now computed and reported the residence times for low and high permeabilities. Residence times, τ , were calculated from the surface-to-volume ratios of cell types with the follow expression

$$\tau = \frac{1}{\frac{S}{V} \kappa}$$

where κ is the permeability. This is equivalent to the expression linking permeability to radius in Nedjati-Gilani's paper.

And exchange times calculated using (Fieremans 2010)

$$\tau_{ex} = \tau f_2$$

Where f_2 is the extracellular volume fraction (assumed to be 0.3 for the following figures)

Figure below demonstrates the impact on residence and exchange times for a range of permeabilities ($\kappa=2-35\mu\text{mms}^{-1}$).

We now added a dedicated figure (Fig.7) regarding exchange estimates only and expanded the discussion under Neural cell membrane permeability to water, also including these new plots. In addition, we have reformatted Fig.6 for better visibility and added the MR effective radius estimates alongside the soma radius distributions.

“ The broad range of permeability values present in the literature result in a large range of residence and exchange times. Consequently, to accurately quantify the impact of permeability on the dMRI signal, an accurate measure of the membrane permeability of the tissue being investigated is needed. Here, we provide reference values of intra-branch, intra-cellular residence times and corresponding exchange times for any possible permeability value within the range 2-35 $\mu\text{m/s}$, observed in vivo} (Fig7). Using a representative value of low (2 $\mu\text{m/s}$) and high (20 $\mu\text{m/s}$) permeability, we also estimated the distribution of intra-cellular and intra-branch residence times, which span from a few milliseconds to hundreds of milliseconds} (Fig.7).”

5. Resource Sharing:

While the descriptors are well defined, they are largely derived from existing modelling frameworks (e.g., DTI, NODDI), which themselves have known limitations. Will the authors share the 3D meshes of the analyzed cells? Making these data publicly available would be a valuable resource for the community, enabling studies of dMRI signal formation (e.g., random walk simulations, Bloch–Torrey equation modeling) and the development of more accurate microstructural models.

Response (point 5)

We agree that providing a set of representative meshes would be a valuable resource for to the community. As such we will happily provide exemplar surface meshes of each cell type and species (800 meshes in total, approximately 50 for each cell type and species) as well as a script to create a surface meshes from any swc file.

Amendments

We will provide representative meshes and the code used to mesh swc files through the data sharing initiative. We have now added into the Abstract, Introduction and Discussion sections mention to this new resource.

Two example meshes for Rat astrocytes.

Minor comments:

Typographical Errors:

Minor typos appear throughout, including "intepreting" → "interpreting", "barnch" → "branch", incomplete sentence "To calculate FA we followed the method outlined in.", and missing figure references such as "The distributions for all the morphological features are reported in supplementary Fig.??." Please proofread carefully.

Clarify Figure 2 Annotation:

The figure illustrating structural descriptors would benefit from directly labeling the soma and projections on the 3D rendering for improved clarity.

Persistence Diagram Methods:

It is mentioned that kernel density estimation was applied to persistence images. Please specify the kernel bandwidth used, and whether it was kept consistent across all cell types to ensure comparability.

Response

Thank you for highlighting these errors.

All minor corrections have been addressed and corrected in the text.

Decoding Gray Matter: large-scale analysis of brain cell morphometry to inform microstructural modelling of diffusion MR signals

Dear Nature Communications Biology

We appreciate the time and effort the reviewers have dedicated to providing their valuable feedback on our manuscript. We are grateful to the reviewers for their insightful comments. We have incorporated changes to address every point raised by the reviewers. Here is a point-by-point response to the reviewers' comments and concerns, with each of the reviewers' point in black and our replies in red&blue lines.

Reviewer #2 (Remarks to Author):

Authors have addressed most of my points from the last review. One point that is missing, or not clear yet, is the translation their ex-vivo histological results to in-vivo dMRI. For example, if I want to simulate diffusion signal in grey matter using the segmented/reconstructed microstructure, how could I compensate the obtained signal with what I would expect to see in in-vivo dMRI measurements? do I need to compensate by tissue shrinkage, temperature, diffusivity, % of cellular apoptosis, etc?.

After this point is clearly discussed in the text, I would approve this manuscript for publication.

Response to Reviewer

We thank the reviewer for raising this important point about translating our ex-vivo measures of morphological characterisation to in-vivo diffusion MRI (dMRI) simulations. We agree that clarifying these issues will strengthen the manuscript and have added a dedicated paragraph (see below). In brief:

- **Temperature / intrinsic diffusivity:**
As for any synthetic signal, be it through an analytical expression or through simulation within a given geometry, the diffusivity is a variable parameter. In first approximation, we can use the Stokes-Einstein relationship to link the diffusivity to the temperature: $D = k_B T / (3\pi\eta\sigma)$, where D is the diffusion constant, k_B is the Boltzmann constant, T is absolute temperature, η is the fluid's shear viscosity, and σ is the particle's diameter. As such the diffusivity can be defined as to reflect the experimental conditions, whether it be ex vivo or in vivo. Thus, whilst temperature affects the value of water diffusivity, it does not complicate the translation of the morphological data to in vivo conditions.
- **Membrane permeability and exchange:**
Post-mortem changes may alter membrane integrity and therefore water exchange, however, as for water diffusivity, this parameter can be tailored to reflect the appropriate conditions when generating synthetic dMRI signals. Many Mote Carlo based simulators of dMRI signals, such as CAMINO, MCDC and MCMR allow for tuneable surface permeability. The remaining challenge will be to determine which permeability value to use. Future work combining in-vivo dMRI methods sensitive to

exchange with simulated signals will help constrain appropriate permeability parameters.

- **Tissue shrinkage during fixation:**

Chemical fixation can induce anisotropic tissue shrinkage of 10–40 %, with extracellular space suffering significant shrinkage. As such it is important to consider the impact of sample shrinkage on the resulting cellular structure and morphology when translating to in vivo conditions. Neuromorpho does record if an archive reported tissue shrinkage and whether any corrections were made to the resulting reconstructions. We have collated this information for the reconstructions used in our analysis, with the majority not reporting on tissue shrinkage. From the Neuromorpho database reported tissue shrinkage ranges from >1 to 50% with an average of 15%. If reliable estimates of shrinkage are available for a given tissue preparation, the structural characteristics and reconstructed meshes can in principle be rescaled accordingly, effectively correcting for any potential bias.

- **Cellular apoptosis or other viability changes:**

Apoptosis is not prominent postmortem, since it is an active cellular process that rapidly decreases postmortem (Krassner 2023), and cell membranes are generally resistant to postmortem decomposition, with reports that there were no associated significant changes in the size and form of the cells even after a 22h postmortem interval (Krassner 2023). Another study found that even after 2 months “histomorphology was excellently preserved” in the brain of a body held at 3°C (Gelpi 2006). Given this information we believe that the cellular reconstructions analysed are representative of their in vivo counter parts. However, if accounting for apoptosis or other changes in cellular composition is necessary to match the experimental conditions (e.g., ex vivo vs in vivo), this can be addressed by scaling accordingly the relative contribution of the total dMRI signal from each cellular components. The total dMRI signal from a macroscopic imaging voxel is the volume weighted sum of the dMRI signals from each cellular compartment. Therefore, the associated weights can be adjusted to match any known composition.

Amendments:

From Ex-vivo Histology to In-vivo dMRI

Although the translation of our ex-vivo histological results to in-vivo dMRI is not straightforward, some considerations can be made to aid the interpretation of in-vivo dMRI measurements and design numerical simulations mirroring more closely the expected in-vivo condition.

- **Temperature / intrinsic diffusivity:** as for any synthetic signal, be it through an analytical expression or through simulation within a given geometry, the diffusivity is a variable parameter. In first approximation, we can use the Stokes-Einstein (Einstein, 1905) relationship to link the diffusivity to the temperature: $D = k_B T / (3\pi\eta\sigma)$, where D is the diffusion constant, k_B is the Boltzmann constant, T is absolute temperature, η is the fluid's shear viscosity, and σ is the particle's diameter. As such the diffusivity can be defined as to reflect the experimental conditions, whether it be ex-vivo or in-vivo. Thus, whilst temperature affects the value of water diffusivity, it does not complicate the translation of the morphological data to in-vivo conditions.

- **Membrane permeability and exchange:** post-mortem changes may alter membrane integrity and therefore water exchange, however, as for water diffusivity, this parameter can be tailored to reflect the appropriate conditions when generating synthetic dMRI signals. Many Monte Carlo based simulators of dMRI signals, such as CAMINO (Cook, 2005), MCDC (Rafael-Patino, 2020) and MCMR (Cottaar, 2025) allow for tuneable surface permeability. The remaining challenge will be to determine which permeability value to use, and this depends on the specific investigation. Future work combining in-vivo dMRI methods sensitive to exchange with simulated signals will help constrain appropriate permeability parameters.
- **Tissue shrinkage during fixation:** tissue samples undergo many processing steps before imaging which can cause shrinkage. Analysis of the reconstructions used here revealed that reported tissue shrinkage ranged from >1% to 50% with an average of 15%. Whilst no attempt was made to correct for this in our analysis, as most reconstructions do not report tissue shrinkage, structural characteristics or the provided meshes can be scaled appropriately given this information to remove potential bias.
- **Cellular apoptosis or other viability changes:** apoptosis is minimal postmortem, as it is an active process that rapidly ceases after death (Krassner 2023). Cell membranes are also resistant to decomposition, with no significant changes in cell size or shape even after 22 hours postmortem (Krassner 2023). As such, the analysed cellular reconstructions likely represent their in-vivo counterparts. If accounting for apoptosis or other compositional changes is required (e.g., ex-vivo vs in-vivo), this can be done by adjusting the relative contributions of each cellular component to the total dMRI signal, which is the volume-weighted sum of signals from all compartments. The total dMRI signal from a macroscopic imaging voxel is the volume weighted sum of the dMRI signals from each cellular compartment. Therefore, the associated weights can be adjusted to match any known composition.

Decoding Gray Matter: large-scale analysis of brain cell morphometry to inform microstructural modeling of diffusion MR signals

Charlie Aird-Rossiter^{1,2,*}, Hui Zhang³, Daniel C. Alexander³,
Derek K. Jones¹, and Marco Palombo^{1,2,*}

¹*Cardiff University Brain Research Imaging Centre (CUBRIC), School of Psychology, Cardiff University, Cardiff, United Kingdom*

²*School of Computer Science and Informatics, Cardiff University, Cardiff, United Kingdom*

³*UCL Hawkes Institute and Department of Computer Science, University College London, London, United Kingdom*

* *Corresponding authors: Charlie Aird-Rossiter, aird-rossiterc@cardiff.ac.uk and Marco Palombo, palombom@cardiff.ac.uk*

Abstract

Grey matter structure is a key focus in neuroscience, as cell morphology varies by type and can be affected by neurological conditions. Understanding these variations is essential for studying brain function and disease.

Diffusion-weighted MRI (dMRI) is a powerful tool for examining cellular microstructure *in vivo*, but its accuracy depends on identifying which morphological features influence its measurements. Despite growing interest, no systematic report has defined key neural cell traits.

We analyzed more than 3,500 3D cellular reconstructions across three species and nine cell types, establishing reference values for critical traits. These fall into three categories: structural, shape, and topological features.

Beyond defining these traits, we assess their relevance for dMRI, identifying which neural features it can be sensitive to. This work provides essential benchmarks for gray matter research, aiding in the interpretation of neuroimaging data and improving brain tissue models.

1 Introduction

Despite the widespread use of biophysical models in dMRI to infer cellular-scale structure, gray matter (GM) remains poorly understood due to a lack of ground-truth morphological data. Unlike white matter (WM), where axonal

morphology is well characterized, GM cellular features—critical for accurate dMRI modeling—are rarely quantitatively analyzed. This knowledge gap leads to overly simplistic or unfounded model assumptions, potentially reducing the reliability of microstructural imaging in GM. Here, we address this challenge directly by systematically characterizing the statistical distribution of key morphological features across different species, providing a much-needed empirical foundation for improving GM dMRI models.

Grey matter is composed of a range of cells, mainly differentiated into neuronal and glial cells, featuring a plethora of morphological characteristics. Neurons are fundamental functional units of the nervous system, specialized in the transmission and integration of electrical and chemical signals within the brain. Supporting the neurons are the glial cells (e.g. astrocytes, microglia and oligodendrocytes) that are crucial in maintaining health and functionality.

First studied in depth by Ramon y Cajal [1], neuronal morphology offers insights into the complex structure and function of the brain.

Neural cells exhibit a remarkable diversity in shape and size, each adapted to its specific function [2]. They can be classified based on morphological, molecular, and physiological characteristics [3]. Both neurons and glial cells share a common structural organization, consisting of a central soma and branching projections, yet they vary significantly across brain regions [4, 5]. For instance, Purkinje cells in the cerebellum display intricate dendritic arborization, whereas granule cells in the cerebral cortex exhibit a much simpler morphology.

The brain contains approximately 86 billion of these neural¹ cells [6]. Cortical GM is composed of 10–40% cell bodies (soma) of neural cells; 40–75% neurites: neuronal dendrites, short-range intra-cortical axons, the stems of long-range axons extending into the WM and glial cell projections which intermingle with each other to form a dense and complex network; 15–30% highly tortuous extra-cellular space (ECS); and 1–5% vasculature [7, 8, 9]. In adults, the glia to neuron ratio is 1.32/1.40 for males/females respectively. The proportion of glial cells (by cell count) was estimated to be 77% oligodendrocytes, 17% astrocytes and 6% microglia [10, 11].

The ECS occupies a volume fraction of 15–30% in normal adult brain tissue, with a typical value of 20%, that falls to 5% during global ischemia (the expected state during classical fixation) [12]. The ECS has an average tortuosity (defined as the ratio between the true diffusion coefficient and the effective diffusivity of small molecules such as inulin and sucrose) of 2–3 [12], due to its labyrinthine porous matrix, the presence of long-chain macromolecules, transient trapping in dead-space microdomains, and transient physical-chemical interaction with the cellular membranes. The average neuron-microvessel distance in brain GM is 20 μm [13].

The morphology of neurons and glia is revealed through staining techniques such as confocal or electron microscopy [14, 15, 16], which can image cellular structures with very high resolution (down to a few nanometers) [17].

¹neural should not be confused with neuronal. Neural means relating to the nervous system (hence referring to any cell type in the brain: from neurons to glia), while neuronal means relating to neurons.

→ 1/3 to 1/4 reduction of ECS.

Scale problem
and lack of detailing
of dMRI images
i.e. signal of cellular
information.

Currently, there are no methods to directly observe cellular microstructure in vivo, as its scale (measured in micrometers) exceeds the resolution of clinical MRI, which typically operates at the millimeter scale [18]. While non-invasive imaging techniques like MRI can provide insights, they cannot directly capture cellular-level details. With many neurological conditions, such as dementia [19], as well as aging [20], altering the brain structure on this cellular scale, there is a strong incentive to develop means of revealing the morphology of neural cells, in-vivo.

Given its sensitivity to the micrometer length scale, dMRI is a promising technique to address the resolution limit of MRI and characterize the brain structure in vivo at the cellular scale, that is, the *microstructure*. However, this sensitivity is indirect, and biophysical modelling of the brain tissue and the subsequent interpretation of the dMRI signal is essential to quantify histologically meaningful features of the cellular structure, and gain specificity to their changes. To this end, the *microstructure imaging* paradigm has been introduced over a decade ago [21]: the approach fits a biophysical model voxel-wise to the set of signals obtained from images acquired with different sensitisations to tissue microstructure, yielding maps of model parameters that it is hoped are proxies of the corresponding underlying microstructural features.

Successful examples of the microstructure imaging paradigm include Neurite Orientation Dispersion and Density Imaging (NODDI) [22] and the White Matter Tract Integrity (WMTI) [23] to characterise the diffusion of water within WM, revealing insight into the structure of axonal bundle tracks and other anatomical features, such as axon diameter [24, 25].

Building on the success of dMRI in WM, there has been growing interest in applying it to GM to characterize cellular morphology in vivo [26, 27, 28, 29, 30, 31, 32]. dMRI has already been used to distinguish different cortical regions and reveal laminar structures within GM [33]. Significant effort has been made to develop also models to better describe the diffusion signal within GM and reveal anatomical information about the microstructure, such as the Soma And Neurite Density Imaging (SANDI) [30] to characterize soma and neurite density, the Neurite EXchange Imaging (NEXI) [31] and the Standard Mmodel with EXchange (SMEX) [32] to characterize the water permeative exchange between neurites and extracellular space, and combinations of these, such as SANDI with exchange (SANDIx) [32].

While substantial effort has been made to design, validate, and translate to clinics biophysical models for WM [21, 34, 35], the GM counterpart is lagging. This disparity stems from the greater complexity of the tissue, which renders the design of biophysical models for GM microstructure imaging more challenging [36].

Building accurate dMRI models for GM requires a fundamental understanding of its microstructural features. Key questions remain: what morphological properties influence the dMRI signal? How can they be measured reliably? How should these properties guide model development? With cellular morphology varying by cell type as well as brain region, there is a necessity for a thorough analysis of characteristics based on these criteria.

typo.

Here, we aim to correct this imbalance with a comprehensive analysis of GM cellular morphology, looking at structural and topological morphology, and shape descriptors of over 3,500 real three-dimensional reconstructions from mouse, rat, monkey, and human brain cortex.

The paper is organized as following. We first provide quantitative information on the anatomy of brain GM tissue at the cellular scale. We characterize the morphology of neural cells using structural, topological and shape descriptors. We then review the range of dMRI measurements and biophysical models available to probe this anatomy and highlight limitations and caveats, ultimately providing guidelines on how to model GM microstructure from dMRI signals.

2 Materials and Methods

2.1 Microscopy Dataset

In order to measure characteristic features for specific cell types, the open access repository Neuromorpho.org [37] which has a comprehensive range of cellular reconstructions available was used. We downloaded and analysed three dimensional reconstructions of 3,598 brain cells, from mouse, rat, monkey, and human, in the form of SWC files². The SWC file defines a set of labelled nodes connected by edges characterizing the three-dimensional structure of each cell Fig.1.

Eight representative cell-types were acquired: microglia, astrocyte, pyramidal, granule, purkinje, glutamatergic, gabaergic, and basket cells, from mouse/rat (N=1,358), monkey (N=948), and human (N=1,292). Although the NeuroMorpho database contains over 270,000 cellular reconstructions, only 4,278 satisfied our inclusion criteria: healthy controls; having complete reconstruction of all the dendrites; containing full information about diameters and angles; and being three-dimensional reconstructions. Of these, 680 did not pass our visual inspection on the quality of the reconstruction and were excluded. Our quality assessment criteria include: consistent estimates of dendritic diameters (e.g., most of the rejected reconstructions had a fixed nominal diameter for all the branches instead of the real one); continuity of the cellular processes; minimal reconstruction artifacts (e.g., most of the rejected reconstructions had artifactual shrinkage in the one direction).

To reveal the skeletal structure of the cells, dendritic spines—small protrusions on neuronal dendrites that form synaptic connections—were excluded from the cellular reconstructions. This prevents potential bias in the statistical analysis of overall cellular morphology, such as artificially reducing branch length and inflating branch order. Additionally, due to the inconsistency in axonal reconstructions (axons being absent or not fully represented) in the data sets, they were not considered in the analysis. As a result, the statistics of the projections are for dendritic projections only.

²The "SWC" encodes for the last names of its initial designers Ed Stockley, Howard Wheal, and Robert Cannon and is an ASCII text-based file that describes three-dimensional neuronal or glial morphology.

(comment post)
~~review~~ full review
↓
Maybe as my ignorance as a non-biologist, but would I expect differences in cell types projections in a specie depending on where in the cortex was measured?
For example, would astrocytes from motor cortex be different to visual cortex?
Maybe that explains the intra-variability between species for a specific cell type.

Figure 1: An example of SWC file and how it relates to the cellular geometry. We highlight the structural elements used to estimate the morphological features. Note that the first node in the SWC file is the so called 'root'. It often coincides with the soma's centre and it is used to compute metrics.

The cellular reconstructions were analysed in Matlab using custom scripts, exploiting functions from validated suites (TREES [38], Blender [39], Toolbox-Graph [40]). All the codes and SWC files used in this work will be made publicly available on GitHub upon publication.

2.2 Structural descriptors

The structural analysis describes the constituent parts of the cell, including the soma and cellular projections, offering vital information about the cell's fundamental structure, such as the effective soma radius and the branch angle between daughter branches.

For the structural analysis, such features or descriptors, determined to be crucial to microstructure modelling based on current literature [41, 30, 42, 43, 44, 45, 46, 47, 48, 49, 50], were estimated from the acquired cellular reconstructions. This set of features allows for a deeper understanding of how each fundamental aspect of brain cell morphology influences the diffusion of molecules within the intracellular space. For instance, the size of the soma and cellular projections can provide insights into the characteristic length scales of intracellular restrictions, while the branching, tortuosity, undulation, and calibre variation of the projections can inform on time-dependent diffusion processes. Additionally, the surface-to-volume ratios of the soma and projections offer valuable information on exchange dynamics.

The features of the soma and cellular projections differ significantly, and our analysis accounts for this by organizing the structural descriptors into relevant categories. These categories are: (1) *soma* (the characteristics defining the cell body); (2) *projections* (the set of characteristics defining the cellular projections'

structure as interconnected branches) and (3) *general* (describing the general cellular characteristics).

Soma

- R_{soma} : the effective radius of the soma (the radius of a sphere of equivalent volume)
- η_{soma} : the proportion of the surface area covered by projection interfaces
- S/V_{soma} : the soma surface to volume ratio

Projections

- $\langle R_{branch} \rangle_s$: the mean effective radius of segments along branch s
- CV_{branch} : a measure of branch beading
- L_{branch} : the branch length
- S/V_{branch} : the branch surface to volume ratio
- $\langle \mu OD_{branch} \rangle_s$: a measure of mean branch undulation
- $\langle R_c \rangle_s$: a measure mean branch curvature
- τ_{branch} : a measure of branch tortuosity
- θ_{branch} : the angle formed by two bifurcating branches

General

- R_{domain} : the extent of the cellular domain
- $N_{projection}$: the number of primary projections radiating from the soma
- BO : the degree of branching of the cellular projections

Fig.2 illustrates these descriptors, and a summary of their definitions is reported in Tab.1.

From the information in the cellular reconstruction SWC file, the nodes/edges defining the projections from those belonging to the soma were separated: all the nodes and corresponding edges within a distance 1.5x the nominal soma radius (radius of the first node in the SWC file) from the first node, namely the 'root', were assigned to the soma; the remaining ones to the projections.

Using this soma threshold the cell was resampled, preserving the nodes that lie within the soma threshold. From this resampled soma the 3D surface mesh was constructed using Blender and the volume, V_{soma} , and surface, S_{soma} , of the soma calculated. The soma volume can be expressed in terms of the effective soma radius R_{soma} . Additionally, we calculated also the soma surface-to-volume ratio as S_{soma}/V_{soma} ; and the fraction of the soma surface covered by the

typo?

why 1.5x?

Morphological feature	Definition
R_{soma}	Radius of sphere of volume equivalent to the soma surface volume
$\langle R_{branch} \rangle_s$ ($\langle R_{branch}^2 \rangle_s$) ^{1/2}	Mean (standard deviation) of subsegments' radius along s
CV_{branch}	Coefficient of variation of branch radius
L_{branch}	Sum of subsegments' length in s
S/V_{branch}	Ratio between the sum of the surfaces of all the subsegments in s and the sum of their volume
$\langle \mu OD_{branch} \rangle_s$	Mean microscopic orientation dispersion
$\langle R_c \rangle_s$	Mean radius of curvature of s
τ_{branch}	Ratio between distance between ends of s and L_{branch} ; branch tortuosity is τ_{branch}^{-1}
BO	The number of consecutive bifurcations of the cellular projections
N_{proj}	The number of primary projections radiating from the soma
η_{soma}	Ratio between the total cross-sectional area of the N_{proj} primary projections and the total surface area
R_{domain}	Distance of the furthest node from the soma

all of these estimated assuming that can contribute as part of distal metric.

Table 1: Definition of the structural descriptors investigated.

cellular projections as sum of the projection connection area divided by the soma surface area (η_{soma}).

From the nodes/edges defining the projections, the individual branches composing the cellular projections were identified, delimited by either branching or termination nodes. The individual branches are comprised of cylindrical sub-segments, defined by the edges and their associated radius. The central line defined by these sub-segments defines a curvilinear path, s . From this path, s , metrics for the projections features were computed. Branch beading is reported as the coefficient of variation of the branch radius along the branch length (CV_{branch}). The branch surface-to-volume ratio (S/V_{branch}) was determined as the sum of the sub-segments area divided by the sum of the sub-segments volume. Branch undulation ($\langle \mu OD_{branch} \rangle_s$) is calculated as the mean angle subtended by the vector of the individual sub-segments and the vector made by the branch start and end points.

Finally, the general metrics were computed. The number of projections $N_{projection}$ was found by identifying the number of branches that cross the soma threshold. And the branch order BO is defined as the number of consecutive bifurcations of the cellular projections.

We provide the mean and standard deviation for each structural descriptor for each cell type for each species analysed. Moreover, we also provide value distributions for features of relevance to biophysical modelling of diffusion in GM, such as R_{soma} ; L_{branch} and estimated intracellular residence times based on the surface-to-volume ratio of the whole cell: S/V_{domain} and the individual branches: S/V_{branch} .

Additionally, the relationship between structural features was analysed by calculating the Spearman's rank correlation coefficient.

2.3 Shape descriptors

Many diffusion MRI models represent cellular structure as a collection of randomly oriented cylinders [22, 54]. Initially applied to the signal from white matter where axons can be simplified to a collection of cylinders/sticks this model has been shown to apply to the dendrites of neurons [30, 31] and has been incorporated into current grey diffusion MRI-based models. The cellular

this could make sense for dMRI models in GM.

Figure 2: Illustration of the structural descriptors investigated for an exemplar cell. We estimated general features of the whole structure and separated soma from projections, processing them individually to estimate a set of other relevant features. Additionally, we display the Gaussian curvature of the soma surface to show that it is a non-spherical geometry (always positive but not constant). A limitation of the current approach (and the majority of existing tools [51, 52, 53]) is the slightly inaccurate definition of the soma surface, as shown in the top right corner (arrows).

structure is modeled as a collection of cylinders, enabling the computation of fractional anisotropy (FA).

To calculate FA we followed the method outlined in. This involves decomposing the cell into its primary projections and further segmenting these into cylinders of length $l = 10\mu m$. From these segments the FA can be calculated by first computing the scatter matrix of the weighted line segments (weighted by their volume), from which the eigenvalues can be found. The FA can be calculated from these eigenvalues using the following equation.

The cellular structure is modeled as a collection of cylinders to compute fractional anisotropy (FA) following the method in [43]. The cell is decomposed into primary projections, segmented into cylinders of length 10m. FA is calculated by first computing the scatter matrix of the weighted line segments orientations (weighted by volume), from which the eigenvalues (τ_i) are derived (Fig.3). FA is then calculated as:

$$FA = \sqrt{\frac{3}{2} \frac{(\tau_1 - \tau)^2 + (\tau_2 - \tau)^2 + (\tau_3 - \tau)^2}{\tau_1^2 + \tau_2^2 + \tau_3^2}}$$

Due to the limited depth of field in some cellular reconstructions, caused by the acquisition method, some cells appear flattened perpendicular to the acqui-

What do you mean?

??
mistake here!

Figure 3: **A demonstration of procedure used to decompose the cellular structure into a set of average lines segments.** A. the complete cell. B. cell decomposed into topological persistence components. C. cell further decomposed into 10 μm segments. D. average line segments fitted to cell segments.

sition plane, leading to an artificially high FA for some cellular orientations'.
 (I think it is important to specify for some orientations) because a long cell that is squashed across its length becomes more anisotropic. A long cell that is squashed perpendicular to its long axis will be come less anisotropic. To account for this, assuming that cells are axially symmetric, if the first eigenvalue was significantly lower than the second (after sorting), $\tau_1 \ll \tau_2$, then the first eigenvalue was set to equal second, $\tau_1 = \tau_2$, and all eigenvalues were normalized such that, $\sum_{i=1}^3 \tau_i = 1$.

??
 typo -
 typo.
 → this assumption is weird.

The mean and standard deviation are reported for all cell types.

In addition to the FA of the line segments, their orientation dispersion (OD) was also computed. OD provides additional insight into the degree of anisotropy in cellular structures, complementing FA by describing the variability in the orientations of the neuronal projections. The variability is typically characterised through a Watson distribution, which is a probability distribution of orientations around the primary axis on the unit sphere [55]. With the degree of clustering defined by the concentration parameter, κ . This concentration parameter can be used to calculate the OD through the following equation [22].

$$OD = \frac{2}{\pi} \arctan(1/\kappa)$$

For cases where the data is not axially symmetric, such as cellular reconstructions with limited depth, the Watson distribution is insufficient. Instead, we applied the more general Bingham distribution, using MATLAB toolbox libDirectional [56] (Fig.4), which accounts for orientation variability along mul-

Figure 4: **A comparison between two cell types, mouse/rat pyramidal and granule cells.** Showing exemplar cells overlaid with decomposed line segments, the line segments centered at the origin, and orientation distribution about the z axis of the line segments and the analytical distribution given the calculated Watson concentration parameter

multiple axes. The Bingham distribution models anisotropic orientation data and provides two concentration parameters (κ_1 and κ_2), corresponding to the clustering along two orthogonal directions.

From the resulting concentration parameters, if the first concentration parameter was significantly larger than the second $\kappa_1 \gg \kappa_2$, indicating the orientation data was highly planar, κ_2 was used as the Watson distribution parameter κ . Otherwise, the average of κ_1 and κ_2 was used as the Watson distribution parameter κ .

This approach provides a robust calculation of the Watson distribution across isotropically and anisotropically orientated data sets

what about of scenarios of $\kappa_1 \sim 1$ or $\kappa_2 \sim 0$.

Figure 5: **A representation of the process of decomposing a cellular structure** (here a mouse/rat pyramidal cell) into its corresponding topological persistence bar code for an apical and basal projection. A. The complete cellular structure of an exemplar cell, apical projection in blue, and exemplar basal projection in red. B. and C. Detail of apical and basal projections being assessed. D. Corresponding barcodes for apical (in blue) and exemplar basal (in red/orange) components. E. persistence diagram of the complete cell, (blue points corresponding to apical and red points to the exemplar basal projection, black points for the remaining projections). F. The resulting persistence image for all mouse/rat pyramidal cells.

2.4 Topological descriptors

Another way of characterising cells is to look at their topology, which can provide a valuable means of characterising their complex branching structures. One such characterisation is the Topological Morphology Descriptor (TMD) [57]. The TMD returns the topological persistence barcode of the given structure. This barcode concisely encodes the branching structure of the cell and has been shown to be highly effective in the generation of synthetic neurons [58]. Here, we apply the TMD to characterize the cellular projections and compare cellular topology across cell-types and species.

Topological persistence analyses at what length scales a given topological feature, here the path length of connected branches, persists. The TMD is computed by tracking the initiation points and termination points, with respect to path length from the soma, of the connected branches at different length scales [59], preserving the longer components and filtering out the shorter ones. This is computed over the individual neurite projections, rather than the entire cell.

The terminal branches are evaluated with respect to their path length from the soma. Branches that share a parent node are compared and the branch

with the shorter path length is filtered and its initiation length (path length of the parent node) and termination length are recorded. The longer branch is preserved and the process is repeated for all remaining branches, until there is only one remaining branch that corresponds to the path from the soma to the terminal point with the greatest path length. The returned barcode is a multi set of pairs of numbers, describing the initiation and termination of each constituent bar in the neurite, with respect to the path distance from the soma. The topological barcode has been shown to retain detailed information of the structure it was computed for and can be used to categorise and identify cell types [58].

In this work the cells were first decomposed into their principal projections, by identifying the projections originating from the soma, and persistence barcodes from these projections were identified. The identified barcodes were used to create persistence images for each cell type (Fig.5). These images visualize the persistence of components based on their initiation and termination lengths. To enhance smoothness, kernel density estimation was applied, and the images were normalized so that the sum of their entries equaled one

Comparisons were made between persistence images by computing the global topological distance, D , between images,

$$D = \frac{\sum_{i=1}^n |Im1_i - Im2_i|}{n}$$

Where n is the number of voxels. This topological distance was calculated for cell types within species and also cell types between species.

3 Results

3.1 Morphological features' reference values

A summary of the typical values of structure features is given in Tab.2. Some features show little variation between cell type and species ($\langle R_{branch} \rangle_s$, CV_{branch} , $\langle \mu OD_{branch} \rangle_s$, and τ_{branch}). The remaining features displayed a wide range of values, suggesting higher inter-cellular and intra-species variability.

3.2 Comparing neuronal and glial cells

Given the growing interest in differentiating neurons and glial cells, the mean values of the structural features were computed across only neuronal and glial cells and reported in Tab.3. The findings show, compared to glial cells, neurons had larger soma and cell domain and longer branches; and reduced branching, branch curvedness, number of primary projections, and proportion of soma surface covered by projections; the remaining features displayed similar values.

to
 a good idea
 D
 top
 different.
 range for
 on different.
 in which range
 similar
 all cells are
 specify
 and define
 probably the dynamic
 range in which
 similar cells are

typo.
 D per cell type
 for example:
 D purkinje, astrocyte (mouse)

3.3 Correlations between Morphological features

The correlations between structural features are illustrated in supplementary Fig.10 . Some consistent and expected patterns are observed, particularly involving surface-to-volume ratio measures: the soma surface-to-volume ratio is obviously negatively correlated with soma radius, and the branch surface-to-volume ratio is obviously negatively correlated with branch radius. Furthermore, the surface-to-volume ratio of both the soma and branches are obviously positively correlated with the surface-to-volume ratio of the domain. Additional, micro-orientation dispersion shows a positive correlation with branch tortuosity, τ , across majority cell types. No further consistent correlations between features were observed across cell types.

3.4 Cellular shape reference values

A summary of the shape descriptors typical values for each cell type and species is reported in Tab.4. Cells with highly oriented and polarized projections, such as Purkinje cells, granule cells and pyramidal neurons have high FA and low orientation dispersion; while most of the glial cells projections are highly dispersed and with low FA.

Cell Type	Species	N	R _{main} (μm)	N _{prof}	BO	SV _{main}	R _{norm} (μm)	Thinnest (%)	SV _{norm} (μm ⁻¹)	SV _{equiv} (μm ⁻¹)	L _{1-branch} (μm)	R _{branch} (μm)	CV _{branch}	μOD _{branch}	R _c (μm)	Thinnest
Microglia	mouse/rat	169(123)	62 ± 70	9 ± 3	9 ± 4	2.5 ± 1.1	≥ 6.1 ± 6.3	≤ 4.7 ± 2.9	0.52 ± 0.38	4.3 ± 1.6	20 ± 13	≥ 1.1 ± 1.3	0.95	0.26 ± 0.05	≥ 34 ± 31	0.89 ± 0.03
	monkey	61(60)	42 ± 8	8 ± 2	10 ± 4	5.2 ± 1.1	2.73 ± 0.50	≤ 1.4 ± 1.1	1.0 ± 0.25	16 ± 2	7.2 ± 5.7	≥ 0.15 ± 0.08	1.2	0.30 ± 0.04	0.91 ± 0.02	0.91 ± 0.02
	human	n.a.	n.a.	n.a.	n.a.	n.a.	n.a.	n.a.	n.a.	n.a.	n.a.	n.a.	n.a.	n.a.	n.a.	n.a.
Astrocyte	mouse/rat	269(247)	40 ± 14	11 ± 7	19 ± 8	2.3 ± 0.65	≥ 2.1 ± 0.96	≤ 86 ± 26	0.88 ± 0.37	6.3 ± 2.2	4.9 ± 2.6	≥ 1.2 ± 0.5	0.55	0.26 ± 0.05	≥ 3.3 ± 1.2	0.88 ± 0.03
	monkey	n.a.	n.a.	n.a.	n.a.	n.a.	n.a.	n.a.	n.a.	n.a.	n.a.	n.a.	n.a.	n.a.	n.a.	n.a.
	human	n.a.	n.a.	n.a.	n.a.	n.a.	n.a.	n.a.	n.a.	n.a.	n.a.	n.a.	n.a.	n.a.	n.a.	n.a.
Oligodendrocyte	mouse/rat	80(80)	71 ± 19	15 ± 5	10 ± 3	0.67 ± 0.24	≥ 8.9 ± 1.7	≤ 61 ± 2.0	0.35 ± 0.06	13 ± 0.1	12.4 ± 3.4	≥ 0.19 ± 0.0	0.62	0.16 ± 0.02	10 ± 1.1	0.81 ± 0.05
	monkey	n.a.	n.a.	n.a.	n.a.	n.a.	n.a.	n.a.	n.a.	n.a.	n.a.	n.a.	n.a.	n.a.	n.a.	n.a.
	human	n.a.	n.a.	n.a.	n.a.	n.a.	n.a.	n.a.	n.a.	n.a.	n.a.	n.a.	n.a.	n.a.	n.a.	n.a.
Pyramidal	mouse/rat	351(330)	427 ± 350	5 ± 3	9 ± 4	3.3 ± 2.1	≥ 7.1 ± 3.6	≤ 18 ± 30	0.38 ± 0.18	12 ± 13	64 ± 30	> 0.5 ± 0.6	1.33	0.26 ± 0.12	≥ 97 ± 173	0.86 ± 0.09
	monkey	867(836)	480 ± 248	6 ± 4	8 ± 3	2.7 ± 1.0	≥ 6.2 ± 3.7	≤ 24 ± 39	0.39 ± 0.17	5.7 ± 3.0	99 ± 75	> 0.55 ± 0.65	1	0.23 ± 0.09	≥ 65 ± 62	0.89 ± 0.06
	human	1076(1070)	287 ± 95	6 ± 2	7 ± 2	1.9 ± 1.4	≥ 10 ± 3.6	≤ 7.2 ± 5.0	0.27 ± 0.09	3.8 ± 2.4	70 ± 21	≥ 0.75 ± 1.1	0.95	0.29 ± 0.06	≥ 43 ± 21	0.90 ± 0.05
Granule	mouse/rat	128(124)	358 ± 124	2 ± 1	5 ± 2	3.6 ± 1.0	≥ 4.1 ± 0.8	≤ 37 ± 19	0.68 ± 0.17	6.7 ± 2.6	169 ± 31	≥ 0.40 ± 0.025	0.51	0.30 ± 0.05	≥ 16 ± 9	0.86 ± 0.04
	monkey	n.a.	n.a.	n.a.	n.a.	n.a.	n.a.	n.a.	n.a.	n.a.	n.a.	n.a.	n.a.	n.a.	n.a.	n.a.
	human	77(77)	538 ± 110	5 ± 1	5 ± 1	2.8 ± 1.0	≥ 5.3 ± 2.7	≤ 77 ± 20	0.76 ± 0.38	3.6 ± 0.9	122 ± 27	≥ 0.5 ± 0.1	0.24	0.20 ± 0.04	≥ 17 ± 8	0.84 ± 0.05
Purkinje	mouse/rat	121(120)	158 ± 37	2 ± 1	10 ± 2	1.7 ± 0.49	≥ 7.6 ± 1.3	≤ 24 ± 18	0.36 ± 0.07	4.7 ± 2.2	8.6 ± 1.5	≥ 0.60 ± 0.35	0.43	0.27 ± 0.05	≥ 21 ± 9	0.95 ± 0.01
	monkey	n.a.	n.a.	n.a.	n.a.	n.a.	n.a.	n.a.	n.a.	n.a.	n.a.	n.a.	n.a.	n.a.	n.a.	n.a.
	human	n.a.	n.a.	n.a.	n.a.	n.a.	n.a.	n.a.	n.a.	n.a.	n.a.	n.a.	n.a.	n.a.	n.a.	n.a.
Basket	mouse/rat	16(13)	440 ± 242	7 ± 3	10 ± 6	2.5 ± 0.65	≥ 6.3 ± 3.0	≤ 35 ± 37	0.43 ± 0.15	15 ± 11	57 ± 36	≥ 0.95 ± 1.6	1.2	0.30 ± 0.12	≥ 11 ± 4	0.88 ± 0.06
	monkey	20(18)	671 ± 302	9 ± 6	19 ± 7	2.5 ± 0.97	≥ 5.9 ± 2.0	≤ 3.7 ± 6.5	0.50 ± 0.18	12 ± 5	64 ± 23	≥ 0.35 ± 0.24	0.31	0.35 ± 0.06	≥ 14 ± 6	0.79 ± 0.06
	human	25(21)	300 ± 344	5 ± 3	7 ± 6	3.2 ± 0.76	≥ 4.75 ± 0.65	≤ 24 ± 23	0.60 ± 0.12	7.7 ± 8.9	43 ± 39	≥ 1.3 ± 0.5	0.86	0.24 ± 0.13	≥ 8.2 ± 5.0	0.90 ± 0.08
Gabaergic	mouse/rat	188(167)	284 ± 186	6 ± 2	11 ± 9	1.3 ± 0.29	≥ 7.6 ± 3.4	≤ 24 ± 24	0.25 ± 0.05	6.5 ± 2.6	57 ± 38	≥ 2.0 ± 1.6	0.08	30 ± 0.07	≥ 47 ± 61	0.87 ± 0.04
	monkey	n.a.	n.a.	n.a.	n.a.	n.a.	n.a.	n.a.	n.a.	n.a.	n.a.	n.a.	n.a.	n.a.	n.a.	n.a.
	human	8(8)	350 ± 186	6 ± 3	11 ± 4	4.3 ± 0.64	≥ 4.3 ± 0.3	≤ 1.0 ± 0.3	0.65 ± 0.09	13.6 ± 2.7	52 ± 18	≥ 0.29 ± 0.13	1.7	0.36 ± 0.07	≥ 12 ± 3	0.79 ± 0.05
Glutamatergic	mouse/rat	36(35)	756 ± 675	6 ± 2	11 ± 6	1.7 ± 0.38	≥ 8.3 ± 2.6	≤ 6 ± 16	0.33 ± 0.05	5.4 ± 1.2	46 ± 17	≥ 0.8 ± 0.6	0.53	0.36 ± 0.06	≥ 16 ± 7	0.85 ± 0.03
	monkey	n.a.	n.a.	n.a.	n.a.	n.a.	n.a.	n.a.	n.a.	n.a.	n.a.	n.a.	n.a.	n.a.	n.a.	n.a.
	human	106(105)	616 ± 111	4 ± 2	6 ± 2	0.84 ± 1.7	≥ 29 ± 10	≤ 0.4 ± 3.3	0.10 ± 0.03	4.0 ± 0.1	145 ± 56	≥ 0.95 ± 0.75	5.8	0.11 ± 0.03	≥ 475 ± 84	0.95 ± 0.02

Table 2: Summary of the morphological features computed for each species and cell type. N is the number of cellular structures investigated with complete information about the neurite structure; same information for soma in brackets. The reported values are mean ± s.d. over the corresponding sample. The '≥' and '≤' are used when the estimated value of the corresponding feature may be slightly (on average < 20%) under- or over-estimated, respectively, given the known limitations of the approach used. n.a. = not available.

Morphological Feature	Value Range	Value Mean	
		All Cell Types	Only Neurons
R_{soma} (μm)	$\geq 2 - 29$	≥ 7	≥ 8
$\langle R_{branch} \rangle_s$ (μm)	$\geq 0.15 - 2$	≥ 0.8	≥ 0.8
CV_{branch}	$0.3 - 5.8$	1.2	1
L_{branch} (μm)	$5 - 145$	54	64
S/V_{branch} (μm^{-1})	$4 - 16$	8	8
$\langle \mu OD_{branch} \rangle_s$	$0.1 - 0.5$	0.3	0.3
$\langle R_c \rangle_s$ (μm)	$\geq 4 - 475$	≥ 55	≥ 64
τ_{branch}	$0.80 - 0.95$	0.88	0.90
BO	$4 - 19$	9	9
N_{proj}	$2 - 11$	6	5
η_{soma} (%)	$\leq 1 - 86$	≤ 18	≤ 14
R_{domain} (μm)	$40 - 756$	334	400
			48
			≥ 4.0
			≥ 0.8
			1
			11
			8
			0.3
			≥ 18
			0.92
			13
			9
			≤ 31

Table 3: Reference values for all the morphological features of neuronal and glial cells. The ranges and mean values obtained from the whole dataset investigated are reported, together with mean values for only neurons and glia. The ' \geq ' and ' \leq ' are used when the estimated value of the corresponding feature may be slightly (on average $< 20\%$) under- or over-estimated, respectively, given the known limitations of the approach used.

Cell Type	Species	Shape		
		Watson parameter	Orientation dispersion	Fractional Anisotropy
Microglia	Mouse/Rat	0.99	0.50	0.21 ± 0.11
	Monkey	1.7	0.34	0.21 ± 0.11
	Human	n.a.	n.a.	n.a.
Astrocyte	Mouse/Rat	2.1	0.28	0.34 ± 0.21
	Monkey	n.a.	n.a.	n.a.
	Human	n.a.	n.a.	n.a.
Oligodendrocyte	Mouse/Rat	2.1	0.28	0.24 ± 0.17
	Monkey	n.a.	n.a.	n.a.
	Human	n.a.	n.a.	n.a.
Pyramidal	Mouse/Rat	1.9	0.31	0.33 ± 0.19
	Monkey	2.0	0.30	0.39 ± 0.25
	Human	0.67	0.62	0.45 ± 0.18
Granule	Mouse/Rat	5.8	0.11	0.75 ± 0.16
	Monkey	n.a.	n.a.	n.a.
	Human	3.4	0.18	0.70 ± 0.15
Purkinje	Mouse/Rat	5.3	0.12	0.74 ± 0.08
	Monkey	n.a.	n.a.	n.a.
	Human	n.a.	n.a.	n.a.
Basket	Mouse/Rat	2.1	0.28	0.25 ± 0.19
	Monkey	0.89	0.54	0.39 ± 0.28
	Human	0.98	0.51	0.21 ± 0.21
Gabaergic	Mouse/Rat	1.1	0.47	0.31 ± 0.16
	Monkey	n.a.	n.a.	n.a.
	Human	2.4	0.25	0.25 ± 0.14
Glutamatergic	Mouse/Rat	1.0	0.50	0.24 ± 0.13
	Monkey	n.a.	n.a.	n.a.
	Human	2.7	0.23	0.74 ± 0.04

Table 4: Summary statistics of shape descriptors for each cell type and species. n.a. = not available.

Figure 7: **Persistence maps for each cell types and all species together.** The persistence map shows at what length scales a given topological feature, here the path length of connected branches, persists. It is computed by tracking the initiation points and termination points, with respect to path length from the soma, of the connected branches at different length scales.

20-40 ms [60, 31, 63], suggesting membrane permeability $2\text{-}35 \mu\text{m/s}$. Using the latter estimates of membrane permeability, and our estimates of the cellular and branch surface-to-volume ratios we estimated distribution of intra-domain and intra-branch residence times that span from a few milliseconds to tens of milliseconds (Fig.6). On average, glial cells have shorter residence times than neurons, and small neurons like granule cells have much shorter residence times than large neurons like pyramidal cells. These observations support the idea of a wide distribution of exchange times within the GM, with water within glial cells in faster exchange with ECS than that within neurons (10-20 ms versus 20-50

Figure 8: Local topological distance between images (top right of matrices) and global distance between images (bottom left) for rodent, human, hominid inter species comparison, and a comparison between rodent and hominid cell types

ms). One potential limitation of our analysis is that we did not include spines, boutons and glial feet, that can occupy up to 20% of the GM volume [64]. Recent studies have demonstrated that diffusion-mediated exchange between the shaft of cellular projections and these small lateral protrusions is another mechanism of water exchange, different from the permeative one, but happening on similar time scales (from a few milliseconds to a few tens of milliseconds) [65, 66, 67]. Therefore, the interplay between diffusion-mediated and permeative exchange cannot be neglected when analysing and interpreting time-dependent dMRI measurements in the GM. Appropriate modelling and acquisition strategies to disentangle the two mechanisms and provide unbiased estimates are still missing, and represent an exciting avenue for future research in the field.

In general, exchange cannot be completely ignored, nor does exchange fully

Figure 6: Probability density functions of soma radius, branch length, intra-branch residence time and intra-cellular residence time for each cell types and all the species. The residence times were calculated from the surface to volume ratios with a permeability of $20 \mu\text{m/s}$ [60] and diffusivity typical of water.

3.5 Distribution of some structural features of interest for dMRI modelling

The full distributions of values for some structural features of interest, obtained by merging together all the estimated values from all the species for each cell type, are shown in Fig.6. Given the increasing interest of the diffusion-based microstructural imaging community in estimating glia microstructure and the striking difference between the neuronal components of cerebral and cerebellar cortices (e.g., Purkinje cells only in cerebellum), we decided to group them into three classes: glia, cortical neurons and cerebellar neurons, and use three different colors to simplify the visualization of the results. The distributions for all the morphological features are reported in supplementary Fig.??.

actually this is more difficult to see. I would suggest to "merge" the histograms of each class in one pdf.

3.6 Topological distance between cell types

The persistence maps of all the cell types for all species together are shown in Fig.7. These persistence maps are used for each species and each cell type to estimate cell-type specific topological distances, shown in Fig.8.

In rodent cells, there is a notably high global topological distance between glial cells and neurons, indicating a fundamentally different topological organisation between them. This difference in topology becomes even more evident when considering the comparison along the same length scale. Since glial cells are significantly smaller in size, they inherently display a much smaller topological persistence. This size-related constraint underscores their distinct spatial and structural properties relative to neurons.

Within rodent neurons, granule cells stand out by exhibiting a higher topological distance compared to other neuronal types. This disparity is likely related to their characteristically low branch order, which reflects their simpler dendritic structures and reduced connectivity compared to more complex neurons.

In human cells, however, no significant or consistent trends in topological distance are observed, suggesting greater variability or less pronounced differences

→ $\sim 4-5 \cdot 10^{-5}$ from Figure 8.

This number scale is weird without the proper context as mentioned in methods.

number?

i.e. in the topographic map, they are clustered near the origin.

even though, would not be the same i.e. small topological distance, when the density lies on the identity axis?

However, two correlations are observed to have high topological distance (gabaergic-glutamatergic / gabaergic-granular) is this due to random or has a specific reason (biological?).

in cellular topology among the various cell types.

When examining hominid cells, which include both monkey and human samples, microglia demonstrate a higher topological distance compared to neurons. This observation reinforces the trend seen in rodent cells, further supporting the notion that glial cells, maintain a distinct and conserved topological profile across different species.

In the cross-species comparison between rodent and hominid cells, microglia consistently show a high topological distance relative to neurons, mirroring the trend observed within each species. Additionally, the comparatively low topological distance for microglia across species suggest that their topological properties are consistent, pointing to a shared structural and functional organization in microglia across species.

4 Discussion

4.1 A Quantitative View of Gray Matter Microstructure

GM Intra-Cellular Space. There is currently a lack of in-depth morphological analysis of brain-cell structures of relevance for modelling water diffusion in the GM intra-cellular space. In this work we propose a first analysis with the aim to fill this gap. We estimated a comprehensive set of morphological features useful to GM microstructure modelling from reconstructions of microscopy data from three species. We estimated that neural soma size ranges from 2 to 30 μm in radius with an average of 7 μm and surface-to-volume ratio S/V $0.5 \mu\text{m}^{-1}$. Neurons have on average soma twice as big as glial cells, fewer projections radiating from the soma and less projection coverage of soma surface. The radius of cellular projections ranges from 0.3 to 4 μm with average value 1.6 μm and S/V $8 \mu\text{m}^{-1}$, similar between neurons and glia. Cellular projections' microscopic orientation dispersion, as defined in [46], is 0.10-0.50, with average value 0.30, similar between neurons and glia; curvature radius is 4-475 μm , with average value of 55 μm . On average, neuronal projections have a curvature radius 6 times larger than glial projections. The branching order of neural cell is 4-20, with average value of 9. Glial cells have branching order 1.5 times larger than neurons. The tortuosity of the branch of neural cell projections is 1.05-1.27, with average value of 1.14, similar between neurons and glial cells. The projections of neural cells extend to distances of 40-756 μm , with neurons on average 400 μm and glial cells on average 50 μm . For completeness, we also report on the relevant features of the GM extra-cellular space and neural cell membrane permeability to water from the literature.

Neural Cell Membrane Permeability to Water. Several works suggest water exchange between unmyelinated neurites and ECS and/or soma occurring on time scales comparable to typical dMRI clinical acquisitions, i.e. 10-100 ms. Although there is not a consensus yet, some works report exchange times t_{ex} for ex vivo mouse brain 5-10 ms [61, 62, 32], suggesting membrane permeability 125 $\mu\text{m/s}$ (like red blood cells), others report t_{ex} for in vivo mouse brain of

An important point about this is that the estimated average and sd (which was not included) are biased if the distribution is not Gaussian/Normal. Based on the Sup. Fig 1; seems that in the case \rightarrow this would be a problem when compared with dMRI. but shown in Table 3.

475 μm ?
or 475 μm ?

18
even though this is true, this ~~also~~ could also depend on the specimen fixation procedure and post-mortem interval.

average out the restriction effects of cell membranes, supporting including both neurite exchange and a soma compartment in a parsimonious biophysical model of diffusion in GM, as proposed by Olesen et al. [32] with the SANDIX model. There are however some caveats to bear in mind when interpreting the model estimates: (a) the soma apparent MR radius is unavoidably an overestimation of the true soma radius; (b) the compartmental signal fractions reflect, apart from volume, also unknown T2 and T1 weighting, which might differ among the compartments; (c) fast exchange across the neurite membrane's could be problematic for models based on the Kärger's model because it could violate the model's assumptions (i.e. barrier-limited diffusion: $t_{ex} \gg d^2/D$, where d is the neurite diameter; and $\delta \ll t_{ex}$) and lead to biased estimates; (d) molecular diffusion in structural features such as dendritic spines and glial leaflets can lead to diffusion-mediated exchange mechanisms that occur on the same time scales of permeative exchange, making the interpretation of exchange measurements through MRI solely due to permeative exchange fundamentally wrong; (e) using single diffusion encoding acquisitions it is impossible to disentangle restriction and exchange; more refined acquisitions could allow the separation and more accurate quantification of these two effects [62, 68]. The estimates of the model parameters should therefore only be taken as an indication of the true tissue features and validation against realistic numerical simulations and/or alternative measurements in controlled phantoms (e.g. biomimetic tissues and brain organoids) and/or post-mortem samples (e.g., optical, confocal or electron microscopy) remains essential.

typo

are you referring for both in vivo and ex vivo?

4.2 General Implications for Biophysical Modelling

Here we provide a few illustrative examples demonstrating how our results can inform biophysical modelling. Assuming an intra-cellular diffusivity $D=2 \mu\text{m}^2/\text{ms}$ representative of water and $0.40 \mu\text{m}^2/\text{ms}$ representative of intracellular brain metabolites, and considering a typical single diffusion encoding acquisition with gradient pulse duration $\delta < 30 \text{ ms}$, gradient pulse separation $\Delta < 70 \text{ ms}$ and diffusion time $td < 60 \text{ ms}$, we can infer from Tab.3 that:

- **Soma restriction can be significant for both water and metabolites:** given the low η_{soma} , the soma can significantly restrict diffusion when (from [69]) $5Dt_d \geq R_{soma}^2$, that is for $t_d \geq 5 \text{ ms}$ for water and $t_d \geq 25 \text{ ms}$ for metabolites, given a soma radius $R_{soma} \approx 7 \mu\text{m}$. This is supported by previous findings which demonstrate the impact of soma contribution on the final dMRI signal [30, 44]
- **Cellular domain restriction is generally negligible for both water and metabolites:** given the size of the cellular domain, it will only significantly restrict molecular diffusion when $5Dt_d \geq R_{domain}^2$, that is for $t_d \geq 160 \text{ ms}$ for water and $t_d \geq 800 \text{ ms}$ for metabolites, given $R_{domain} \geq 40 \mu\text{m}$, which is far longer than typically used diffusion times t_d .

I assume also that this is in clinical scanners mainly.

I don't get the difference between these two. Cellular domain you refer to soma + projections? or total YCE?

this is in-vivo conditions y assume...

which, can be the results of these study measurements still be used since histological results suffer of apoptosis, shrinkage & deformation and cannot be compared with in vivo?

I can see that could be used as a benchmark but the discussion and interpretation below can be biased.

When you mention significance, are you referring that the difference can be significant? If so, then I would need a significant test.

- **Impact of projections curvedness is generally negligible for both water and metabolites:** the impact of curvedness can only be significant when (from [70], $2D\Delta, 2D\delta \geq (<R_c>_s)^2$), that is for $\Delta, \delta \geq 750$ ms for water and $\Delta, \delta \geq 3750$ ms for metabolites, given $<R_c>_s \approx 55$ μm , which is far longer than the values typically. This is further supported by [44].
- **Impact of projection undulation can be significant for both water and metabolites:** the values of $<\mu OD_{branch}>_s$ estimated from the real cellular data match those simulated in [46], from which it can be concluded that undulation can bias the estimation of projection radius in grey matter. However, the average branch radius $<R_{branch}>_s \approx 0.8$ μm is far below the resolution limit of conventional water dMRI techniques [71].
- **Impact of branching is generally negligible for both water and metabolites:** given the branch length $L_{branch} \approx 54$ μm , the exchange between branches is negligible for $t_d \ll L_{branch}^2/(2D) \approx 750$ ms for water and $t_d \ll 3750$ ms for metabolites, significantly longer than typical t_d used. Further supported by [42, 44].
- **Impact of membrane permeability of cellular projections can be significant for water.** The average intra-branch residence times given the estimated S/V_{branch} are ≈ 20 ms for neurons and ≈ 10 ms for glial cells. The permeative exchange is thus negligible only for $t_d < 20$ ms for neuronal projections and for $t_d < 10$ ms for glial projections; which is not satisfied in conventional water dMRI applications. Further supported by [31, 44, 63].
- **Impact of projections orientation dispersion can be significant.** Cells with highly oriented and polarized projections, such as Purkinje cells, granule cells and pyramidal neurons have high FA (> 0.50) and low orientation dispersion (< 0.25); while the projections of most of the glial cells and other neuronal cells have FA (< 0.50) and high orientation dispersion (> 0.25). This supports growing evidence that FA, e.g., from DTI, and orientation dispersion estimates, e.g. from NODDI, can discriminate different cytoarchitectural domains (or layers) in hippocampus, cortex and cerebellum [72, 73, 74, 75, 76].
- **Microglia topology is similar across different species,** suggesting that a single biophysical model of diffusion in microglia would likely hold across species. In contrast, all the other cell-types likely need a dedicated model that accounts for the species-specific topological differences. Exception may be granule cells, that show small differences across species as well.

← typically reported, measured...?

What about in Connectome I and II scanners? That could be reached.

That is not true for humans as shown in Figure 8. Or did I miss something?

4.3 Limitations

Our investigation, while thorough, is not exhaustive (nor could it be). Other features could certainly be measured and tabulated, such as spine density. However, we focus here on those deemed relevant for biophysical modeling in the current literature on microstructure imaging via dMRI. We will release our code openly and freely to allow future work to complement this study with additional features and information as needed. Our focus is on a selected set of real three-dimensional reconstructions, as we aim to characterize cellular morphology in the healthy brain. Future studies can use this code to incorporate more and improved reconstructions of healthy brain tissue as they become available (e.g., through updates to NeuroMorpho or large electron microscopy studies [77, 78]). We provide a few illustrative examples to demonstrate how this study can inform biophysical modeling of dMR signals. However, the information obtained from our investigation has broader applications, and we hope the scientific community will find it valuable for advancing our understanding of gray matter microstructure as a whole. Throughout the manuscript, we discussed the limitations of the morphometric approach used (e.g., an expected underestimation of $< 20\%$ for soma volume and branch radius). Within the constraints of currently available tools, we provide reference values that were previously unavailable. We have taken great care to report error estimates and uncertainties for all measurements to account for these limitations. Future work is needed to improve morphometric algorithms, but this is beyond the scope of the current study.

5 Conclusion

This work provides quantitative information on brain cell structures essential to design sensible biophysical models of the dMRI signal in gray matter. Reporting typical values of relevant features of brain cell morphologies, this study represents a valuable guidebook for the microstructure imaging community and provides illustrative examples demonstrating how to inform biophysical modelling.

Acknowledgments

This work, CAR and MP are supported by the UKRI Future Leaders Fellowship MR/T020296/2. DKJ was supported by a Wellcome Trust Strategic Award (104943/Z/14/Z) and Wellcome Discovery Award (227882/Z/23/Z).

References

- [1] Constantino Sotelo. Viewing the brain through the master hand of ramón y cajal. *Nature Reviews Neuroscience*, 4(1):71–77, 2003.

- [2] Hongkui Zeng and Joshua R Sanes. Neuronal cell-type classification: challenges, opportunities and the path forward. *Nature Reviews Neuroscience*, 18(9):530–546, 2017.
- [3] Nathan W Gouwens, Staci A Sorensen, Jim Berg, Changkyu Lee, Tim Jarsky, Jonathan Ting, Susan M Sunkin, David Feng, Costas Anastassiou, Eliza Barkan, et al. Classification of electrophysiological and morphological types in mouse visual cortex. *BioRxiv*, page 368456, 2018.
- [4] Linda J Lawson, Victor Hugh Perry, Pietro Dri, and Siamon Gordon. Heterogeneity in the distribution and morphology of microglia in the normal adult mouse brain. *Neuroscience*, 39(1):151–170, 1990.
- [5] Yun-Long Tan, Yi Yuan, and Li Tian. Microglial regional heterogeneity and its role in the brain. *Molecular psychiatry*, 25(2):351–367, 2020.
- [6] Christopher S. Von Bartheld, Jami Bahney, and Suzana Herculano-Houzel. The search for true numbers of neurons and glial cells in the human brain: A review of 150 years of cell counting. *Journal of Comparative Neurology*, 524:3865–3895, December 2016.
- [7] William Bondareff and Joseph J. Pysh. Distribution of the extracellular space during postnatal maturation of rat cerebral cortex. *The Anatomical Record*, 160:773–780, April 1968.
- [8] Muhammad A. Spocter, William D. Hopkins, Sarah K. Barks, Serena Bianchi, Abigail E. Hehmeyer, Sarah M. Anderson, Cheryl D. Stimpson, Archibald J. Fobbs, Patrick R. Hof, and Chet C. Sherwood. Neuropil distribution in the cerebral cortex differs between humans and chimpanzees. *Journal of Comparative Neurology*, 520:2917–2929, September 2012.
- [9] Alessandro Motta, Manuel Berning, Kevin M. Boergens, Benedikt Staffler, Marcel Beining, Sahil Loomba, Philipp Hennig, Heiko Wissler, and Moritz Helmstaedter. Dense connectomic reconstruction in layer 4 of the somatosensory cortex. *Science*, 366:eaay3134, November 2019.
- [10] D.P. Pelvig, H. Pakkenberg, A.K. Stark, and B. Pakkenberg. Neocortical glial cell numbers in human brains. *Neurobiology of Aging*, 29:1754–1762, November 2008.
- [11] Lisette Salvesen, Kristian Winge, Tomasz Brudek, Tina Klitmøller Agander, Annemette Løkkegaard, and Bente Pakkenberg. Neocortical Neuronal Loss in Patients with Multiple System Atrophy: A Stereological Study. *Cerebral Cortex*, page bhv228, October 2015.
- [12] Eva Syková and Charles Nicholson. Diffusion in Brain Extracellular Space. *Physiological Reviews*, 88:1277–1340, October 2008.

- [13] Takuma Mabuchi, Jacinta Lucero, Anne Feng, James A Koziol, and Gregory J Del Zoppo. Focal Cerebral Ischemia Preferentially Affects Neurons Distant from Their Neighboring Microvessels. *Journal of Cerebral Blood Flow & Metabolism*, 25:257–266, February 2005.
- [14] Marijke AM Lemmens, Harry WM Steinbusch, Bart PF Rutten, and Christoph Schmitz. Advanced microscopy techniques for quantitative analysis in neuromorphology and neuropathology research: current status and requirements for the future. *Journal of Chemical Neuroanatomy*, 40(3):199–209, 2010.
- [15] Kevin L Briggman and Davi D Bock. Volume electron microscopy for neuronal circuit reconstruction. *Current opinion in neurobiology*, 22(1):154–161, 2012.
- [16] Jack Reddaway, Peter Eulalio Richardson, Ryan J Bevan, Jessica Stone-
man, and Marco Palombo. Microglial morphometric analysis: so many options, so little consistency. *Frontiers in neuroinformatics*, 17:1211188, 2023.
- [17] Ruchi Parekh and Giorgio A Ascoli. Neuronal morphology goes digital: a research hub for cellular and system neuroscience. *Neuron*, 77(6):1017–1038, 2013.
- [18] Eugene Lin and Adam Alessio. What are the basic concepts of temporal, contrast, and spatial resolution in cardiac ct? *Journal of cardiovascular computed tomography*, 3(6):403–408, 2009.
- [19] JJ Rodriguez, M Olabarria, A Chvatal, and A Verkhatsky. Astroglia in dementia and alzheimer’s disease. *Cell Death & Differentiation*, 16(3):378–385, 2009.
- [20] Ennio Pannese. Morphological changes in nerve cells during normal aging. *Brain Structure and Function*, 216(2):85–89, 2011.
- [21] Daniel C Alexander, Tim B Dyrby, Markus Nilsson, and Hui Zhang. Imaging brain microstructure with diffusion mri: practicality and applications. *NMR in Biomedicine*, 32(4):e3841, 2019.
- [22] Hui Zhang, Torben Schneider, Claudia A Wheeler-Kingshott, and Daniel C Alexander. Noddi: practical in vivo neurite orientation dispersion and density imaging of the human brain. *Neuroimage*, 61(4):1000–1016, 2012.
- [23] Els Fieremans, Jens H Jensen, and Joseph A Helpert. White matter characterization with diffusional kurtosis imaging. *Neuroimage*, 58(1):177–188, 2011.
- [24] Daniel C Alexander, Penny L Hubbard, Matt G Hall, Elizabeth A Moore, Maurice Ptito, Geoff JM Parker, and Tim B Dyrby. Orientationally invariant indices of axon diameter and density from diffusion mri. *Neuroimage*, 52(4):1374–1389, 2010.

- [25] Jelle Veraart, Daniel Nunes, Umesh Rudrapatna, Els Fieremans, Derek K Jones, Dmitry S Novikov, and Noam Shemesh. Noninvasive quantification of axon radii using diffusion mri. *elife*, 9:e49855, 2020.
- [26] Sune N Jespersen, Christopher D Kroenke, Leif Østergaard, Joseph JH Ackerman, and Dmitriy A Yablonskiy. Modeling dendrite density from magnetic resonance diffusion measurements. *Neuroimage*, 34(4):1473–1486, 2007.
- [27] M.E. Komlosh, F. Horkay, R.Z. Freidlin, U. Nevo, Y. Assaf, and P.J. Basser. Detection of microscopic anisotropy in gray matter and in a novel tissue phantom using double Pulsed Gradient Spin Echo MR. *Journal of Magnetic Resonance*, 189:38–45, November 2007.
- [28] Noam Shemesh, Evren Özarslan, Peter J. Basser, and Yoram Cohen. Accurate noninvasive measurement of cell size and compartment shape anisotropy in yeast cells using double-pulsed field gradient MR. *NMR in Biomedicine*, 25:236–246, February 2012.
- [29] Trong-Kha Truong, Arnaud Guidon, and Allen W. Song. Cortical Depth Dependence of the Diffusion Anisotropy in the Human Cortical Gray Matter In Vivo. *PLoS ONE*, 9:e91424, March 2014.
- [30] Marco Palombo, Andrada Ianus, Michele Guerreri, Daniel Nunes, Daniel C Alexander, Noam Shemesh, and Hui Zhang. Sandi: a compartment-based model for non-invasive apparent soma and neurite imaging by diffusion mri. *Neuroimage*, 215:116835, 2020.
- [31] Ileana O Jelescu, Alexandre de Skowronski, Françoise Geffroy, Marco Palombo, and Dmitry S Novikov. Neurite exchange imaging (nexi): A minimal model of diffusion in gray matter with inter-compartment water exchange. *NeuroImage*, 256:119277, 2022.
- [32] Jonas L Olesen, Leif Østergaard, Noam Shemesh, and Sune N Jespersen. Diffusion time dependence, power-law scaling, and exchange in gray matter. *NeuroImage*, 251:118976, 2022.
- [33] Yaniv Assaf. Imaging laminar structures in the gray matter with diffusion mri. *Neuroimage*, 197:677–688, 2019.
- [34] Ileana O Jelescu and Matthew D Budde. Design and validation of diffusion mri models of white matter. *Frontiers in physics*, 5:61, 2017.
- [35] Dmitry S Novikov, Els Fieremans, Sune N Jespersen, and Valerij G Kiselev. Quantifying brain microstructure with diffusion mri: Theory and parameter estimation. *NMR in Biomedicine*, 32(4):e3998, 2019.
- [36] Ileana O Jelescu, Marco Palombo, Francesca Bagnato, and Kurt G Schilling. Challenges for biophysical modeling of microstructure. *Journal of Neuroscience Methods*, 344:108861, 2020.

- [37] Giorgio A Ascoli, Duncan E Donohue, and Maryam Halavi. Neuromorpho.org: a central resource for neuronal morphologies. *Journal of Neuroscience*, 27(35):9247–9251, 2007.
- [38] Hermann Cuntz, Friedrich Forstner, Alexander Borst, and Michael Häusser. The trees toolbox—probing the basis of axonal and dendritic branching, 2011.
- [39] Blender Online Community. *Blender - a 3D modelling and rendering package*. Blender Foundation, Stichting Blender Foundation, Amsterdam, 2018.
- [40] (<https://www.mathworks.com/matlabcentral/fileexchange/5355-toolbox-graph>), MATLAB Central File Exchange. Retrieved July 15, 2024.
- [41] Marco Palombo, Clémence Ligneul, Chloé Najac, Juliette Le Douce, Julien Flament, Carole Escartin, Philippe Hantraye, Emmanuel Brouillet, Gilles Bonvento, and Julien Valette. New paradigm to assess brain cell morphology by diffusion-weighted mr spectroscopy in vivo. *Proceedings of the National Academy of Sciences*, 113(24):6671–6676, 2016.
- [42] Andrada Ianus, Daniel C Alexander, Hui Zhang, and Marco Palombo. Mapping complex cell morphology in the grey matter with double diffusion encoding mr: A simulation study. *Neuroimage*, 241:118424, 2021.
- [43] Mikkel B Hansen, Sune N Jespersen, Lindsey A Leigland, and Christopher D Kroenke. Using diffusion anisotropy to characterize neuronal morphology in gray matter: the orientation distribution of axons and dendrites in the neuromorpho.org database. *Frontiers in integrative neuroscience*, 7:31, 2013.
- [44] Jonas Lyng Olesen and Sune Nørhøj Jespersen. Stick power law scaling in neurons withstands realistic curvature and branching. In *International Society for Magnetic Resonance in Medicine Annual Meeting*, 2020.
- [45] Markus Nilsson, Jimmy Lätt, Freddy Ståhlberg, Danielle van Westen, and Håkan Hagslätt. The importance of axonal undulation in diffusion mr measurements: a monte carlo simulation study. *NMR in Biomedicine*, 25(5):795–805, 2012.
- [46] Jan Brabec, Samo Lasič, and Markus Nilsson. Time-dependent diffusion in undulating thin fibers: Impact on axon diameter estimation. *NMR in Biomedicine*, 33(3):e4187, 2020.
- [47] Dmitry S Novikov and Valerij G Kiselev. Surface-to-volume ratio with oscillating gradients. *Journal of magnetic resonance*, 210(1):141–145, 2011.
- [48] Hong-Hsi Lee, Antonios Papaioannou, Sung-Lyoung Kim, Dmitry S Novikov, and Els Fieremans. A time-dependent diffusion mri signature of axon caliber variations and beading. *Communications biology*, 3(1):354, 2020.

- [49] Dmitry S Novikov, Jens H Jensen, Joseph A Helpert, and Els Fieremans. Revealing mesoscopic structural universality with diffusion. *Proceedings of the National Academy of Sciences*, 111(14):5088–5093, 2014.
- [50] Marco Palombo, Daniel C Alexander, and Hui Zhang. A generative model of realistic brain cells with application to numerical simulation of the diffusion-weighted mr signal. *NeuroImage*, 188:391–402, 2019.
- [51] Sergio Luengo-Sanchez, Concha Bielza, Ruth Benavides-Piccione, Isabel Fernaud-Espinosa, Javier DeFelipe, and Pedro Larrañaga. A univocal definition of the neuronal soma morphology using gaussian mixture models. *Frontiers in neuroanatomy*, 9:137, 2015.
- [52] Marwan Abdellah, Juan Hernando, Stefan Eilemann, Samuel Lapere, Nicolas Antille, Henry Markram, and Felix Schürmann. Neuromorphovis: a collaborative framework for analysis and visualization of neuronal morphology skeletons reconstructed from microscopy stacks. *Bioinformatics*, 34(13):i574–i582, 2018.
- [53] Marwan Abdellah, Juan José García Cantero, Nadir Román Guerrero, Alessandro Foni, Jay S Coggan, Corrado Calì, Marco Agus, Eleftherios Zisis, Daniel Keller, Markus Hadwiger, et al. Ultralizer: a framework for creating multiscale, high-fidelity and geometrically realistic 3d models for in silico neuroscience. *Briefings in bioinformatics*, 24(1):bbac491, 2023.
- [54] Dmitry S Novikov, Jelle Veraart, Ileana O Jelescu, and Els Fieremans. Rotationally-invariant mapping of scalar and orientational metrics of neuronal microstructure with diffusion mri. *NeuroImage*, 174:518–538, 2018.
- [55] Kanti V Mardia and Peter E Jupp. *Directional statistics*. John Wiley & Sons, 2009.
- [56] Gerhard Kurz, Igor Gilitschenski, Florian Pfaff, Lukas Drude, Uwe D. Hanebeck, Reinhold Haeb-Umbach, and Roland Y. Siegwart. Directional statistics and filtering using libDirectional. *Journal of Statistical Software*, 89(4):1–31, 2019.
- [57] Lida Kanari, Paweł Dłotko, Martina Scolamiero, Ran Levi, Julian Shillcock, Kathryn Hess, and Henry Markram. A topological representation of branching neuronal morphologies. *Neuroinformatics*, 16:3–13, 2018.
- [58] Lida Kanari, Hugo Dictus, Athanassia Chalimourda, Alexis Arnaudon, Werner Van Geit, Benoit Coste, Julian Shillcock, Kathryn Hess, and Henry Markram. Computational synthesis of cortical dendritic morphologies. *Cell Reports*, 39(1), 2022.
- [59] Lida Kanari, Adélie Garin, and Kathryn Hess. From trees to barcodes and back again: theoretical and statistical perspectives. *Algorithms*, 13(12):335, 2020.

- [60] Donghan M. Yang, James E. Huettner, G. Larry Bretthorst, Jeffrey J. Neil, Joel R. Garbow, and Joseph J.H. Ackerman. Intracellular water preexchange lifetime in neurons and astrocytes. *Magnetic Resonance in Medicine*, 79:1616–1627, March 2018.
- [61] Nathan H Williamson, Rea Ravin, Dan Benjamini, Hellmut Merkle, Melanie Falgairolle, Michael James O’Donovan, Dvir Blivis, Dave Ide, Teddy X Cai, Nima S Ghorashi, Ruiliang Bai, and Peter J Basser. Magnetic resonance measurements of cellular and sub-cellular membrane structures in live and fixed neural tissue. *eLife*, 8:e51101, December 2019.
- [62] Teddy X. Cai, Nathan H. Williamson, Rea Ravin, and Peter J. Basser. Disentangling the Effects of Restriction and Exchange With Diffusion Exchange Spectroscopy. *Frontiers in Physics*, 10:805793, March 2022.
- [63] Eloïse Mougel, Julien Valette, and Marco Palombo. Investigating exchange, structural disorder, and restriction in gray matter via water and metabolites diffusivity and kurtosis time-dependence. *Imaging Neuroscience*, 2:1–14, 04 2024.
- [64] Narayanan Kasthuri, Kenneth Jeffrey Hayworth, Daniel Raimund Berger, Richard Lee Schalek, José Angel Conchello, Seymour Knowles-Barley, Dongil Lee, Amelio Vázquez-Reina, Verena Kaynig, Thouis Raymond Jones, Mike Roberts, Josh Lyskowski Morgan, Juan Carlos Tapia, H. Sebastian Seung, William Gray Roncal, Joshua Tzvi Vogelstein, Randal Burns, Daniel Lewis Sussman, Carey Eldin Priebe, Hanspeter Pfister, and Jeff William Lichtman. Saturated reconstruction of a volume of neocortex. *Cell*, 162(3):648–661, 2015.
- [65] Marco Palombo, Clemence Ligneul, Edwin Hernandez-Garzon, and Julien Valette. Can we detect the effect of spines and leaflets on the diffusion of brain intracellular metabolites? *NeuroImage*, 182:283–293, 2018.
- [66] Kadir Şimşek and Marco Palombo. Diffusion in dendritic spines: impact on permeative exchange estimation with time-dependent diffusion-weighted mri. *Proc. Intl. Soc. Mag. Reson. Med. 2024.*, 3456, 2024.
- [67] Arthur Chakwizira, Kadir Şimşek, Marco Palombo, Filip Szczepankiewicz, Linda Knutsson, and Markus Nilsson. Water exchange as measured by diffusion mri with free gradient waveforms: A potential biomarker of dendritic spine morphology. *Proc. Intl. Soc. Mag. Reson. Med. 2024.*, 3463, 2024.
- [68] Arthur Chakwizira, Carl-Fredrik Westin, Jan Brabec, Samo Lasič, Linda Knutsson, Filip Szczepankiewicz, and Markus Nilsson. Diffusion MRI with pulsed and free gradient waveforms: Effects of restricted diffusion and exchange. *NMR in Biomedicine*, 36:e4827, January 2023.
- [69] Paul T Callaghan. *Principles of nuclear magnetic resonance microscopy*. Clarendon press, 1993.

- [70] Evren Özarlan, Cem Yolcu, Magnus Herberthson, Hans Knutsson, and Carl-Fredrik Westin. Influence of the size and curvature of neural projections on the orientationally averaged diffusion mr signal. *Frontiers in physics*, 6:17, 2018.
- [71] Markus Nilsson, Samo Lasič, Ivana Drobnjak, Daniel Topgaard, and Carl-Fredrik Westin. Resolution limit of cylinder diameter estimation by diffusion mri: The impact of gradient waveform and orientation dispersion. *NMR in Biomedicine*, 30(7):e3711, 2017.
- [72] Nian Wang, Jieying Zhang, Gary Cofer, Yi Qi, Robert J. Anderson, Leonard E. White, and G. Allan Johnson. Neurite orientation dispersion and density imaging of mouse brain microstructure. *Brain Structure and Function*, 224:1797–1813, 2019.
- [73] Nian Wang, Leonard E. White, Yi Qi, Gary Cofer, and G. Allan Johnson. Cytoarchitecture of the mouse brain by high resolution diffusion magnetic resonance imaging. *NeuroImage*, 216:116876, 2020.
- [74] G. Allan Johnson, Yuqi Tian, David G. Ashbrook, Gary P. Cofer, James J. Cook, James C. Gee, Adam Hall, Kathryn Hornburg, Catherine C. Kaczorowski, Yi Qi, Fang-Cheng Yeh, Nian Wang, Leonard E. White, and Robert W. Williams. Merged magnetic resonance and light sheet microscopy of the whole mouse brain. *Proceedings of the National Academy of Sciences*, 120(17):e2218617120, 2023.
- [75] Nian Wang, Surendra Maharjan, Andy P. Tsai, Peter B. Lin, Yi Qi, Abigail Wallace, Megan Jewett, Fang Liu, Gary E. Landreth, and Adrian L. Oblak. Integrating multimodality magnetic resonance imaging to the allen mouse brain common coordinate framework. *NMR in Biomedicine*, 36(5):e4887, 2023.
- [76] Xinyue Han, Surendra Maharjan, Jie Chen, Yi Zhao, Yi Qi, Leonard E. White, G. Allan Johnson, and Nian Wang. High-resolution diffusion magnetic resonance imaging and spatial-transcriptomic in developing mouse brain. *NeuroImage*, 297:120734, 2024.
- [77] Nicholas L. Turner, Thomas Macrina, J. Alexander Bae, Runzhe Yang, Alyssa M. Wilson, Casey Schneider-Mizell, Kisuk Lee, Ran Lu, Jingpeng Wu, Agnes L. Bodor, Adam A. Bleckert, Derrick Brittain, Emmanouil Froudarakis, Sven Dorkenwald, Forrest Collman, Nico Kemnitz, Dodam Ih, William M. Silversmith, Jonathan Zung, Aleksandar Zlateski, Ignacio Tartavull, Szi-chieh Yu, Sergiy Popovych, Shang Mu, William Wong, Chris S. Jordan, Manuel Castro, JoAnn Buchanan, Daniel J. Bumbarger, Marc Takeno, Russel Torres, Gayathri Mahalingam, Leila Elabbady, Yang Li, Erick Cobos, Pengcheng Zhou, Shelby Suckow, Lynne Becker, Liam Paninski, Franck Polleux, Jacob Reimer, Andreas S. Tolias, R. Clay Reid, Nuno Maçarico da Costa, and H. Sebastian Seung. Reconstruction of

neocortex: Organelles, compartments, cells, circuits, and activity. *Cell*, 185(6):1082–1100.e24, Mar 2022.

- [78] Alexander Shapson-Coe, Michał Januszewski, Daniel R. Berger, Art Pope, Yuelong Wu, Tim Blakely, Richard L. Schalek, Peter H. Li, Shuohong Wang, Jeremy Maitin-Shepard, Neha Karlupia, Sven Dorkenwald, Evelina Sjostedt, Laramie Leavitt, Dongil Lee, Jakob Troidl, Forrest Collman, Luke Bailey, Angerica Fitzmaurice, Rohin Kar, Benjamin Field, Hank Wu, Julian Wagner-Carena, David Aley, Joanna Lau, Zudi Lin, Donglai Wei, Hanspeter Pfister, Adi Peleg, Viren Jain, and Jeff W. Lichtman. A petavoxel fragment of human cerebral cortex reconstructed at nanoscale resolution. *Science*, 384(6696):eadk4858, 2024.

6 Supplementary

Figure 9: Structural feature distribution for all species

Figure 10: Spearman's rank correlation between structural features for all species and cell types Cell colour indicates the strength of the correlation and black cells indicate correlations that were found not to be significant (p-value adjusted according to Bonferroni correction)